# Minibatch vs Local SGD for Heterogeneous Distributed Learning

**Blake Woodworth**
Toyota Technological
Institute at Chicago
`blake@ttic.edu`

**Kumar Kshitij Patel**
Toyota Technological
Institute at Chicago
`kkpatel@ttic.edu`

**Nathan Srebro**
Toyota Technological
Institute at Chicago
`nati@ttic.edu`

## Abstract

We analyze Local SGD (aka parallel or federated SGD) and Minibatch SGD in the heterogeneous distributed setting, where each machine has access to stochastic gradient estimates for a different, machine-specific, convex objective; the goal is to optimize w.r.t. the average objective; and machines can only communicate intermittently. We argue that, (i) Minibatch SGD (even without acceleration) dominates all existing analysis of Local SGD in this setting, (ii) accelerated Minibatch SGD is optimal when the heterogeneity is high, and (iii) present the first upper bound for Local SGD that improves over Minibatch SGD in a non-homogeneous regime.

## 1   Introduction

Given the massive scale of many modern machine learning models and datasets, it has become important to develop better methods for distributed training. A particularly important setting for distributed stochastic optimization/learning, and the one we will consider in this work is characterized by, (1) training data that is distributed across many parallel devices rather than centralized in a single node; (2) this data is distributed *heterogeneously*, meaning that each individual machine has data drawn from a *different* distribution; and (3) the frequency of communication between the devices is limited. The goal is to find a single consensus predictor that performs well on all the local distributions simultaneously [3, 4]. The heterogeneity of the data significantly increases the difficulty of distributed learning because the machines' local objectives may be completely different, so a perfect model for one distribution might be terrible for all the others. Limited communication between devices can make it even more challenging to find a good consensus.

One possible approach is using Minibatch Stochastic Gradient Descent (SGD). Between communications, each machine computes one large minibatch stochastic gradient using its local data; then the machines average their local minibatch gradients, yielding one extra-large minibatch gradient comprising data from all the machines' local distributions, which is used for a single SGD update. Minibatch SGD can also be accelerated to improve its convergence rate [8, 9]. This algorithm is simple, ubiquitous, and performs very successfully in a variety of settings.

However, there has recently been great interest in another algorithm, Local SGD (also known as Parallel SGD or Federated SGD) [14, 20, 27], which has been suggested as an improvement over the naïve approach of Minibatch SGD. For Local SGD, each machine independently runs SGD on its local objective and, each time they communicate, the machines average together their local iterates. Local SGD is a very appealing approach—unlike Minibatch SGD, each machine is constantly improving its local model's performance on the local objective, even when the machines are not communicating with each other, so the number of updates is decoupled from the number of communications. In addition to the intuitive benefits, Local SGD has also performed well in many applications [13, 25, 26].

But can we show that Local SGD is in fact better then Minibatch SGD for convex, heterogeneous objectives? Does it enjoy better guarantees, and in what settings? To answer these questions we need to analyze the performance of Local SGD and Minibach SGD.

A number of recent papers have analyzed the convergence properties of Local SGD in the heterogeneous data setting [e.g. 2, 11, 12, 22]. But, as we will discuss, none of these Local SGD guarantees can show improvement over Minibatch SGD, even without acceleration, and in many regimes they are much worse. Is this a weakness of the analysis? Can the bounds be improved to show that Local SGD is actually better than Minibatch SGD in certain regimes? Or even at least as good?

Until recently, the situation was similar also for homogeneous distributed optimization, where all the machines' data distributions are the same. There were many published analyses of Local SGD, none of which showed any improvement over Minibatch SGD, or even matched Minibatch SGD's performance. Only recently Woodworth et al. [23] settled the issue for the homogeneous case, showing that on the one hand, when communication is rare, Local SGD provably outperforms even accelerated Minibatch SGD, but on the other, Local SGD does not always match Minibatch SGD's performance guarantees and in some situations is provably *worse* than Minibatch SGD.

How does this situation play out in the more challenging, and perhaps more interesting, heterogeneous setting? Some have suggested that the more difficult heterogeneous setting is where we should expect Local SGD to really shine, and where its analysis becomes even more relevant. So, how does heterogeneity affect both Local SGD and Minibatch SGD, and the comparison between them? Do any existing analyses of Local SGD show improvement over Minibatch SGD in the heterogeneous setting? Is Local SGD still better than Minibatch SGD when communication is rare? Does the added complexity of heterogeneity perhaps necessitate the more sophisticated Local SGD approach, as some have suggested? What is the optimal method in this more challenging setting?

In fact, Karimireddy et al. [11] recently argued that heterogeneity can be particularly problematic for Local-SGD, proving a lower bound for it that shows degradation as heterogeneity increases. In Section 4, we discuss how this lower bound implies that Local SGD is strictly *worse* than Minibatch SGD when the level of heterogeneity is very large. However, even with Karimireddy et al.'s lower bound, it is not clear whether or not Local SGD can improve over Minibatch SGD for merely moderately heterogeneous objectives.

In this paper, we expand on Karimireddy et al.'s observation about the ill-suitability of Local SGD to the heterogeneous setting. We prove that existing analysis of Local SGD for heterogeneous data cannot be substantially improved unless the setting is very near-homogeneous. This is disappointing for Local SGD, because it indicates that unless the level of heterogeneity is very low, the performance it can ensure is truly worse than Minibatch SGD, even without acceleration and regardless of the frequency of communication. At the same time, we provide a more refined analysis of Local SGD showing that Local SGD does, in fact, improve over Minibatch SGD when the level of heterogeneity is sufficiently small (i.e. the problem is not exactly homogeneous, but it is at least near-homogeneous). This is the first result to show that Local SGD improves over Minibatch SGD in *any* non-homogeneous setting.

We further show that with even moderately high heterogeneity (or when we do not restrict the dissimilarity between machines), Accelerated Minibatch SGD is in fact optimal for heterogeneous stochastic distributed optimization! This is because, as we show, Minibatch SGD and its accelerated variant are immune to the heterogeneity of the problem. Perhaps the most important conclusion of our study is that we identify a regime, in which the data is moderately heterogeneous, where Accelerated Minibatch SGD may *not* be optimal and where Local SGD is certainly worse than Minibatch SGD, and so new methods may be needed.

## 2  Setup

We consider heterogeneous distributed stochastic convex optimization/learning using $M$ machines, each of which has access to its own data distribution $\mathcal{D}^m$. The goal is to find an approximate minimizer of the average of the local objectives:

$$\min_{x \in \mathbb{R}^d} F(x) := \frac{1}{M} \sum_{m=1}^{M} F_m(x) := \frac{1}{M} \sum_{m=1}^{M} \mathbb{E}_{z^m \sim \mathcal{D}^m} f(x; z^m) \tag{1}$$

This objective captures, for example, supervised learning where $z = (z_{\text{features}}, z_{\text{label}})$ and $f(x; z)$ is the loss of the predictor, parametrized by $x$, on the instance $z$. The per-machine distribution $\mathcal{D}^m$ can be thought of as the empirical distribution of data on machine $m$, or as source distribution which varies between servers, regions or devices.

We focus on a setting in which each machine performs local computations using samples from its own distribution, and is able to communicate with other machines periodically in order to build consensus. This situation arises, for example, when communication is expensive relative to local computation, so it is advantageous to limit the frequency of communication to improve performance.

Concretely, we consider distributed first-order algorithms where each machine computes a total of $T$ stochastic gradients, and is limited to communicate with the others $R$ times. We divide the optimization process into $R$ rounds, where each round consists of each machine calculating and processing $K = T/R$ stochastic gradients, and then communicating with all other machines[1]. Each stochastic gradient for machine $m$ is given by $\nabla f(x; z^m)$ for an independent $z^m \sim \mathcal{D}^m$.

A simple algorithm for this setting is **Minibatch SGD**. During each round, each machine computes $K$ stochastic gradients $\{g^m_{r,k}\}_{k\in[K]}$ at the same point $x_r$, and communicates its average $g^m_r$. Then, the averages from all machines are averaged, to obtain an estimate $g_r$ of the gradient of the overall objective. The estimate $g_r$ is based on all $KM$ stochastic gradients, and is used to obtain the iterate $x_{r+1}$. Overall, we perform $R$ steps of SGD, each step based on a mini-batch of $KM$ (non-i.i.d.) samples. To summarize, initializing at some $x_0$, Minibatch SGD operates as follows,

$$g^m_{r,k} = \nabla f(x_r; z^m_{r,k}), \;\; z^m_{r,k} \sim \mathcal{D}^m, \;\; m = 1 \dots M, k = 0 \dots K - 1,$$

$$g_r = \frac{1}{M} \sum_{m=1}^{M} g^m_r \quad \text{where } g^m_r = \frac{1}{K} \sum_{k=1}^{K} g^m_{r,k}, \tag{2}$$

$$x_{r+1} = x_r - \eta_r g_r,$$

Alternatively, we can perform the same stochastic gradient calculation and aggregation of $g_r$ but replace the simple SGD updates with more sophisticated updates and carefully tuned momentum parameters. Throughout this paper, we use "**Accelerated Minibatch SGD**" to refer to two accelerated variants of SGD: AC-SA [8] for convex objectives, and multi-stage AC-SA [9] for strongly convex objectives. Algorithmic details including pseudo-code are provided in Appendix C.2.

Unlike Minibatch SGD, where each machine spends an entire round calculating stochastic gradients at the same point, **Local SGD** allows the machines to update local iterates throughout the round based on their own stochastic gradient estimates. Each round starts with a common iterate on all machines, then each machine executes $K$ steps of SGD on its own local objective and communicates the final iterate. These iterates are averaged to form the starting point for the next round. Using $x^m_{r,k}$ to denote the iterate after $r$ rounds and $k$ local steps on machine $m$ and initializing at $x^m_{0,0} = x_0$ for all $m \in [M]$, Local SGD operates as follows,

$$g^m_{r,k} := \nabla f(x^m_{r,k}; z^m_{r,k}), \;\; z^m_{r,k} \sim \mathcal{D}^m, \;\; m = 1 \dots M, k = 0 \dots K - 1,$$

$$x^m_{r,k+1} = x^m_{r,k} - \eta_{r,k} g^m_{r,k} \tag{3}$$

$$x^m_{r+1,0} = x_{r+1} := \tfrac{1}{M} \sum_{m=1}^{M} x^m_{r,K}.$$

**An alternative viewpoint: reducing communication.** Another way of viewing the problem is as follows: consider as a baseline processing $T$ stochastic gradient on each machine, but communicating at every step, i.e. $T$ times, thus implementing $T$ steps of SGD, using a mini-batch of $M$ samples (one from each machine). Can we achieve the same performance as this baseline while communicating less frequently? Communicating only $R$ times instead of $T$ precisely brings us back to the model we are considering, and all the methods discussed above ($R$ steps of MB-SGD with mini-batches of size $KM = TM/R$, or Local SGD with $T$ steps per machine and $R$ averaging steps) reduce the communication. Checking that $R < T$ rounds achieve the same accuracy as the dense communication baseline is a starting point, but the question is how small can we push $R$ (while keeping $T = KR$ fixed) before accuracy degrades. Better error guarantees (in terms of $K$, $M$ and $R$) mean we can use a smaller $R$ with less degradation, and the smallest $R$ with no asymptotic degradation can be directly calculated from the error guarantee [see, e.g., discussion in 6].

In order to prove convergence guarantees, we rely on several assumptions about the problem. Central to our analysis will be a parameter $\zeta_*^2$ which, in some sense, describes the level of heterogeneity in the problem. It is possible to analyze heterogeneous distributed optimization without any bound on the relatedness of the local objective—this is the typical setup in the consensus optimization literature [e.g. 4, 15, 16, 19] and indeed our analysis of Minibatch SGD does not rely on this parameter. Nevertheless, such an assumption *is* required by the existing analyses of Local SGD [2, 11, 12] which we would like to compare to, and, as we will show, is in fact necessary for the convergence of Local SGD. Following prior work [11, 12], we define

$$\zeta_*^2 = \tfrac{1}{M} \sum\nolimits_{m=1}^{M} \left\| \nabla F_m(x^*) \right\|^2 \tag{4}$$

Since $\tfrac{1}{M} \sum_{m=1}^{M} \nabla F_m(x^*) = \nabla F(x^*) = 0$, this captures, in some sense, the variation in the local gradients *at the optimum*. When $\zeta_*^2 = 0$, all $F_m$ share at least one minimizer, and when $\zeta_*^2$ is large, there is great disagreement between the local objectives. While homogeneous objectives (i.e. $F_m = F$) have $\zeta_*^2 = 0$, the converse is not true! Even when $\zeta_*^2 = 0$, $\nabla F_m(x)$ might be different than $\nabla F_n(x)$ for $x \neq x^*$.

Throughout, we assume that $f(\cdot; z)$ is $H$-**smooth** for all $z$ meaning

$$f(y; z) \leq f(x; z) + \langle \nabla f(x; z), \, y - x \rangle + \tfrac{H}{2} \left\| x - y \right\|^2 \qquad \forall_{x,y,z}, \tag{5}$$

We assume the **variance of the stochastic gradients** on each machine is bounded, either uniformly

$$\mathbb{E}_{z^m \sim \mathcal{D}^m} \left\| \nabla f(x; z^m) - \nabla F_m(x) \right\|^2 \leq \sigma_m^2 \quad \forall_{x,m} \quad \text{and} \quad \sigma^2 = \tfrac{1}{M} \sum\nolimits_{m=1}^{M} \sigma_m^2 \tag{6}$$

or **only at the optimum** $x^*$, i.e.

$$\mathbb{E}_{z^m \sim \mathcal{D}^m} \left\| \nabla f(x^*; z^m) - \nabla F_m(x^*) \right\|^2 \leq \sigma_{*,m}^2. \quad \forall_m \quad \text{and} \quad \sigma_*^2 = \tfrac{1}{M} \sum\nolimits_{m=1}^{M} \sigma_{*,m}^2 \tag{7}$$

We consider guarantees of two forms: For **strongly convex local objectives**, we consider guarantees that depend on the **parameter of strong convexity**, $\lambda$, for all the machines' objectives,

$$F_m(y) \geq F_m(x) + \langle \nabla F_m(x), \, y - x \rangle + \tfrac{\lambda}{2} \left\| x - y \right\|^2 \qquad \forall_{x,y,m} \tag{8}$$

as well as the **initial sub-optimality** $F(0) - F(x^*) \leq \Delta$, besides the smoothness, heterogeneity bound, and variance as discussed above.

We also consider guarantees for **weakly convex objectives** that just rely on each local objective $F_m$ being convex (not necessarily strongly), as well as a bound on the norm of the optimum $\|x^*\| \leq B$, besides the smoothness, homogeneity bound, and variance as discussed above.

## 3 Minibatch SGD and Accelerated Minibatch SGD

To begin, we analyze the worst-case error of Minibatch and Accelerated Minibatch SGD in the heterogeneous setting. A simple observation is that, despite the heterogeneity of the objective, the minibatch gradients $g_r$ (2) are unbiased estimates of $\nabla F$, i.e. the overall objective's gradient:

$$\mathbb{E} g_r = \mathbb{E} \left[ \frac{1}{MK} \sum_{m=1}^{M} \sum_{k=1}^{K} \nabla f(x_r; z_{r,k}^m) \right] = \frac{1}{MK} \sum_{m=1}^{M} \sum_{k=1}^{K} \nabla F_m(x_r) = \nabla F(x_r) \tag{9}$$

We are therefore updating using unbiased estimates of $\nabla F$, and can appeal to standard analysis for (accelerated) SGD. To do so, we calculate the variance of these estimates:

$$\mathbb{E} \left\| g_r - \nabla F(x_r) \right\|^2 = \mathbb{E} \left\| \frac{1}{MK} \sum_{m=1}^{M} \sum_{k=1}^{K} \nabla f(x_r; z_{r,k}^m) - \nabla F(x_r) \right\|^2 \tag{10}$$

$$= \frac{1}{M^2 K^2} \sum_{m=1}^{M} \sum_{k=1}^{K} \mathbb{E} \left\| \nabla f(x_r; z_{r,k}^m) - \nabla F_m(x_r) \right\|^2 \leq \frac{\sigma^2}{MK} \tag{11}$$

Interestingly, the variance is always reduced by $MK$ and is not effected by the heterogeneity $\zeta_*$. Plugging this calculation[2] into the analysis of SGD (see details in Appendix C) yields:

**Theorem 1.** *A weighted average of the Minibatch SGD iterates satisfies for a universal constant $c$*

$$\mathbb{E}F(\hat{x}) - F^* \leq c \cdot \left( \frac{HB^2}{R} + \frac{\sigma_* B}{\sqrt{MKR}} \right) \qquad \text{\textit{under the convex assumptions,}}$$

$$\mathbb{E}F(\hat{x}) - F^* \leq c \cdot \left( \frac{H\Delta}{\lambda} \exp\left( -\frac{\lambda R}{2H} \right) + \frac{\sigma_*^2}{\lambda MKR} \right) \qquad \text{\textit{under the strongly convex assumptions.}}$$

*and Accelerated Minibatch SGD[3] guarantees*

$$\mathbb{E}F(\hat{x}) - F^* \leq c \cdot \left( \frac{HB^2}{R^2} + \frac{\sigma B}{\sqrt{MKR}} \right) \qquad \text{\textit{under the convex assumptions,}}$$

$$\mathbb{E}F(\hat{x}) - F^* \leq c \cdot \left( \Delta \exp\left( -\frac{\sqrt{\lambda} R}{c' \sqrt{H}} \right) + \frac{\sigma^2}{\lambda MKR} \right) \qquad \text{\textit{under the strongly convex assumptions.}}$$

The most important feature of these guarantees is that they are completely independent of $\zeta_*^2$. Since these upper bounds are known to be tight in the homogeneous case (where $\zeta_*^2 = 0$) [17, 18], this means that both algorithms are essentially immune to the heterogeneity of the problem, performing equally well for homogeneous objectives as they do for arbitrarily heterogeneous ones.

## 4   Local SGD for heterogeneous data

Recently, Bayoumi et al. [2] and Koloskova et al. [12] analyzed Local SGD for the heterogeneous data setting. Their guarantees in the convex case along with the analysis of (Accelerated) Minibatch SGD are summarized in Table 1, Table 2 in Appendix A shows the comparison in the strongly convex case. SCAFFOLD[4] [11], a related method for heterogeneous distributed optimization, is also included.

Upon inspection, among the previously published upper bounds for heterogeneous Local SGD (that we are aware of), Koloskova et al.'s is the tightest, and dominates the others. However, even this guarantee is the sum of the Minibatch SGD bound plus additional terms, and is thus worse in every regime, and cannot show improvement over Minibatch SGD. But does this reflect a weakness of their analysis, or a true weakness of Local SGD? Indeed, Woodworth et al. [23] showed a tighter upper bound for Local SGD *in the homogeneous case*, which improves over Koloskova et al. [12] (for $\zeta_* = 0$) and *does* show improvement over Minibatch SGD when communication is infrequent. But can we generalize Woodworth et al.'s bound also to the heterogeneous case? Optimistically, we might hope that the $(\zeta_*/R)^{2/3}$ dependence on heterogeneity in these bounds could be improved. After all, Minibatch SGD's rate is independent of $\zeta_*^2$, so perhaps Local SGD's could be, too?

Unfortunately, it is already known that some dependence on $\zeta_*$ is necessary, as Karimireddy et al. [11] have shown a lower bound[5] of $\zeta_*^2/(\lambda R^2)$ in the strongly convex case, which suggests a lower bound of $\zeta_* B/R$ in the convex case. But perhaps the Koloskova et al. analysis can be improved to match this bound? If the $\zeta_* B/R$ term from Karimireddy et al.'s lower bound were possible, it would be lower order than $HB^2/R$ for $\zeta_* < HB$, and we would see no slow-down until the level of heterogeneity is fairly large. On the other hand, if the $(\zeta_*/R)^{2/3}$ term from Koloskova et al. cannot be improved, then we see a slowdown as soon as $\zeta_* = \Omega(HB/R)$, i.e. even for very small $\zeta_*$!

We now show that the poor dependence on $\zeta_*$ from the Koloskova et al. analysis cannot be improved. Consequently, for sufficiently heterogeneous data, Local SGD is strictly worse than Minibatch SGD, regardless of the frequency of communication, unless the level of heterogeneity is very small.

| Method/Analysis | Worst-Case Error (i.e. $\mathbb{E}F(\hat{x}) - F^* \lesssim$) |
|---|---|
| Minibatch SGD <br> Theorem 1 | $\frac{HB^2}{R} + \frac{\sigma_* B}{\sqrt{MKR}}$ |
| Accelerated Minibatch SGD <br> Theorem 1 | $\frac{HB^2}{R^2} + \frac{\sigma B}{\sqrt{MKR}}$ |
| Local SGD <br> Koloskova et al. [12] | $\frac{HB^2}{R} + \frac{\sigma_* B}{\sqrt{MKR}} + \frac{\left(H\zeta_*^2 B^4\right)^{1/3}}{R^{2/3}} + \frac{\left(H\sigma_*^2 B^4\right)^{1/3}}{K^{1/3}R^{2/3}}$ |
| Local SGD <br> Bayoumi et al. [2] | $\frac{HB^2}{R} + \frac{B\sqrt{\sigma_*^2+\zeta_*^2}}{\sqrt{MKR}} + \frac{\left(H(\sigma_*^2+\zeta_*^2)B^4\right)^{1/3}}{R^{2/3}}$ |
| SCAFFOLD <br> Karimireddy et al. [11] | $\frac{HB^2}{R} + \frac{\sigma B}{\sqrt{MKR}} + \frac{\zeta_*^2}{HR} + \frac{\sigma\zeta_*}{H\sqrt{MKR}}$ |
| Local SGD <br> Theorem 3 | $\frac{HB^2}{\mathbf{K}R} + \frac{\sigma_* B}{\sqrt{MKR}} + \frac{\left(H\bar{\zeta}^2 B^4\right)^{1/3}}{R^{2/3}} + \frac{\left(H\sigma^2 B^4\right)^{1/3}}{K^{1/3}R^{2/3}}$ |
| Local SGD Lower Bound <br> Theorem 2 | $\min\left\{\frac{HB^2}{R}, \frac{\left(H\zeta_*^2 B^4\right)^{1/3}}{R^{2/3}}\right\} + \frac{\sigma B}{\sqrt{MKR}} + \frac{\left(H\sigma^2 B^4\right)^{1/3}}{K^{2/3}R^{2/3}}$ |
| Algorithm-Independent Lower Bound <br> Theorem 4 | $\min\left\{\frac{HB^2}{R^2}, \frac{\zeta_*^2}{HR^2}\right\} + \frac{\sigma B}{\sqrt{MKR}}$ |

Table 1: Guarantees under the convex assumptions. See (12) for a definition and discussion of $\bar{\zeta}$.

**Theorem 2.** *For any $M$, $K$, and $R$ there exist objectives in four dimensions such that Local SGD initialized at zero and using any fixed stepsize $\eta$ will have suboptimality at least*

$$\mathbb{E}F(\hat{x}) - F^* \geq c \cdot \left( \min\left\{ \frac{HB^2}{R}, \frac{\left(H\zeta_*^2 B^4\right)^{1/3}}{R^{2/3}} \right\} + \frac{\left(H\sigma^2 B^4\right)^{1/3}}{K^{2/3}R^{2/3}} + \frac{\sigma B}{\sqrt{MKR}} \right)$$

$$\mathbb{E}F(\hat{x}) - F^* \geq c \cdot \left( \min\left\{ \Delta \exp\left(-\frac{6\lambda R}{H}\right), \frac{H\zeta_*^2}{\lambda^2 R^2} \right\} + \min\left\{ \Delta, \frac{H\sigma^2}{\lambda^2 K^2 R^2} \right\} + \frac{\sigma^2}{\lambda MKR} \right)$$

*under the convex and strongly convex assumptions (for $H \geq 16\lambda$), respectively.*

This is proven in Appendix D using a similar approach to a lower bound for the homogeneous case from Woodworth et al. [23], and it is conceptually similar to the lower bounds for heterogeneous objectives of Karimireddy et al. [11]. Koloskova et al. [12] also prove a lower bound, but specifically for 1-strongly convex objectives, which obscures the important role of the strong convexity parameter.

In the convex case, this lower bound closely resembles the upper bound of Koloskova et al. [12]. Focusing on the case $H = B = \sigma^2 = 1$ to emphasize the role of $\zeta_*^2$, the only gaps are between (i) a term $1/(K^{1/3}R^{2/3})$ vs $1/(K^{2/3}R^{2/3})$—a gap which also exists in the homogeneous case [23]—and (ii) another term $\max\left\{1/R, (\zeta_*/R)^{2/3}\right\}$ vs $\min\left\{1/R, (\zeta_*/R)^{2/3}\right\}$. Consequently, for $\zeta_*^2 \geq 1/R \Rightarrow (\zeta_*/R)^{2/3} \geq 1/R$, the lower bound shows that Local SGD has error at least $1/R + 1/\sqrt{MKR}$ and thus performs strictly worse than Minibatch, regardless of $K$. This is quite surprising— Local SGD is often suggested as an improvement over Minibatch SGD for the heterogeneous setting, yet we see that even a small degree of heterogeneity can make it much worse. Furthermore, increasing the duration of each round, $K$, is often thought of as more beneficial for Local SGD than Minibatch SGD, but the lower bound indicates it does little to help Local SGD in the heterogeneous setting. We make a qualitatively similar comparison for the strongly convex case in Appendix A.

The lower bounds indicate that it is not possible to radically improve over the Koloskova et al. analysis, and thus over Minibatch SGD for even moderate heterogeneity, without stronger assumptions. In order to obtain an improvement over Minibatch SGD in a heterogeneous setting, at least with very low heterogeneity, we introduce a modification to the heterogeneity measure $\zeta_*$ which bounds the difference between the local objectives' gradients everywhere (not just at $x^*$):

$$\sup_x \frac{1}{M} \sum_{m=1}^M \|\nabla F_m(x) - \nabla F(x)\|^2 \leq \bar{\zeta}^2 \qquad (12)$$

This quantity precisely captures homogeneity since $\bar{\zeta}^2 = 0$ if and only if $F_m = F$ (up to an irrelevant additive constant). In terms of $\bar{\zeta}^2$, we are able to analyze Local SGD and see a smooth transition from the heterogeneous ($\bar{\zeta}^2$ large) to homogeneous ($\bar{\zeta}^2 = 0$) setting.

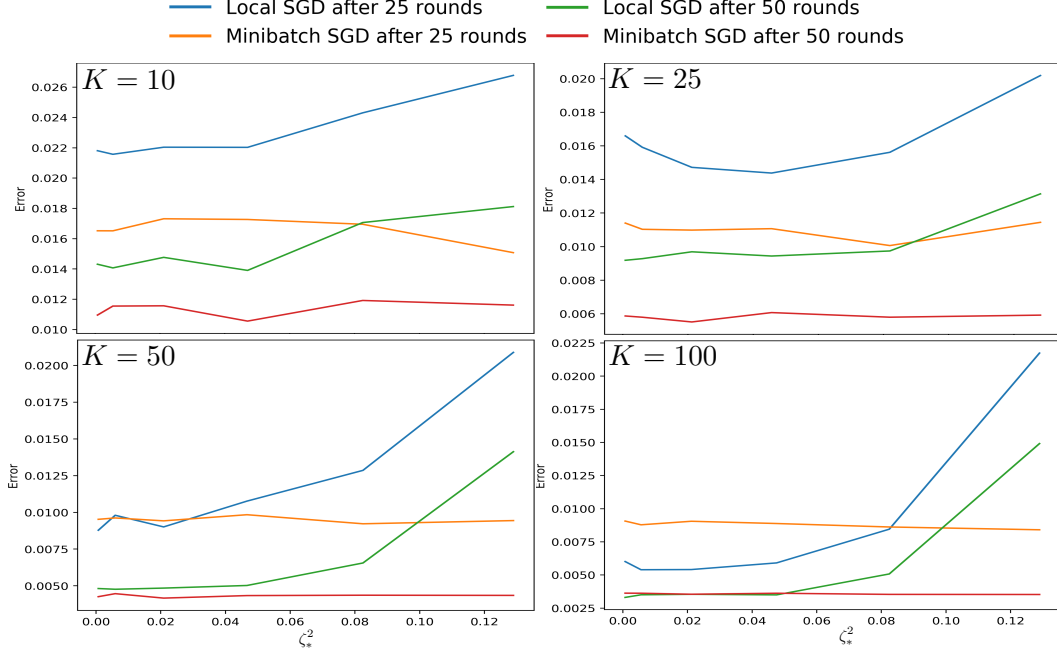

Figure 1: Binary logistic regression between even vs odd digits of MNIST. Twenty-five "tasks" were constructed, one for each combination of $i$ vs $j$ for even $i$ and odd $j$. For $p \in \{0, 20, 40, 60, 80, 100\}$, we assigned to each of $M = 25$ machines $p\%$ data from task $m$, and $(100 - p)\%$ data from a mixture of all tasks. For several choices of $R$ and $K$, we plot the error (averaged over four runs) versus the value of $\zeta_*^2$ resulting from each choice of $p$. For both algorithms, we used the best fixed stepsize for each choice of $K$, $R$, and $\zeta_*$ individually. Additional details are provided in Appendix F.

**Theorem 3.** *When* $\sup_x \frac{1}{M} \sum_{m=1}^M \|\nabla F_m(x) - \nabla F(x)\|^2 \leq \bar{\zeta}^2$, *an average of the Local SGD iterates guarantees under the convex and strongly convex assumptions, respectively*

$$\mathbb{E} F(\hat{x}) - F^* \leq c \cdot \left( \frac{HB^2}{KR} + \frac{\left(H\bar{\zeta}^2 B^4\right)^{1/3}}{R^{2/3}} + \frac{\left(H\sigma^2 B^4\right)^{1/3}}{K^{1/3} R^{2/3}} + \frac{\sigma_* B}{\sqrt{MKR}} \right),$$

$$\mathbb{E} F(\hat{x}) - F^* \leq c \cdot \left( \frac{H^2 B^2}{HKR + \lambda K^2 R^2} + \left( \frac{H\bar{\zeta}^2}{\lambda^2 R^2} + \frac{H\sigma^2}{\lambda^2 KR^2} \right) \log\left( \frac{H}{\lambda} + KR \right) + \frac{\sigma_*^2}{\lambda MKR} \right).$$

We prove the Theorem in Appendix E. This is the first analysis of Local SGD, or any other method for heterogeneous distributed optimization, showing any improvement over Minibatch SGD in any heterogeneous regime[6]. When $\bar{\zeta} = 0$, Theorem 3 reduces to the homogeneous analysis of Local SGD of Woodworth et al. [23, Theorem 2], which already showed that in that case, we see improvement when $K \gtrsim R$. Theorem 3 degrades smoothly when $\bar{\zeta} > 0$, and shows improvement for Local SGD over Minibatch SGD also when $\bar{\zeta}^2 \lesssim 1/R$ in the convex case, i.e. with low, yet positive, heterogeneity. It is yet unclear whether this rate of convergence can be ensured in terms of $\zeta_*^2$ rather than $\bar{\zeta}^2$.

**Experimental evidence** Finally, while Theorem 2 proves that Local SGD is worse than Minibatch SGD unless $\zeta_*^2$ is very small *in the worst case*, one might hope that for "normal" heterogeneous problems, Local SGD might perform better than its worst case error suggests. However, a simple binary logistic regression experiment on MNIST indicates that this behavior likely extends significantly beyond the worst case. The results, depicted in Figure 1, show that Local SGD performs worse than Minibatch SGD unless both $\zeta_*$ is very small and $K$ is large. Finally, we also observe that Minibatch SGD's performance is essentially unaffected by $\zeta_*$ empirically as predicted by theory.

# 5 Accelerated Minibatch SGD is optimal for highly heterogeneous data

In the previous section, we showed that Local SGD is strictly worse than Minibatch SGD whenever $\zeta_*^2 \geq HB/R$ (in the convex case). But perhaps a different method can improve over Accelerated Minibatch SGD? After all, Accelerated Minibatch SGD's convergence rate can only partially be improved by increasing $K$, the amount of local computation per round of communication. Even when $K \to \infty$, Accelerated Minibatch SGD is only guaranteed suboptimality $1/R^2$ or $\exp(-\lambda R)$. Can this rate be improved? Now, we show that the answer depends on the level of heterogeneity—unless $\zeta_*^2$ is sufficiently small, no algorithm can improve over Accelerated Minibatch SGD, which is optimal. To state the lower bound, we follow Carmon et al. [5] and define:

**Definition 1** (Distributed zero-respecting algorithm). *Let* $\text{supp}(v) = \{i \in [d] : v_i \neq 0\}$ *and* $\pi_m(t, m')$ *be the last time before $t$ when machines $m$ and $m'$ communicated. We say that an optimization algorithm is distributed zero-respecting if the $t$th query on the $m$th machine, $x_t^m$ satisfies,*

$$\text{supp}(x_t^m) \subseteq \bigcup_{s<t} \text{supp}(\nabla f(x_s^m; z_s^m)) \cup \bigcup_{m' \neq m} \bigcup_{s \leq \pi_m(t,m')} \text{supp}(\nabla f(x_s^{m'}; z_s^{m'})).$$

That is, the coordinates of a machine's iterates are non-zero only where gradients seen by that machine were non-zero. This encompasses many first-order optimization algorithms, including Minibatch SGD, Accelerated Minibatch SGD, Local SGD, coordinate descent methods, etc.

**Theorem 4.** *For any $M$, $K$, and $R$, there exist two quadratic objectives satisfying the convex and strongly convex assumptions (for $H \geq 7\lambda$) such that the output of any distributed zero-respecting algorithm will have suboptimality in the convex and strongly convex case respectively,*

$$F(\hat{x}) - F^* \geq c\left(\min\left\{\frac{\zeta_*^2}{HR^2}, \frac{HB^2}{R^2}\right\} + \frac{\sigma B}{\sqrt{MKR}}\right),$$

$$F(\hat{x}) - F^* \geq c\left(\min\left\{\frac{\lambda\zeta_*^2}{H^2}, \frac{\Delta\sqrt{\lambda}}{\sqrt{H}}\right\}\exp\left(-\frac{8R\sqrt{\lambda}}{\sqrt{H}}\right) + \frac{\sigma^2}{\lambda MKR}\right).$$

This is proven in Appendix G using techniques similar to those of Woodworth and Srebro [24] and Arjevani and Shamir [1]. However, care must be taken to control the parameter $\zeta_*$ for the hard instance and to see how this affects the final lower bound.

**Corollary 1.** *Accelerated Minibatch SGD is optimal when $\zeta_* \geq HB$ in the convex case, and is optimal up to log factors when $\zeta_*^2 \geq H^{3/2}/\sqrt{\lambda}$ in the strongly convex case.*

Thus, Accelerated Minibatch SGD is optimal for large $\zeta_*$, but when $\zeta_*$ is smaller, the lower bound does not match, and it might be possible to improve. Indeed, focusing on the convex case, there is a substantial regime $HB/\sqrt{R} \leq \zeta_* \leq HB$ in which it may or may not be possible to improve over Accelerated Minibatch SGD. Is it possible to improve in this regime, and if so, with what algorithm? From the previous section, we know that Local SGD is certainly *not* such an algorithm. Thus, an important question for future work is **what can be done when $\zeta_*$ is bounded, but not insignificant,** and what new algorithms may be able to improve over Accelerated Minibatch SGD in this regime?

# 6 Inner and outer stepsizes

In understanding the relationship between Minibatch SGD and Local SGD, and thinking of how to improve over them, it is useful to consider a unified method that interpolates between them. This involves taking SGD steps locally with one stepsize, and then when the machines communicate, they take a step in the resulting direction with a second, different stepsize. Such a dual-stepsize approach was already presented and analyzed as FEDAVG by Karimireddy et al. [11]. We will refer to these two stepsizes as "inner" and "outer" stepsizes, respectively, and consider

$$x_{r,k}^m = x_{r,k-1}^m - \eta_{\text{inner}} \nabla f(x_{r,k-1}^m; z_{r,k-1}^m) \qquad \forall_{m \in [M], k \in [K]}$$

$$x_{r+1,0}^m = x_{r,0}^m - \eta_{\text{outer}} \frac{1}{M} \sum_{n=1}^{M} \sum_{k=1}^{K} \nabla f(x_{r,k-1}^m; z_{r,k-1}^m) \qquad \forall_{m \in [M], r \in [R]}$$

(13)

Choosing $\eta_{\text{inner}} = 0$, this is equivalent to Minibatch SGD with stepsize $\eta_{\text{outer}}$, and choosing $\eta_{\text{inner}} = \eta_{\text{outer}}$ recovers Local SGD. Therefore, when the stepsizes are chosen optimally, this algorithm is always at least as good as both Minibatch and Local SGD achieving the minimum of Theorems 1 and 3—the former by choosing $\eta_{\text{inner}} = 0$ and the latter by choosing $\eta_{\text{inner}} = \eta_{\text{outer}}$. This analysis improves over Karimireddy et al.'s analysis of FEDAVG as we discuss in Appendix B. An important question is whether this method might be able to *improve* over both alternatives by choosing $0 \ll \eta_{\text{inner}} \ll \eta_{\text{outer}}$. Unfortunately, existing analysis does not resolve this question.

## 7 Using a Subset of Machines in Each Round

An interesting modification of our problem setting, recently formulated and studied by Karimireddy et al. [11], is a case in which only a random subset of $S \leq M$ machines are used in each round, as is the typical case in Federated Learning [10]. In the homogeneous case this makes no difference, since all machines are identical anyway. However, in a heterogeneous setting, having all machines participate at each round means that at each round we can obtain estimates of *all* components of the objective (1), which is very different from only seeing a different $S < M$ components each round, while still wanting to minimize the average of all $M$ components.

We could still consider Minibatch SGD in this setting, but in each round we would thus use a gradient estimate based on $SK$ stochastic gradients, $K$ from each of $S$ machines chosen uniformly at random (without replacement). This is still an unbiased estimator of the overall gradient $\nabla F(x_r)$, with variance $\frac{\sigma^2}{SK} + (1 - \frac{S}{M})\frac{\bar{\zeta}^2}{S}$ (and at $x^*$ the variance is $\frac{\sigma_*^2}{SK} + (1 - \frac{S}{M})\frac{\zeta_*^2}{S}$), guaranteeing the following:

$$\mathbb{E}F(\hat{x}) - F^* \leq O\left( \frac{HB^2}{R} + \frac{\sigma_* B}{\sqrt{SKR}} + \sqrt{1 - \frac{S}{M}} \cdot \frac{\zeta_* B}{\sqrt{SR}} \right), \tag{14}$$

$$\mathbb{E}F(\hat{x}) - F^* \leq O\left( \lambda B^2 \exp\left(\frac{-\lambda R}{H}\right) + \frac{\sigma_*^2}{\lambda SKR} + \left(1 - \frac{S}{M}\right) \cdot \frac{\zeta_*^2}{\lambda SR} \right). \tag{15}$$

Similar guarantees are also available for Accelerated Minibatch SGD with $\sigma$ and $\bar{\zeta}$ replacing $\sigma_*$ and $\zeta_*$. The guarantees (14) and (15) are valid also for Minibatch SGD (i.e. with $\eta_{\text{inner}} = 0$) and thus for the inner/outer stepsize updates (13). These guarantees improve over what was shown for the inner/outer stepsize variant by Karimireddy et al., but they also present SCAFFOLD which shows benefits over Minibatch SGD in some regimes under this setting (see discussion in Appendix B)

## 8 Discussion

Despite extensive efforts to analyze Local SGD both in homogeneous and heterogeneous settings, nearly all efforts have failed to show that Local SGD improves over Minibatch SGD in any situation. In fact, as far as we are aware, only Woodworth et al. [23] were able to show any improvement over Minibatch SGD in any regime, and their work is confined to the easier homogeneous setting. This raised the question of what happens in the heterogeneous case, which is substantially more difficult than the homogeneous setting, and where it is plausibly necessary to deviate from the quite naïve approach of (Accelerated) Minibatch SGD. In this paper, we conducted a careful analysis and comparison of Local and Minibatch SGD in the heterogeneous case and showed that moving from the homogeneous to heterogeneous setting significantly harms the performance of Local SGD. For instance, in the convex case Local SGD is strictly worse than Minibatch SGD unless $\zeta_*^2 < H^2 B^2/R$. This indicates that any benefits of Local over Minibatch SGD actually lie in the homogeneous or near-homogenous setting, and not in highly hetrogenous settings (unless additional assumptions are made). We also show similar benefits for Local SGD as those shown by Woodworth et al., extending them to a near-homogeneous regime, whenever $\bar{\zeta}^2$ is bounded and less than $1/R$. This is the first analysis that shows any benefit for Local SGD over Minibatch SGD in any non-homogeneous regime. At the other extreme, with high heterogeneity $\zeta_* > HB$, we show that Accelerated Mini-Batch SGD is optimal. But this leaves open the regime of medium heterogeneity, of $HB/\sqrt{R} \ll \zeta_* \ll HB$, where it might be possible to improve over (Accelerated) Minibatch SGD, but *not* using Local SGD—can other methods be devised for this regime?

## Broader impact

Distributed optimization is an important problem with significant implications for the success and feasibility of large scale machine learning. In recent years, the optimization community has expended significant efforts into trying to analyze and understand a seemingly simple and appealing algorithm—Local SGD. The recent work of Woodworth et al. [23] shows that much of these efforts were in vain, resulting in guarantees that do not show improvement over the familiar Minibatch SGD. However, their analysis was limited to the homogeneous case. Understanding distributed learning for heterogeneous data is important as the setting arises quite naturally, for example, when different servers or devices have data with different characteristics. Our analysis can provide clarity for the field by highlighting (i) the important regimes and important questions to focus on, (ii) what types of guarantees can help progress, and (iii) what regimes require new algorithms to be developed. Such improved focus can significantly improve and accelerate development, which in turn can have significant practical implications wherever distributed learning is used (which is increasingly everywhere).

As with many other distributed optimization papers, we focus here on reaching consensus between the different machines. However, in many, if not most, heterogeneous data settings, this could be undesirable, since insisting on a single solution (e.g. single predictor or single model) for different distributions may be sub-optimal, and may also perform very poorly on outlier distributions, which could disproportionately affect atypical users, minority groups or groups for which less training data is available. Therefore, it may sometimes be preferable to take a different, pluralistic approach to distributed learning, where different predictors are learned for each individual distribution, but where commonalities are leveraged to improve performance. This can be more fair and/or more efficient. Although in this paper we focus on achieving consensus, in part so as to align it with existing literature, the ideas developed here can also be applicable in studying pluralistic approaches.

## Acknowledgments and Disclosure of Funding

This work is partially supported by NSF/BSF award 1718970, NSF-DMS 1547396, and a Google Faculty Research Award. BW is supported by a Google PhD Fellowship.

## Footnotes

[1]Variable-length rounds of communication are also possible, but do not substantially change the picture.

[2] A similar calculation establishes the variance at $x^*$ is $\sigma_*^2 / MK$.

[3]This analysis can likely also be stated in terms of $\sigma_*$, but this does not easily follow from existing work.

[4]Karimireddy et al. [11] also analyzed a variant of heterogeneous Local SGD (refereed to as FEDAVG), that, as we discuss in Appendix B, ends up essentially equivalent to Minibatch SGD. The guarantee is thus not on the performance of Local SGD as in (3), but rather a loose upper bound for Minibatch SGD, and so we do not include it in the Tables. Karimireddy et al. consider the more general framework where only a random subset of the machines are used in each round—in the Tables here we present the analysis as it applies to our setting where all machines are used in each round. See Appendix B for more details.

[5]As stated by Karimireddy et al. [11, Theorem II], the lower bound is for their FEDAVG method, which is the same as (13) in Section 6. However, their lower bound should be qualified, since with an optimal choice of stepsize parameters the $\zeta_*$-dependence can be avoided and the lower bound does not hold (see Section 6 and Appendix B). The more accurate statement is that their lower bound is for "traditional" Local SGD, i.e. when $\eta_{\text{inner}} = \eta_{\text{outer}}$ in the notation of Section 6, or $\eta_g = 1$ in Karimireddy et al.'s notation.

[6]Karimireddy et al. [11] establish a guarantee for SCAFFOLD that improves over Minibatch SGD in a setting where only a random subset of the machines are available in each iteration and $\zeta_*$ is sufficiently small. Here we refer to the distributed optimization setting of Section 2, where all machines are used in each iteration.

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
