[Supplementary Material 1 · full-paper.pdf]

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

| Method/Analysis | Worst-Case Error (i.e. $\mathbb{E}F(\hat{x}) - F^* \lesssim$) |
|---|---|
| Minibatch SGD<br>Theorem 1 | $\frac{H\Delta}{\lambda}\exp\left(\frac{-\lambda R}{H}\right) + \frac{\sigma_*^2}{\lambda MKR}$ |
| Accelerated Minibatch SGD<br>Theorem 1 | $\Delta\exp\left(\frac{-\sqrt{\lambda}R}{\sqrt{H}}\right) + \frac{\sigma^2}{\lambda MKR}$ |
| Local SGD<br>Koloskova et al. [12] | $\frac{\sigma_*^2}{\lambda MKR} + \frac{H\zeta_*^2}{\lambda^2 R^2} + \frac{H\sigma_*^2}{\lambda^2 KR^2} \quad$ for $R \geq \tilde{\Omega}\left(\frac{H}{\lambda}\log\Delta\right)$ |
| SCAFFOLD<br>Karimireddy et al. [11] | $\left(H\Delta + \frac{\lambda\zeta_*^2}{H^2}\right)\exp\left(\frac{-\lambda R}{H}\right) + \frac{\sigma^2}{\lambda MKR}$ |
| Local SGD<br>Theorem 3 | $\frac{HB^2}{HKR+\lambda K^2 R^2} + \frac{\sigma_* B}{\sqrt{MKR}} + \frac{H\bar{\zeta}^2}{\lambda^2 R^2} + \frac{H\sigma^2}{\lambda^2 KR^2}$ |
| Local SGD Lower Bound<br>Theorem 2 | $\min\left\{\Delta\exp\left(\frac{-\lambda R}{H}\right), \frac{H\zeta_*^2}{\lambda^2 R^2}\right\} + \frac{\sigma^2}{\lambda MKR} + \min\left\{\Delta, \frac{H\sigma^2}{\lambda^2 K^2 R^2}\right\}$ |
| Algorithm-Independent<br>Lower Bound Theorem 2 | $\min\left\{\frac{\Delta\sqrt{\lambda}}{\sqrt{H}}, \frac{\lambda\zeta_*^2}{H^2}\right\}\exp\left(-\frac{\sqrt{\lambda}R}{\sqrt{H}}\right) + \frac{\sigma^2}{\lambda MKR}$ |

Table 2: Guarantees under the strongly convex assumptions, with log factors omitted. See (12) for a definition and discussion of $\bar{\zeta}$.

## A Discussion of Local SGD lower bound in strongly convex setting

In the strongly convex case, the lower bound from Theorem 2 nearly matches the upper bound of Koloskova et al.. Focusing on the case $H = B = \sigma = 1$ in order to emphasize the role of $\zeta_*$, the only differences are between a term $1/(KR^2)$ versus $1/(K^2 R^2)$—a gap which also exists in the homogeneous case [23]—and between $\exp(-\lambda R) + \zeta_*^2/(\lambda^2 R^2)$ and $\min\{\exp(-\lambda R), \zeta_*^2/(\lambda^2 R^2)\}$. The latter gap is somewhat more substantial than the convex case, but nevertheless indicates that the $\zeta_*^2/(\lambda^2 R^2)$ rate cannot be improved until the number of rounds of communication is at least the condition number (or $\zeta_*^2$ is very large).

## B Discussion of Karimireddy et al. [11]

We compare our results to Karimireddy et al. [11], who presented an analysis of the inner/outer stepsize variant of Section 6 (as FEDAVG, with a different stepsize parametrization—see below) as well as the novel method SCAFFOLD which incorporates variance reduction.

Karimireddy et al. [11, Theorem V] show that for the inner/outer stepsize updates (13), with optimal choice of stepsizes, in the weakly convex case

$$\mathbb{E}F(\hat{x}) - F^* \leq O\left(\frac{HB^2}{R} + \frac{\sigma B}{\sqrt{SKR}} + \frac{(H\zeta_*^2 B^4)^{1/3}}{R^{2/3}} + \sqrt{1 - \frac{S}{M}} \cdot \frac{\zeta_* B}{\sqrt{SR}}\right) \tag{16}$$

and in the strongly convex case (for $R \geq \Omega\left(\frac{H}{\lambda}\right)$)

$$\mathbb{E}F(\hat{x}) - F^* \leq O\left(\lambda B^2 \exp\left(\frac{-\lambda R}{H}\right) + \frac{\sigma^2}{\lambda SKR} + \frac{H\zeta_*^2}{\lambda^2 R^2} + \left(1 - \frac{S}{M}\right) \cdot \frac{\zeta_*^2}{\lambda SR}\right). \tag{17}$$

But these are loose upper bounds: as discussed in Section 6, the Minibatch SGD guarantees also apply to the inner/outer variant (by using $\eta_{\text{inner}} = 0$). The Minibatch SGD guarantees (14) and (15) can therefor be viewed also as guarantees on the inner/outer variant (i.e. Karimireddy et al.'s FEDAVG) that improve over (16) and (17) in several ways: (a) they avoid the the third terms in (16) and (17); (b) they improve the fourth terms by a factor of $\sqrt{S}$ and $S$ respectively; and (c) they avoid the requirement $R > H/\lambda$.

Karimireddy et al.'s presentation actually uses a different step-size parametrization that does not allow for $\eta_{\text{inner}} = 0$: they use $\eta_l = \eta_{\text{inner}}$ and $\eta_g = \eta_{\text{outer}}/\eta_{\text{inner}}$. We prefer the presentation using

$\eta_{\text{inner}}$ and $\eta_{\text{outer}}$ in order to emphasize the relationship with Minibatch SGD and in order to explicitly allow $\eta_{\text{inner}} = 0$. But in any case, even using their parametrization, $\eta_l = \eta_{\text{inner}}$ could be taken arbitrarily close to zero making the deviation from Minibatch SGD arbitrarily small. Indeed, the Karimireddy et al.'s bounds (16) and (17) are obtained when $\eta_{\text{inner}}$ is already so small that the algorithm is essentially equivalent to Minibatch SGD.

But Karimireddy et al.'s main contribution was the presentation of SCAFFOLD, which incorporates machine-specific control iterates that reduce the inter-machine variances. For SCAFFOLD, [11, Theorem VII] show that in the weakly convex case[7]:

$$\mathbb{E}F(\hat{x}) - F^* \leq O\left(\frac{HB^2}{R} + \frac{\sigma B}{\sqrt{SKR}} + \frac{M\zeta_*^2}{HSR} + \frac{\sigma\zeta_*\sqrt{M}}{HS\sqrt{KR}}\right), \tag{18}$$

and in the strongly convex case (for $R \geq \max\{\frac{H}{\lambda}, \frac{M}{S}\}$)

$$\mathbb{E}F(\hat{x}) - F^* \leq O\left(\lambda\left(B^2 + \frac{M\zeta_*^2}{SH^2}\right)\exp\left(-\min\left\{\frac{\lambda}{H}, \frac{S}{M}\right\}R\right) + \frac{\sigma^2}{\lambda SKR}\right). \tag{19}$$

These guarantees are also obtained when $\eta_{\text{inner}}$ is so close to zero that this is essentially a minibatch method, in this case "Minibatch SAGA" [cf. 7].

Although SCAFFOLD is aimed specifically at the setting where a subset of machines are used in each round (i.e. $S < M$), let us first check whether it provides benefits in our "standard" setting (introduced in Sections 2), where all machines are used each round (i.e. $S = M$). In this case, the SCAFFOLD bounds (18) and (19) may improve over the loose upper bounds (16) and (17), but this is only due to the looseness in these upper bounds. The SCAFFOLD upper bounds (for $S = M$) do not actually improve over the Minibatch SGD guarantees of Theorem 1, and only include additional terms (see Tables 1 and 2). That is, the SCAFFOLD upper bounds, when $S = M$, are valid also for FEDAVG (since as discussed above, Minibatch SGD gurantees are valid also for FEDAVG) and so do not show a benefit in the setting where all machines are used each round. This is perhaps not surprising, since in this setting there is no need to reduce inter-machine variance, and so no benefit from variance reduction.

Let us turn then to the setting of Section 7, where only a random subset of $S < M$ machines are used in each iteration, and for which SCAFFOLD was developed. In this setting, SCAFFOLD does show a benefit in some regimes. Let us focus on the weakly convex case and compare the SCAFFOLD guarantee (18) with the Minibatch SGD guarantee (14). We can verify that, e.g. when $\sigma = 0$ and $\zeta_* = HB$, SCAFFOLD improves over Minibatch SGD if $\frac{M}{R} \ll \frac{S}{M} \ll \frac{R}{M}$, and $S < M$, but the SCAFFOLD guarantee (18) is worse then Minibatch SGD if $\frac{S}{M} \ll \frac{M}{R}$. More generally, the SCAFFOLD guarantee is worse than Minibatch SGD if $\sigma = 0$ and $\frac{S}{M} \ll \frac{\zeta_*^2}{H^2B^2}\min(1, \frac{M}{R})$. Also for strongly convex objectives, SCAFFOLD improves over Minibatch SGD in some regimes but the guarantee (19) is worse than Minibatch SGD in other regimes. Care is required to map out the precise regimes and how they depend on the various problem parameters.

## C    Proof of Theorem 1

In this Appendix, we prove Theorem 1, starting with an analysis of Minibatch SGD, and proceeding to analyze Accelerated Minibatch SGD.

### C.1    Minibatch SGD for heterogeneous objectives

For the proof, we will use the following standard property of convex functions:

**Lemma 1** (Co-Coercivity of the Gradient). *For any $H$-smooth and convex $F$, and any $x$, and $y$,*

$$\|\nabla F(x) - \nabla F(y)\|^2 \leq H\langle\nabla F(x) - \nabla F(y),\, x - y\rangle,$$

*and*

$$\|\nabla F(x) - \nabla F(y)\|^2 \leq 2H(F(x) - F(y) - \langle\nabla F(y),\, x - y\rangle).$$

Also note the following result from Stich [21], which is useful for optimizing the step-sizes.

**Lemma 2** (Stich [21], Lemma 3). *Consider non-negative sequences* $\{r_t\}_{t \geq 0}$ *and* $\{s_t\}_{t \geq 0}$, *which satisfy:*

$$r_{t+1} \leq (1 - a\eta_t)r_t - b\eta_t s_t + c\eta_t^2, \tag{20}$$

*for non-negative step-sizes* $\eta_t \leq \frac{1}{d}, \forall t$, *for a parameter* $d \geq a$, $d > 0$. *Then there exist a choice of step-sizes* $\eta_t$ *and averaging weights* $w_t$, *such that:*

$$\frac{b}{W_T} \sum_{t=0}^{T} sw_t + ar_{T+1} \leq 32dr_0 \exp\left[-\frac{aT}{2d}\right] + \frac{36c}{aT}, \tag{21}$$

*for* $W_T := \sum_{t=0}^{T} w_t$.

Finally we can prove the following result for Minibatch SGD in this setting.

**Theorem 5.** *Under the convex assumptions, the average of the iterates of Minibatch SGD guarantees for a universal constant* $c$,

$$\mathbb{E}F(\hat{x}) - F^* \leq c \cdot \frac{HB^2}{R} + c \cdot \frac{\sigma_* B}{\sqrt{MKR}}.$$

*Under the strongly convex assumptions, a weighted average of its iterates guarantees*

$$\mathbb{E}F(\hat{x}) - F^* \leq c \cdot \frac{H\Delta}{\lambda} \exp\left[-\frac{\lambda R}{8H}\right] + c \cdot \frac{\sigma_*^2}{\lambda MKR}.$$

*Proof.* By the $\lambda$-strong convexity of $F$, where $\lambda$ might be equal to zero:

$$\mathbb{E}\|x_{t+1} - x^*\|^2 = \mathbb{E}\left\|x_t - \eta_t \frac{1}{KM} \sum_{m=1}^{M} \sum_{k=1}^{K} \nabla f(x_t; z_t^{m,k}) - x^*\right\|^2, \tag{22}$$

$$= \mathbb{E}\|x_t - x^*\|^2 - 2\eta_t \mathbb{E}\langle \nabla F(x_t), x_t - x^* \rangle$$

$$+ \eta_t^2 \mathbb{E}\left\|\frac{1}{KM} \sum_{m=1}^{M} \sum_{k=1}^{K} \nabla f(x_t; z_t^{m,k})\right\|^2, \tag{23}$$

$$\leq (1 - \lambda\eta_t)\mathbb{E}\|x_t - x^*\|^2 - 2\eta_t \mathbb{E}[F(x_t) - F^*]$$

$$+ \eta_t^2 \mathbb{E}\left\|\frac{1}{KM} \sum_{m=1}^{M} \sum_{k=1}^{K} \nabla f(x_t; z_t^{m,k})\right\|^2. \tag{24}$$

By the $H$-smoothness of $f(\cdot\,; z)$, we can bound the final term with

$$\mathbb{E}\left\|\frac{1}{KM}\sum_{m=1}^{M}\sum_{k=1}^{K}\nabla f(x_t; z_t^{m,k})\right\|^2$$

$$= \mathbb{E}\left\|\frac{1}{KM}\sum_{m=1}^{M}\sum_{k=1}^{K}\left[\nabla f(x_t; z_t^{m,k}) - \nabla f(x^*; z_t^{m,k}) + \nabla f(x^*; z_t^{m,k})\right]\right\|^2, \tag{25}$$

$$\leq 2\mathbb{E}\left\|\frac{1}{KM}\sum_{m=1}^{M}\sum_{k=1}^{K}\left[\nabla f(x_t; z_t^{m,k}) - \nabla f(x^*; z_t^{m,k})\right]\right\|^2$$

$$+ 2\mathbb{E}\left\|\frac{1}{KM}\sum_{m=1}^{M}\sum_{k=1}^{K}\nabla f(x^*; z_t^{m,k})\right\|^2, \tag{26}$$

$$\leq \frac{2}{KM}\sum_{m=1}^{M}\sum_{k=1}^{K}\mathbb{E}\left\|\nabla f(x_t; z_t^{m,k}) - \nabla f(x^*; z_t^{m,k})\right\|^2$$

$$+ 2\mathbb{E}\left\|\frac{1}{KM}\sum_{m=1}^{M}\sum_{k=1}^{K}\nabla f(x^*; z_t^{m,k}) - \nabla F_m(x^*)\right\|^2, \tag{27}$$

$$\leq \frac{4H}{KM}\sum_{m=1}^{M}\sum_{k=1}^{K}\mathbb{E}\left[f(x_t; z_t^{m,k}) - f(x^*; z_t^{m,k}) - \left\langle\nabla f(x^*; z_t^{m,k}),\, x_t - x^*\right\rangle\right] + \frac{2\sigma_*^2}{MK}, \tag{28}$$

$$= 4H\mathbb{E}[F(x_t) - F^*] + \frac{2\sigma_*^2}{MK}. \tag{29}$$

Where, for the third inequality we used Lemma 1. Plugging this back into (24), then for $\eta_t \leq \frac{1}{4H}$,

$$\mathbb{E}\|x_{t+1} - x^*\|^2 \leq (1 - \lambda\eta_t)\mathbb{E}\|x_t - x^*\|^2 - 2\eta_t(1 - 2H\eta_t)\mathbb{E}[F(x_t) - F^*] + \frac{2\eta_t^2\sigma_*^2}{MK}, \tag{30}$$

$$\leq (1 - \lambda\eta_t)\mathbb{E}\|x_t - x^*\|^2 - \eta_t\mathbb{E}[F(x_t) - F^*] + \frac{2\eta_t^2\sigma_*^2}{MK}, \tag{31}$$

$$\mathbb{E}[F(x_t) - F^*] \leq \left(\frac{1}{\eta_t} - \lambda\right)\mathbb{E}\|x_t - x^*\|^2 - \frac{1}{\eta_t}\mathbb{E}\|x_{t+1} - x^*\|^2 + \frac{2\eta_t\sigma_*^2}{MK}. \tag{32}$$

Now we look at $\lambda = 0$ and $\lambda > 0$ separately.

**Convex case ($\lambda = 0$):**   Choose a constant step-size,

$$\eta_t = \eta = \min\left\{\frac{1}{4H},\, \frac{B\sqrt{MK}}{\sigma_*\sqrt{T}}\right\}. \tag{33}$$

Then the averaged iterate $\bar{x}_R = \frac{1}{R}\sum_{t=1}^{R} x_t$ satisfies:

$$\mathbb{E}F(\bar{x}_R) - F^* \leq \frac{1}{R}\sum_{t=1}^{R}\mathbb{E}F(x_t) - F^*, \tag{34}$$

$$\leq \frac{1}{R}\sum_{t=1}^{R}\frac{1}{\eta}\mathbb{E}\|x_t - x^*\|^2 - \frac{1}{\eta}\mathbb{E}\|x_{t+1} - x^*\|^2 + \frac{2\eta\sigma_*^2}{MK}, \tag{35}$$

$$\leq \frac{\|x_0 - x^*\|^2}{\eta R} + \frac{2\eta\sigma_*^2}{MK}, \tag{36}$$

$$\leq \max\left\{\frac{4HB^2}{R},\, \frac{\sigma_* B}{\sqrt{MKR}}\right\} + \frac{2\sigma_* B}{\sqrt{MKR}}, \tag{37}$$

$$\leq \frac{4HB^2}{R} + \frac{3\sigma_* B}{\sqrt{MKR}}. \tag{38}$$

**Strongly convex case ($\lambda > 0$):**   Rewriting (31),

$$\mathbb{E}\|x_{t+1} - x^*\|^2 \leq (1 - \lambda\eta_t)\mathbb{E}\|x_t - x^*\|^2 - \eta_t\mathbb{E}[F(x_t) - F^*] + \frac{2\eta_t^2\sigma_*^2}{MK}, \tag{39}$$

we note that it satisfies the conditions for Lemma 2 for the specific assignment:

$$r_t = \mathbb{E}\|x_t - x^*\|^2, \ s_t = \mathbb{E}[F(x_t) - F^*], \tag{40}$$

$$a = \lambda, \ b = 1, \ c = \frac{2\sigma_*^2}{MK}, \ d = 4H, \ T = R. \tag{41}$$

Thus using Lemma 2, and applying Jensen's inequality we can guarantee the following convergence rate for the averaged iterate $\hat{x}_R = \frac{1}{W_R}\sum_{t=1}^{R} w_t x_t$,

$$\mathbb{E}[F(\hat{x}_R) - F^*] \leq 128H\|x_0 - x^*\|^2 \exp\left[-\frac{\lambda R}{8H}\right] + \frac{72\sigma_*^2}{\lambda MKR}, \tag{42}$$

using step-size $\eta_t$ and $w_t$ given by,

$$
\begin{array}{lll}
if \ R \leq \dfrac{4H}{\lambda}, & \eta_t = \dfrac{1}{4H}, & w_t = (1 - \lambda\eta_t)^{-(t+1)}, \\[2mm]
if \ R > \dfrac{4H}{\lambda} \ and \ t < t_0, & \eta_t = \dfrac{1}{4H}, & w_t = 0, \\[2mm]
if \ R > \dfrac{4H}{\lambda} \ and \ t \geq t_0, & \eta_t = \dfrac{2}{\lambda(\kappa + t - t_0)}, & w_t = (\kappa + t - t_0)^2,
\end{array}
$$

where $\kappa = \frac{8H}{\lambda}$ and $t_0 = \lceil\frac{R}{2}\rceil$. We conclude by observing that $HB^2 \leq \frac{H\Delta}{\lambda}$.  □

## C.2   Accelerated Minibatch SGD for heterogeneous objectives

We first recall some classical results from Ghadimi and Lan [8, 9] for accelerated variants of minibatch SGD. These results are for minimizing $F(x) := \mathbb{E}_{z\sim\mathcal{D}}f(x, z)$ where $F$ is $H$-smooth and $\lambda(\geq 0)$-strongly convex. The algorithms use unbiased stochastic gradients $\{g_t\}_{t\in[T]}$, i.e., for all $t$, $\mathbb{E}[g_t(x)] = \nabla F(x)$ which have bounded variance for all $x$, i.e., $\mathbb{E}\|g_t(x) - \nabla F(x)\|^2 \leq \sigma^2$.

First consider the AC-SA algorithm Ghadimi and Lan [c.f., Sec 3.1, 8]), with step-size parameters $\{\alpha_t\}_{t\geq 1}$ and $\{\gamma_t\}_{t\geq 1}$ s.t. $\alpha_1 = 1$, $\alpha_t \in (0, 1)$ for any $t \geq 2$ and $\gamma_t > 0$ for any $t \geq 1$. The algorithm maintains three intertwined sequences $\{x_t\}$, $\{x_t^{ag}\}$, and $\{x_t^{md}\}$, updated as follows:

1. Set the initial points $x_0^{ag} = x_0 \in X$ and $t = 1$;

2. Set $x_t^{md} = \frac{(1-\alpha_t)(\lambda+\gamma_t)}{\gamma_t+(1-\alpha_t^2)\lambda}x_{t-1}^{ag} + \frac{\alpha_t[(1-\alpha_t)\mu+\gamma_t]}{\gamma_t+(1-\alpha_t^2)\lambda}x_{t-1}$;

3. Call the stochastic oracle to get the gradient $g_t$ at the point $x_t^{md}$;

4. Set $x_t = \arg\min_{x\in X}\left\{\alpha_t[\langle g_t, x\rangle + \frac{\lambda}{2}\|x_t^{md} - x\|^2] + [(1-\alpha_t)\frac{\lambda}{2} + \frac{\gamma_t}{2}]\|x_{t-1} - x\|^2\right\}$;

5. Set $x_t^{ag} = \alpha_t x_t + (1 - \alpha_t)x_{t-1}^{ag}$;

6. Set $t \leftarrow t + 1$ and go to step 1.

We have the following (almost optimal) convergence rate for strongly convex functions using AC-SA (see, Sec 3.1 in [8]).

**Lemma 3.** *(Ghadimi and Lan [8], Proposition 9) Let $\hat{x}^{ag}$ be computed by $T$ steps of AC-SA using stochastic gradients of variance $\sigma^2$, then for a universal constant c,*

$$\mathbb{E}[F(x^{ag}) - F(x^\star)] \leq c \cdot \frac{H\|x_0 - x^\star\|^2}{T^2} + c \cdot \frac{\sigma^2}{\lambda T}.$$

It can be adapted to the weakly convex case by noting that, if $\tilde{F}(x) := F(x) + \frac{\lambda}{2}\|x_0 - x\|^2$ for any $\lambda, x_0$, and $x^\star = \arg\min F(x)$ then,

$$\min_y \mathbb{E}\left[\tilde{F}(y)\right] \leq \mathbb{E}\left[F(x^\star) + \frac{\lambda}{2}\|x_0 - x^\star\|^2\right],$$

$$\Rightarrow -\mathbb{E}\left[F(x^\star)\right] \leq -\mathbb{E}\left[\min_y \tilde{F}(y)\right] + \frac{\lambda}{2}\|x_0 - x^\star\|^2,$$

$$\Rightarrow \mathbb{E}\left[F(x) - F(x^\star)\right] \leq \mathbb{E}\left[\tilde{F}(x) - \min_y \tilde{F}(y)\right] + \frac{\lambda}{2}\|x_0 - x^\star\|^2, \forall x.$$

This also holds if we optimize the right hand side w.r.t. $\lambda$. In other words, a guarantee for the strongly convex case, can be converted to the weakly convex case, by regularizing with $\frac{\lambda}{2}\|x_0 - x^\star\|^2$ with optimal value of $\lambda$. This gives the following result,

**Lemma 4.** *Let $\hat{x}^{ag}$ be computed by $T$ steps of AC-SA on the regularized objective $\tilde{F}(x) = F(x) + \frac{\sigma}{2\|x_0 - x^*\|\sqrt{T}}\|x - x_0\|^2$, where the stochastic gradients have variance $\sigma^2$, then for a constant $c$*

$$\mathbb{E}\left[F(\hat{x}^{ag}) - F(x^\star)\right] \leq c \cdot \frac{H\|x_0 - x^\star\|^2}{T^2} + c \cdot \frac{\sigma\|x_0 - x^\star\|}{\sqrt{T}}.$$

This is minimax optimal for weakly convex functions. Next we consider the multi-stage accelerated SGD algorithm Ghadimi and Lan [c.f., Sec 3, 9] which uses the above AC-SA algorithm. Let $p_0 \in X$, have bounded sub-optimality $F(p_0) - F(x^\star) \leq \Delta$, then for $k = 1, 2, \ldots$,

1. Run $N_k$ iterations of the generic AC-SA by using $x_0 = p_{k-1}$, $\{\alpha_t\}_{t\geq 1}$, and $\{\gamma_t\}_{t\geq 1}$, with relevant definitions as follows,

$$N_k = \left\lceil \max\left\{4\sqrt{\frac{2H}{\lambda}}, \frac{128\sigma^2}{3\lambda\Delta 2^{-(k+1)}}\right\}\right\rceil,$$

$$\alpha_t = \frac{2}{t+1}, \gamma_t = \frac{4\phi_k}{t(t+1)},$$

$$\phi_k = \max\left\{2H, \left[\frac{\lambda\sigma^2}{3\Delta 2^{-(k-1)}N_k(N_k+1)(N_k+2)}\right]^{1/2}\right\};$$

2. Set $p_k = x_{N_k}^{ag}$ where $x_{N_k}^{ag}$ is the solution obtained in the previous step.

We have following optimal rate for strongly convex functions for this algorithm,

**Lemma 5.** *(Ghadimi and Lan [9], Proposition 7) Let $\hat{x}^{ag}$ be computed by $T$ steps of multi-stage AC-SA using stochastic gradients of variance $\sigma^2$, then for a universal constant $c$,*

$$\mathbb{E}\left[F(\hat{x}^{ag}) - F(x^\star)\right] \leq c \cdot \Delta \exp\left[-\sqrt{\frac{\lambda}{H}}T\right] + c \cdot \frac{\sigma^2}{\lambda T}.$$

**Theorem 6.** *Under the convex assumptions, performing AC-SA on the regularized objective $\tilde{F}(x) = F(x) + \frac{\sigma}{2B\sqrt{MKR}}\|x\|^2$ guarantees for a universal constant $c$*

$$\mathbb{E}F(\hat{x}) - F^* \leq c \cdot \frac{HB^2}{R^2} + c \cdot \frac{\sigma B}{\sqrt{MKR}}.$$

*Under the strongly convex assumptions, the multi-stage AC-SA algorithm guarantees*

$$\mathbb{E}F(\hat{x}) - F^* \leq c \cdot \Delta \exp\left[-\sqrt{\frac{\lambda}{H}}R\right] + c \cdot \frac{\sigma^2}{\lambda MKR}.$$

*Proof.* In order to use the previous lemmas, first note that the stochastic gradient at time $t$ at point $x$ is given by $\frac{1}{MK}\sum_{m=1}^M \sum_{k=1}^K \nabla f(x; z_{m,k}^t)$, where $z_{m,k}^t \sim^{i.i.d.} \mathcal{D}^m$ for all machines. Fortunately,

its still an unbiased gradient estimate, i.e., $\mathbb{E}\left[\frac{1}{MK}\sum_{m=1}^{M}\sum_{k=1}^{K}\nabla f(x; z_{m,k}^t)\right] = \nabla F(x)$ since the iterates on each machine are sampled i.i.d. Its variance is given by,

$$\mathbb{E}\left\|\frac{1}{MK}\sum_{m=1}^{M}\sum_{k=1}^{K}\nabla f(x; z_{m,k}^t) - \nabla F(x)\right\|^2 \tag{43}$$

$$= \frac{1}{M^2 K^2}\sum_{m=1}^{M}\sum_{k=1}^{K}\mathbb{E}\left\|\nabla f(x; z_{m,k}^t) - \nabla F_m(x)\right\|^2 \tag{44}$$

$$\leq \frac{1}{M^2 K^2}\sum_{m=1}^{M}\sum_{k=1}^{K}\sigma^2 \tag{45}$$

$$= \frac{\sigma^2}{MK} \tag{46}$$

Plugging this into Lemmas 4 and 5 completes the proof. $\qquad\square$

## D  Proof of Theorem 2

Consider the following function $F : \mathbb{R}^4 \to \mathbb{R}$:

$$F(x) = \frac{1}{2}(F_1(x) + F_2(x)) = \frac{1}{2}\left(\mathbb{E}_{z^1 \sim \mathcal{D}^1} f(x; z^1) + \mathbb{E}_{z^2 \sim \mathcal{D}^2} f(x; z^2)\right) \tag{47}$$

The distribution $z^1 \sim \mathcal{D}^1$ is described by $z^1 = (1, z)$ for $z \sim \mathcal{N}(0, \sigma^2)$. Similarly, $z^2 \sim \mathcal{D}^2$ is specified by $z^2 = (2, z)$ for $z \sim \mathcal{N}(0, \sigma^2)$. The lower bound construction will be based on just two functions. For $M > 2$ machines, we simply assign the first $\lfloor M/2 \rfloor$ machines $F_1$ and the next $\lfloor M/2 \rfloor$ machines $F_2$. This diminishes the lower bound by at most a $(M-1)/M$ factor. Therefore, we continue with the case $M = 2$.

Following Woodworth et al. [23], we define the local functions $F_1$ and $F_2$ via the auxiliary function

$$g(x_1, x_2, x_3, z) = \frac{\mu}{2}(x_1 - c)^2 + \frac{H}{2}\left(x_2 - \frac{\sqrt{\mu c}}{\sqrt{H}}\right)^2 + \frac{H}{8}\left(x_3^2 + [x_3]_+^2\right) + z^\top x_3 \tag{48}$$

$$G(x_1, x_2, x_3) = \mathbb{E}_z g(x_1, x_2, x_3, z) \tag{49}$$

where $c > 0$ and $\mu \in \left[\lambda, \frac{H}{16}\right]$ are parameters to be determined later, and where $[x]_+ := \max\{x, 0\}$. Then, we define

$$f(x; (1, z)) = g(x_1, x_2, x_3, z) + \frac{Lx_4^2}{2} + \zeta_* x_4 \tag{50}$$

$$f(x; (2, z)) = g(x_1, x_2, x_3, z) + \frac{\lambda x_4^2}{2} - \zeta_* x_4 \tag{51}$$

for a parameter $L \in [\lambda, H]$ to be determined later. Therefore,

$$F_1(x) = \mathbb{E}_{z^1 \sim \mathcal{D}^1} f(x; z^1) = G(x_1, x_2, x_3) + \frac{Lx_4^2}{2} + \zeta_* x_4 \tag{52}$$

$$F_2(x) = \mathbb{E}_{z^2 \sim \mathcal{D}^2} f(x; z^2) = G(x_1, x_2, x_3) + \frac{\lambda x_4^2}{2} - \zeta_* x_4 \tag{53}$$

It is clear from inspection that both $F_1$ and $F_2$, and consequently $F$, are $H$-smooth and $\lambda$-strongly convex. Furthermore, the variance of the gradients is bounded by $\sigma^2$ for both $\mathcal{D}^1$ and $\mathcal{D}^2$.

It is clear that $G$ attains its minimum of zero at $\left[c, \frac{\sqrt{\mu c}}{\sqrt{H}}, 0\right]$ so $\nabla G\left(c, \frac{\sqrt{\mu c}}{\sqrt{H}}, 0\right) = 0$, and thus

$$\nabla F\left(c, \frac{\sqrt{\mu c}}{\sqrt{H}}, 0, 0\right) = \nabla G\left(c, \frac{\sqrt{\mu c}}{\sqrt{H}}, 0\right) + \left(\frac{\zeta_*}{2} - \frac{\zeta_*}{2}\right)e_4 = 0 \tag{54}$$

From now on, we use $x^* = \left[c, \frac{\sqrt{\mu c}}{\sqrt{H}}, 0, 0\right]$ to denote the minimizer of $F$, which has norm

$$\|x^*\|^2 = \left(1 + \frac{\mu}{H}\right)c^2 \leq 2c^2 \tag{55}$$

We can therefore ensure $\|x^*\|^2 \leq B^2$ by choosing $c^2 \leq \frac{B^2}{2}$. Furthermore, the initial suboptimality

$$F(0,0,0,0) - F^* = \mu c^2 \tag{56}$$

Therefore, we can ensure $F(0,0,0,0) - F^* \leq \Delta$ by choosing $c^2 \leq \frac{\Delta}{\mu}$. We conclude by showing that for this objective, $\zeta_*^2$ bounded by

$$\frac{1}{2} \sum_{m=1}^{2} \|\nabla F_m(x^*)\|^2 = \|\nabla F_2(x^*)\|^2 = \|\nabla F_1(x^*)\|^2 = \zeta_*^2 \tag{57}$$

Therefore, this objective has the desired level of heterogeneity.

Therefore, we have shown that the objective satisfies all of the necessary conditions for the lower bound. All that remains is to lower bound the error of Local SGD with a constant stepsize $\eta$ applied to this function.

**Lemma 6.** *For $\mu \leq 2L$, then Local SGD with any constant stepsize $\eta \leq \frac{1}{L}$ applied to $F_1$ and $F_2$ after being initialized at zero results in $\hat{x}_4$ such that*

$$\frac{(L+\mu)\hat{x}_4^2}{4} \geq \frac{\zeta_*^2(L+\mu)}{16\mu^2}\left(\frac{L-\mu}{L} - (1-\mu\eta)^K\right)^2 \mathbb{1}_{\left\{\eta \leq \frac{1}{L}\right\}} \mathbb{1}_{\left\{(1-\mu\eta)^K \leq \frac{L-\mu}{L}\right\}}$$

*Proof.* Since the coordinates of $F_1$ and $F_2$ are completely decoupled, the behavior of the fourth coordinate of the iterates can be analyzed separately from the others.

Let $x_{k,r}^{(1)}$ denote the fourth coordinate of machine 1's iterate at the $k$th iteration of round $r$, and similarly for $x_{k,r}^{(2)}$. The local SGD dynamics give

$$x_{k+1,r}^{(1)} = x_{k,r}^{(1)} - \eta\left(Lx_{k,r}^{(1)} + \zeta_*\right) = -\frac{\zeta_*}{L} + (1-L\eta)\left(x_{k,r}^{(1)} + \frac{\zeta_*}{L}\right) \tag{58}$$

$$x_{k,r}^{(2)} = x_{k,r}^{(2)} - \eta\left(-\zeta_* + \mu x_{k,r}^{(2)}\right) = \frac{\zeta_*}{\mu} + (1-\mu\eta)\left(x_{k,r}^{(2)} - \frac{\zeta_*}{\mu}\right) \tag{59}$$

and $\hat{x}_4 = \frac{1}{2}\left(x_{K,R}^{(1)} + x_{K,R}^{(2)}\right) = x_{0,R+1}$. Unravelling this recursion, we have that

$$x_{0,r+1} = x_{0,r+1}^{(1)} = x_{0,r+1}^{(2)} = \frac{1}{2}\left(\frac{\zeta_*}{\mu} - \frac{\zeta_*}{L} + (1-\mu\eta)^K\left(x_{0,r} - \frac{\zeta_*}{\mu}\right) + (1-L\eta)^K\left(x_{0,r} + \frac{\zeta_*}{L}\right)\right) \tag{60}$$

Furthermore, if $\eta \leq \frac{1}{L}$ then $(1-L\eta) \geq 0$, so if $x_{0,r} \geq 0$ then

$$x_{0,r+1} \geq \frac{\zeta_*}{2\mu} - \frac{\zeta_*}{2L} + (1-\mu\eta)^K\left(\frac{x_{0,r}}{2} - \frac{\zeta_*}{2\mu}\right) \geq \frac{\zeta_*}{2\mu}\left(\frac{L-\mu}{L} - (1-\mu\eta)^K\right) \tag{61}$$

Finally, since $x_{0,0} = 0 \geq 0$, the condition $x_{0,r} \geq 0$ will hold throughout optimization, so

$$\hat{x}_4 \geq \frac{\zeta_*}{2\mu}\left(\frac{L-\mu}{L} - (1-\mu\eta)^K\right) \tag{62}$$

Therefore, if $\eta \leq \frac{1}{L}$ and $(1-\mu\eta)^K \leq \frac{L-\mu}{L}$ then

$$\frac{(L+\mu)\hat{x}_4^2}{4} \geq \frac{\zeta_*^2(L+\mu)}{16\mu^2}\left(\frac{L-\mu}{L} - (1-\mu\eta)^K\right)^2 \tag{63}$$

This completes the proof. □

We now prove the theorem:

**Theorem 2.** *For any $M$, $K$, and $R$ there exist objectives in four dimensions such that Local SGD initialized at zero and using any fixed stepsize $\eta$ will have suboptimality at least*

$$\mathbb{E}F(\hat{x}) - F^* \geq c \cdot \left(\min\left\{\frac{HB^2}{R}, \frac{\left(H\zeta_*^2 B^4\right)^{1/3}}{R^{2/3}}\right\} + \frac{\left(H\sigma^2 B^4\right)^{1/3}}{K^{2/3}R^{2/3}} + \frac{\sigma B}{\sqrt{MKR}}\right)$$

$$\mathbb{E}F(\hat{x}) - F^* \geq c \cdot \left(\min\left\{\Delta \exp\left(-\frac{6\lambda R}{H}\right), \frac{H\zeta_*^2}{\lambda^2 R^2}\right\} + \min\left\{\Delta, \frac{H\sigma^2}{\lambda^2 K^2 R^2}\right\} + \frac{\sigma^2}{\lambda MKR}\right)$$

*under the convex and strongly convex assumptions (for $H \geq 16\lambda$), respectively.*

*Proof.* Since the four different coordinates are completely decoupled from each other, it suffices to analyze each coordinate separately.

In the course of proving [Theorem 3 23], Woodworth et al. prove that

$$\mathbb{E}G(\hat{x}_1, \hat{x}_2, \hat{x}_3) - G\left(c, \frac{\sqrt{\mu c}}{\sqrt{H}}, 0\right)$$

$$\geq \frac{\mu c^2 (1 - \mu \eta)^{KR}}{2} + \frac{\mu c^2}{2} \mathbb{1}_{\{\eta > \frac{2}{H}\}} + \frac{H \eta^2 \sigma^2}{18432} \mathbb{1}_{\{\eta \leq \frac{2}{H}\}} \mathbb{1}_{\{\eta \geq \frac{8}{HKR}\}} \quad (64)$$

Furthermore, by Lemma 6

$$\frac{(L + \lambda)\hat{x}_4^2}{4} \geq \frac{\zeta_*^2 (L + \mu)}{16 \mu^2} \left(\frac{L - \mu}{L} - (1 - \mu \eta)^K\right)^2 \mathbb{1}_{\{\eta \leq \frac{1}{L}\}} \mathbb{1}_{\{(1 - \mu \eta)^K \leq \frac{L - \mu}{L}\}} \quad (65)$$

Therefore, choosing $L = \frac{H}{2}$

$$\mathbb{E}F(\hat{x}) - F^* = \mathbb{E}G(\hat{x}_1, \hat{x}_2, \hat{x}_3) - G\left(c, \frac{\sqrt{\mu c}}{\sqrt{H}}, 0\right) + \frac{H + 2\lambda}{8} \hat{x}_4^2 \quad (66)$$

$$\geq \frac{\mu c^2 (1 - \mu \eta)^{KR}}{2} + \frac{\mu c^2}{2} \mathbb{1}_{\{\eta > \frac{2}{H}\}} + \frac{H \eta^2 \sigma^2}{18432} \mathbb{1}_{\{\eta \leq \frac{2}{H}\}} \mathbb{1}_{\{\eta \geq \frac{8}{HKR}\}}$$

$$+ \frac{\zeta_*^2 (H + 2\mu)}{32 \mu^2} \left(\frac{H - 2\mu}{H} - (1 - \mu \eta)^K\right)^2 \mathbb{1}_{\{\eta \leq \frac{2}{H}\}} \mathbb{1}_{\{(1 - \mu \eta)^K \leq \frac{H - 2\mu}{H}\}} \quad (67)$$

**Stochastic terms**

First, we will show a lower bound in terms of $\sigma^2$ using solely the first three terms of (67). Consider three cases:

**Case 1** $\eta \geq \frac{2}{H}$: In this case, from the second term of (67) we see that

$$\mathbb{E}F(\hat{x}) - F^* \geq \frac{\mu c^2}{2} \quad (68)$$

**Case 2** $\frac{1}{2\mu KR} \leq \eta \leq \frac{2}{H}$: In this case, the third term of (67) shows

$$\mathbb{E}F(\hat{x}) - F^* \geq \frac{H \eta^2 \sigma^2}{18432} \quad (69)$$

where we recalled that $\mu \leq \frac{H}{16}$, so $\eta \geq \frac{1}{2\mu KR} \geq \frac{8}{HKR}$. This is non-decreasing in $\eta$, so for any $\eta$

$$\mathbb{E}F(\hat{x}) - F^* \geq \frac{H \sigma^2}{73728 \mu^2 K^2 R^2} \quad (70)$$

**Case 3** $\eta \leq \frac{2}{H}$ and $\eta \leq \frac{1}{2\mu KR}$: In this case, from the first term of (67),

$$\mathbb{E}F(\hat{x}) - F^* \geq \frac{\mu c^2 (1 - \mu \eta)^{KR}}{2} \geq \frac{\mu c^2 \left(1 - \frac{1}{2KR}\right)^{KR}}{2} \geq \frac{\mu c^2}{4} \quad (71)$$

**Combination:** Combining these three cases, we conclude that for any $\eta$

$$\mathbb{E}F(\hat{x}) - F^* \geq \min\left\{\frac{\mu c^2}{2}, \frac{H \sigma^2}{73728 \mu^2 K^2 R^2}, \frac{\mu c^2}{4}\right\} = \min\left\{\frac{\mu c^2}{3}, \frac{H \sigma^2}{73728 \mu^2 K^2 R^2}\right\} \quad (72)$$

This lower bound holds for any stepsize, and any $\mu \in \left[\lambda, \frac{H}{16}\right]$ and regardless of $\zeta_*$. In the strongly convex case, we recall that $F(0) - F(x^*) = \mu c^2$, therefore, we choose $\mu = \lambda$, and $c^2 = \frac{\Delta}{\lambda}$ so the lower bound reads (for a universal constant $\beta$)

$$\mathbb{E}F(\hat{x}) - F^* \geq \beta \cdot \min\left\{\Delta, \frac{H \sigma^2}{\lambda^2 K^2 R^2}\right\} \quad (73)$$

To conclude, it is well known that any first-order method which accesses at most $MKR$ stochastic gradients with variance $\sigma^2$ for a $\lambda$-strongly convex objective will suffer error at least $\beta \frac{\sigma^2}{\lambda MKR}$ in the worst case [17]. Therefore, the strongly convex lower bound is

$$\mathbb{E}F(\hat{x}) - F^* \geq \beta \cdot \min\left\{\Delta, \frac{H\sigma^2}{\lambda^2 K^2 R^2}\right\} + \beta \cdot \frac{\sigma^2}{\lambda MKR} \tag{74}$$

In the convex case, we recall that $\|x^*\|^2 \leq 2c^2$, so we choose $c^2 = \frac{B^2}{2}$, and set $\mu = \left(\frac{H\sigma^2}{B^2 K^2 R^2}\right)^{1/3}$ so the lower bound reads

$$\mathbb{E}F(\hat{x}) - F^* \geq \beta \cdot \frac{\left(H\sigma^2 B^4\right)}{K^{2/3} R^{2/3}} \tag{75}$$

To conclude, it is well known that any first-order method which accesses at most $MKR$ stochastic gradients with variance $\sigma^2$ for a convex objective with $\|x^*\| \leq B$ will suffer error at least $\beta \frac{\sigma B}{\sqrt{MKR}}$ in the worst case [17]. Therefore, the convex lower bound is

$$\mathbb{E}F(\hat{x}) - F^* \geq \beta \cdot \frac{\left(H\sigma^2 B^4\right)}{K^{2/3} R^{2/3}} + \beta \cdot \frac{\sigma B}{\sqrt{MKR}} \tag{76}$$

**Heterogeneity terms**

Next, we consider solely the first, second, and fourth terms of (67) in order to show a lower bound with respect to $\zeta_*$. Again, we consider three cases:

**Case 1** $\eta \geq \frac{2}{H}$**:**  Again, in this case, from the second term of (67) we see that

$$\mathbb{E}F(\hat{x}) - F^* \geq \frac{\mu c^2}{2} \tag{77}$$

**Case 2** $\eta \leq \frac{2}{H}$ **and** $(1-\mu\eta)^K > \frac{H-2\mu}{H}$**:**  In this case, from the first term of (67), we have

$$\mathbb{E}F(\hat{x}) - F^* \geq \frac{\mu c^2 (1-\mu\eta)^{KR}}{2} \tag{78}$$

$$\geq \frac{\mu c^2}{2}\left(1 - \frac{2\mu}{H}\right)^R \tag{79}$$

$$\geq \frac{\mu c^2}{2}\left(\left(1 - \frac{4\mu}{H}\left(1 - \frac{1}{e}\right)\right)^{\frac{H}{4\mu}}\right)^{\frac{4\mu R}{H}} \tag{80}$$

$$\geq \frac{\mu c^2}{2}\exp\left(-\frac{4\mu R}{H}\right) \tag{81}$$

**Case 3** $\eta \leq \frac{2}{H}$ **and** $(1-\mu\eta)^K \leq \frac{H-2\mu}{H}$**:**  In this case, from the first and fourth terms of (67), we have

$$\mathbb{E}F(\hat{x}) - F^* \geq \frac{\mu c^2}{2}(1-\mu\eta)^{KR} + \frac{\zeta_*^2(H+2\mu)}{32\mu^2}\left(\frac{H-2\mu}{H} - (1-\mu\eta)^K\right)^2 \tag{82}$$

Suppose that $(1-\mu\eta)^K \geq \frac{H-2\mu}{H} - \frac{1}{4R}$, then

$$\frac{\mu c^2}{2}(1-\mu\eta)^{KR} \geq \frac{\mu c^2}{2}\left(1 - \frac{2\mu}{H} - \frac{1}{4R}\right)^R \tag{83}$$

Then, if $R \geq \frac{H}{4\mu}$, then

$$\frac{\mu c^2}{2}(1-\mu\eta)^{KR} \geq \frac{\mu c^2}{2}\left(1 - \frac{3\mu}{H}\right)^R \geq \frac{\mu c^2}{2}\left(\left(1 - \frac{6\mu}{H}\left(1 - \frac{1}{e}\right)\right)^{\frac{H}{6\mu}}\right)^{\frac{6\mu R}{H}} \geq \frac{\mu c^2}{2}\exp\left(-\frac{6\mu R}{H}\right) \tag{84}$$

Otherwise, if $R \leq \frac{H}{4\mu}$, then

$$\frac{\mu c^2}{2}(1 - \mu\eta)^{KR} \geq \frac{\mu c^2}{2}\left(1 - \frac{1}{2R}\right)^R \geq \frac{\mu c^2}{4} \geq \frac{\mu c^2}{4}\exp\left(-\frac{6\mu R}{H}\right) \tag{85}$$

Therefore, when $(1 - \mu\eta)^K \geq \frac{H-2\mu}{H} - \frac{1}{4R}$,

$$\mathbb{E}F(\hat{x}) - F^* \geq \frac{\mu c^2}{4}\exp\left(-\frac{6\mu R}{H}\right) \tag{86}$$

On the other hand, if $(1 - \mu\eta)^K \leq \frac{H-2\mu}{H} - \frac{1}{4R}$, then

$$\mathbb{E}F(\hat{x}) - F^* \geq \frac{\zeta_*^2(H + 2\mu)}{32\mu^2}\left(\frac{H - 2\mu}{H} - (1 - \mu\eta)^K\right)^2 \tag{87}$$

$$\geq \frac{\zeta_*^2(H + 2\mu)}{32\mu^2}\left(\frac{1}{4R}\right)^2 \tag{88}$$

$$\geq \frac{H\zeta_*^2}{512\mu^2 R^2} \tag{89}$$

**Combination:** Combining these three cases, we conclude that

$$\mathbb{E}F(\hat{x}) - F^* \geq \min\left\{\frac{\mu c^2}{4}\exp\left(-\frac{6\mu R}{H}\right), \frac{H\zeta_*^2}{512\mu^2 R^2}\right\} \tag{90}$$

In the strongly convex case, we recall that $F(0) - F(x^*) = \mu c^2$, so we choose $\mu = \lambda$ and $c^2 = \frac{\Delta}{\lambda}$ so that the objective satisfies the strongly convex assumptions. Now, the lower bound reads (for a universal constant $\beta$)

$$\mathbb{E}F(\hat{x}) - F^* \geq \beta \cdot \min\left\{\Delta\exp\left(-\frac{6\lambda R}{H}\right), \frac{H\zeta_*^2}{512\lambda^2 R^2}\right\} \tag{91}$$

In the convex case, we recall that $\|x^*\|^2 \leq 2c^2$, so we choose $c^2 = \frac{B}{2}$ so that the convex assumptions are satisfied. We now have two options, if $R \leq \frac{H^2 B^2}{\zeta_*^2}$, then we pick $\mu = \left(\frac{H\zeta_*^2}{B^2 R^2}\right)^{1/3}$ so that the lower bound reads

$$\mathbb{E}F(\hat{x}) - F^* \geq \beta \cdot \frac{\left(H\zeta_*^2 B^4\right)^{1/3}}{R^{2/3}}\exp\left(-\frac{6\zeta_*^{2/3} R^{1/3}}{H^{2/3} B^{2/3}}\right) \tag{92}$$

$$\geq \beta \cdot \frac{\left(H\zeta_*^2 B^4\right)^{1/3}}{R^{2/3}}\exp(-6) \tag{93}$$

$$\geq \beta' \cdot \frac{\left(H\zeta_*^2 B^4\right)^{1/3}}{R^{2/3}} \tag{94}$$

On the other hand, if $R \geq \frac{H^2 B^2}{\zeta_*^2}$, then we pick $\mu = \frac{H}{6R}$ so the lower bound reads

$$\mathbb{E}F(\hat{x}) - F^* \geq \beta \cdot \min\left\{\frac{HB^2}{R}, \frac{\zeta_*^2}{H}\right\} = \beta \cdot \frac{HB^2}{R} \tag{95}$$

Consequently,

$$\mathbb{E}F(\hat{x}) - F^* \geq \beta \cdot \min\left\{\frac{HB^2}{R}, \frac{\left(H\zeta_*^2 B^4\right)^{1/3}}{R^{2/3}}\right\} \tag{96}$$

Combining these with the stochastic terms completes the proof. $\qquad\square$

# E   Proof of Theorem 3

We prove the theorem with the help of several technical lemmas.

**Lemma 7.** *For any stepsize $\eta_t \leq \frac{1}{10H}$*

$$\mathbb{E}[F(\bar{x}_t) - F^*] \leq \left(\frac{1}{\eta_t} - \lambda\right)\mathbb{E}\|\bar{x}_t - x^*\|^2 - \frac{1}{\eta_t}\mathbb{E}\|\bar{x}_{t+1} - x^*\|^2 + \frac{3\sigma_*^2\eta_t}{M} + \frac{2H}{M}\sum_{m=1}^{M}\mathbb{E}\|\bar{x}_t - x_t^m\|^2$$

*Proof.* This lemma and its proof are nearly identical to [Lemma 8 12]. We include a proof here in order to keep the paper self-contained.

Let $\bar{x}_{t+1} = \frac{1}{M}\sum_{m=1}^{M}x_t^m$ be the average of the machines' local iterates at time $t$. Then,

$$\mathbb{E}\|\bar{x}_{t+1} - x^*\|^2 = \mathbb{E}\left\|\bar{x}_t - \frac{\eta_t}{M}\sum_{m=1}^{M}\nabla F_m(x_t^m) - x^*\right\|^2 + \eta_t^2\mathbb{E}\left\|\frac{1}{M}\sum_{m=1}^{M}\nabla f(x_t^m; z_t^m) - \nabla F_m(x_t^m)\right\|^2 \tag{97}$$

Beginning with the first term of (97):

$$\mathbb{E}\left\|\bar{x}_t - \frac{\eta_t}{M}\sum_{m=1}^{M}\nabla F_m(x_t^m) - x^*\right\|^2$$

$$= \mathbb{E}\|\bar{x}_t - x^*\|^2 + \eta_t^2\mathbb{E}\left\|\frac{1}{M}\sum_{m=1}^{M}\nabla F_m(x_t^m)\right\|^2 - \frac{2\eta_t}{M}\sum_{m=1}^{M}\mathbb{E}\langle\bar{x}_t - x^*, \nabla F_m(x_t^m)\rangle \tag{98}$$

We can bound the second term of (98) with:

$$\eta_t^2\mathbb{E}\left\|\frac{1}{M}\sum_{m=1}^{M}\nabla F_m(x_t^m)\right\|^2$$

$$\leq 2\eta_t^2\mathbb{E}\left\|\frac{1}{M}\sum_{m=1}^{M}\nabla F_m(x_t^m) - \nabla F_m(\bar{x}_t)\right\|^2 + 2\eta_t^2\mathbb{E}\left\|\frac{1}{M}\sum_{m=1}^{M}\nabla F_m(\bar{x}_t) - \nabla F_m(x^*)\right\|^2 \tag{99}$$

$$\leq \frac{2\eta_t^2}{M}\sum_{m=1}^{M}\mathbb{E}\|\nabla F_m(x_t^m) - \nabla F_m(\bar{x}_t)\|^2 + 2\eta_t^2\mathbb{E}\|\nabla F(\bar{x}_t) - \nabla F(x^*)\|^2 \tag{100}$$

$$\leq \frac{2H^2\eta_t^2}{M}\sum_{m=1}^{M}\mathbb{E}\|x_t^m - \bar{x}_t\|^2 + 4H\eta_t^2\mathbb{E}[F(\bar{x}_t) - F(x^*)] \tag{101}$$

For the third term of (98):

$$-\frac{2\eta_t}{M}\sum_{m=1}^{M}\mathbb{E}\langle\bar{x}_t - x^*, \nabla F_m(x_t^m)\rangle$$

$$= -\frac{2\eta_t}{M}\sum_{m=1}^{M}\mathbb{E}\langle x_t^m - x^*, \nabla F_m(x_t^m)\rangle + \frac{2\eta_t}{M}\sum_{m=1}^{M}\mathbb{E}\langle x_t^m - \bar{x}_t, \nabla F_m(x_t^m)\rangle \tag{102}$$

$$\leq -\frac{2\eta_t}{M}\sum_{m=1}^{M}\mathbb{E}\left[F_m(x_t^m) - F_m(x^*) + \frac{\lambda}{2}\|x_t^m - x^*\|^2\right]$$

$$+ \frac{2\eta_t}{M}\sum_{m=1}^{M}\mathbb{E}\left[F_m(x_t^m) - F_m(\bar{x}_t) + \frac{H}{2}\|x_t^m - \bar{x}_t\|^2\right] \tag{103}$$

$$\leq -2\eta_t\mathbb{E}\left[F(\bar{x}_t) - F(x^*) + \frac{\lambda}{2}\|\bar{x}_t - x^*\|^2\right] + \frac{H\eta_t}{M}\sum_{m=1}^{M}\|x_t^m - \bar{x}_t\|^2 \tag{104}$$

Finally, for the second term of (97)

$$\eta_t^2 \mathbb{E}\left\| \frac{1}{M} \sum_{m=1}^{M} \nabla f(x_t^m; z_t^m) - \nabla F_m(x_t^m) \right\|^2$$

$$= \frac{\eta_t^2}{M^2} \sum_{m=1}^{M} \mathbb{E}\|\nabla f(x_t^m; z_t^m) - \nabla F_m(x_t^m)\|^2 \tag{105}$$

$$\leq \frac{3\eta_t^2}{M^2} \sum_{m=1}^{M} \left[ \mathbb{E}\|\nabla f(x_t^m; z_t^m) - \nabla f(\bar{x}_t; z_t^m)\|^2 + \mathbb{E}\|\nabla f(\bar{x}_t; z_t^m) - \nabla f(x^*; z_t^m)\|^2 \right.$$

$$\left. + \mathbb{E}\|\nabla f(x^*; z_t^m) - \nabla F_m(x^*)\|^2 \right] \tag{106}$$

$$\leq \frac{3\eta_t^2}{M^2} \sum_{m=1}^{M} \left[ H^2 \mathbb{E}\|x_t^m - \bar{x}_t\|^2 + 2H\mathbb{E}[F_m(\bar{x}_t) - F_m(x^*)] + \sigma_{*,m}^2 \right] \tag{107}$$

$$\leq \frac{3\eta_t^2}{M^2} \sum_{m=1}^{M} \left[ H^2 \mathbb{E}\|x_t^m - \bar{x}_t\|^2 + 2H\mathbb{E}[F(\bar{x}_t) - F(x^*)] + \sigma_{*,m}^2 \right] \tag{108}$$

Combining all these results back into (97), we have

$$\mathbb{E}\|\bar{x}_{t+1} - x^*\|^2 \leq (1 - \lambda\eta_t)\mathbb{E}\|\bar{x}_t - x^*\|^2 + \frac{H\eta_t + 5H^2\eta_t^2}{M} \sum_{m=1}^{M} \mathbb{E}\|x_t^m - \bar{x}_t\|^2$$

$$+ (10H\eta_t^2 - 2\eta_t)\mathbb{E}[F(\bar{x}_t) - F(x^*)] + \frac{3\eta_t^2\sigma_*^2}{M} \tag{109}$$

$$\leq (1 - \lambda\eta_t)\mathbb{E}\|\bar{x}_t - x^*\|^2 + \frac{2H\eta_t}{M} \sum_{m=1}^{M} \mathbb{E}\|x_t^m - \bar{x}_t\|^2$$

$$- \eta_t\mathbb{E}[F(\bar{x}_t) - F(x^*)] + \frac{3\eta_t^2\sigma_*^2}{M} \tag{110}$$

where for the final line we used that $\eta_t \leq \frac{1}{10H}$. Rearranging completes the proof. $\qquad\square$

**Lemma 8.** *If* $\sup_{x,m}\|\nabla F_m(x) - \nabla F(x)\|^2 \leq \bar{\zeta}^2$, *then for any fixed stepsize* $\eta$

$$\frac{1}{M} \sum_{m=1}^{M} \mathbb{E}\|x_t^m - \bar{x}_t\|^2 \leq 3K\sigma^2\eta^2 + 6K^2\eta^2\bar{\zeta}^2$$

*Similarly, the decreasing stepsize* $\eta_t = \frac{2}{\lambda(a+t+1)}$ *for any* $a$

$$\frac{1}{M} \sum_{m=1}^{M} \mathbb{E}\|x_t^m - \bar{x}_t\|^2 \leq 3K\sigma^2\eta_{t-1}^2 + 6K^2\bar{\zeta}^2\eta_{t-1}^2$$

*Proof.* By Jensen's inequality

$$\mathbb{E}\|x_t^m - \bar{x}_t\|^2 \leq \frac{1}{M} \sum_{n=1}^{M} \mathbb{E}\|x_t^m - x_t^n\|^2 \tag{111}$$

Therefore, it suffices to bound $\mathbb{E}\|x_t^m - x_t^n\|^2$, which we do now:

$$
\begin{aligned}
&\mathbb{E}\|x_t^m - x_t^n\|^2 \\
&\leq \mathbb{E}\left\| x_{t-1}^m - x_{t-1}^n - \eta_{t-1}\big(\nabla F(x_{t-1}^m) - \nabla F(x_{t-1}^n)\big) \right. \\
&\quad \left. + \eta_{t-1}\big(\nabla F(x_{t-1}^m) - \nabla F_m(x_{t-1}^m) - \nabla F(x_{t-1}^n) + \nabla F_n(x_{t-1}^n)\big) \right\|^2 + \eta_{t-1}^2 \sigma_m^2 \qquad (112)\\
&\leq \inf_{\gamma>0}\left(1 + \frac{1}{\gamma}\right)\mathbb{E}\left\| x_{t-1}^m - x_{t-1}^n - \eta_{t-1}\big(\nabla F(x_{t-1}^m) - \nabla F(x_{t-1}^n)\big) \right\|^2 \\
&\quad + (1+\gamma)\eta_{t-1}^2 \mathbb{E}\left\| \nabla F(x_{t-1}^m) - \nabla F_m(x_{t-1}^m) - \nabla F(x_{t-1}^n) + \nabla F_n(x_{t-1}^n) \right\|^2 + \eta_{t-1}^2 \sigma_m^2 \\
&\hspace{13cm}(113)\\
&\leq \inf_{\gamma>0}\left(1 + \frac{1}{\gamma}\right)(1-\lambda\eta_{t-1})\mathbb{E}\left\| x_{t-1}^m - x_{t-1}^n \right\|^2 + \eta_{t-1}^2\sigma_m^2 \\
&\quad + (1+\gamma)\eta_{t-1}^2 \mathbb{E}\left\| \nabla F(x_{t-1}^m) - \nabla F_m(x_{t-1}^m) \right\|^2 \\
&\quad + (1+\gamma)\eta_{t-1}^2 \mathbb{E}\left\| \nabla F(x_{t-1}^n) - \nabla F_n(x_{t-1}^n) \right\|^2 \\
&\quad - 2(1+\gamma)\eta_{t-1}^2 \mathbb{E}\left\langle \nabla F(x_{t-1}^m) - \nabla F_m(x_{t-1}^m),\, \nabla F(x_{t-1}^n) - \nabla F_n(x_{t-1}^n) \right\rangle \qquad (114)
\end{aligned}
$$

For the third inequality we used Lemma 1. Therefore,

$$
\begin{aligned}
&\frac{1}{M^2}\sum_{m=1}^{M}\sum_{n=1}^{M}\mathbb{E}\|x_t^m - x_t^n\|^2 \\
&\qquad \leq \frac{1}{M^2}\sum_{m=1}^{M}\inf_{\gamma>0}\left(1+\frac{1}{\gamma}\right)(1-\lambda\eta_{t-1})\mathbb{E}\left\| x_{t-1}^m - x_{t-1}^n \right\|^2 + \eta_{t-1}^2\sigma_m^2 + 2(1+\gamma)\eta_{t-1}^2\bar{\zeta}^2
\end{aligned}
$$
$$(115)$$

We will unroll this recurrence, using that $x_{t_0}^m = x_{t_0}^n$ for all $m, n$ where $t_0$ is the most recent time that the iterates were synchronized, so $t - t_0 \leq K - 1$. Taking $\gamma = K - 1$, we have

$$
\begin{aligned}
\frac{1}{M^2}\sum_{m=1}^{M}\sum_{n=1}^{M}\mathbb{E}\|x_t^m - x_t^n\|^2 &= \sum_{i=t_0}^{t-1}\big(\eta_i^2\sigma^2 + 2(1+\gamma)\eta_i^2\bar{\zeta}^2\big)\prod_{j=i+1}^{t-1}\left(1+\frac{1}{\gamma}\right)(1-\lambda\eta_j) \qquad (116)\\
&\leq \sum_{i=t_0}^{t-1}\big(\eta_i^2\sigma^2 + 2K\eta_i^2\bar{\zeta}^2\big)\prod_{j=i+1}^{t-1}\left(1+\frac{1}{K-1}\right)(1-\lambda\eta_j) \qquad (117)\\
&\leq \sum_{i=t_0}^{t-1}\big(\eta_i^2\sigma^2 + 2K\eta_i^2\bar{\zeta}^2\big)\left(1+\frac{1}{K-1}\right)^{K-1}\prod_{j=i+1}^{t-1}(1-\lambda\eta_j) \quad (118)\\
&\leq 3\big(\sigma^2 + 2K\bar{\zeta}^2\big)\sum_{i=t_0}^{t-1}\eta_i^2\prod_{j=i+1}^{t-1}(1-\lambda\eta_j) \qquad (119)
\end{aligned}
$$

For a constant stepsize $\eta$,

$$
\begin{aligned}
\frac{1}{M^2}\sum_{m=1}^{M}\sum_{n=1}^{M}\mathbb{E}\|x_t^m - x_t^n\|^2 &\leq 3\big(\sigma^2 + 2K\bar{\zeta}^2\big)\sum_{i=t_0}^{t-1}\eta^2 \qquad (120)\\
&\leq 3K\big(\sigma^2 + 2K\bar{\zeta}^2\big)\eta^2 \qquad (121)
\end{aligned}
$$

For decreasing stepsize $\eta_t = \frac{2}{\lambda(a+t+1)}$

$$\frac{1}{M^2}\sum_{m=1}^{M}\sum_{n=1}^{M}\mathbb{E}\|x_t^m - x_t^n\|^2 \leq 3(\sigma^2 + 2K\bar{\zeta}^2)\sum_{i=t_0}^{t-1}\eta_i^2 \prod_{j=i+1}^{t-1}\frac{a+j-1}{a+j+1} \tag{122}$$

$$= 3(\sigma^2 + 2K\bar{\zeta}^2)\sum_{i=t_0}^{t-1}\eta_i^2 \frac{(a+i)(a+i+1)}{(a+t)(a+t+1)} \tag{123}$$

$$= 3(\sigma^2 + 2K\bar{\zeta}^2)\sum_{i=t_0}^{t-1}\eta_i^2 \frac{\eta_{t-1}\eta_t}{\eta_{i-1}\eta_i} \tag{124}$$

$$\leq 3(\sigma^2 + 2K\bar{\zeta}^2)\sum_{i=t_0}^{t-1}\eta_i^2 \frac{\eta_{t-1}^2}{\eta_i^2} \tag{125}$$

$$= 3K(\sigma^2 + 2K\bar{\zeta}^2)\eta_{t-1}^2 \tag{126}$$

$\square$

**Theorem 3.** *When* $\sup_x \frac{1}{M}\sum_{m=1}^{M}\|\nabla F_m(x) - \nabla F(x)\|^2 \leq \bar{\zeta}^2$, *an average of the Local SGD iterates guarantees under the convex and strongly convex assumptions, respectively*

$$\mathbb{E}F(\hat{x}) - F^* \leq c \cdot \left(\frac{HB^2}{KR} + \frac{(H\bar{\zeta}^2 B^4)^{1/3}}{R^{2/3}} + \frac{(H\sigma^2 B^4)^{1/3}}{K^{1/3}R^{2/3}} + \frac{\sigma_* B}{\sqrt{MKR}}\right),$$

$$\mathbb{E}F(\hat{x}) - F^* \leq c \cdot \left(\frac{H^2 B^2}{HKR + \lambda K^2 R^2} + \left(\frac{H\bar{\zeta}^2}{\lambda^2 R^2} + \frac{H\sigma^2}{\lambda^2 KR^2}\right)\log\left(\frac{H}{\lambda} + KR\right) + \frac{\sigma_*^2}{\lambda MKR}\right).$$

*Proof.* By Lemma 7, for any $\eta_t \leq \frac{1}{10H}$

$$\mathbb{E}[F(\bar{x}_t) - F^*] \leq \left(\frac{1}{\eta_t} - \lambda\right)\mathbb{E}\|\bar{x}_t - x^*\|^2 - \frac{1}{\eta_t}\mathbb{E}\|\bar{x}_{t+1} - x^*\|^2 + \frac{3\sigma_*^2 \eta_t}{M} + \frac{2H}{M}\sum_{m=1}^{M}\mathbb{E}\|\bar{x}_t - x_t^m\|^2 \tag{127}$$

By Lemma 8, when $\eta_t = \eta$ is constant then

$$\frac{1}{M}\sum_{m=1}^{M}\mathbb{E}\|x_t^m - \bar{x}_t\|^2 \leq 3K\sigma^2\eta^2 + 6K^2\eta^2\bar{\zeta}^2 \tag{128}$$

and when $\eta_t = \frac{2}{\lambda(a+t+1)}$

$$\frac{1}{M}\sum_{m=1}^{M}\mathbb{E}\|x_t^m - \bar{x}_t\|^2 \leq 3K\sigma^2\eta_{t-1}^2 + 6K^2\bar{\zeta}^2\eta_{t-1}^2 \tag{129}$$

We now consider the convex and strongly convex cases separately:

**Convex case:** In the convex case, we use a constant stepsize $\eta$, so

$$\mathbb{E}[F(\bar{x}_t) - F^*]$$

$$\leq \frac{1}{\eta}\mathbb{E}\|\bar{x}_t - x^*\|^2 - \frac{1}{\eta}\mathbb{E}\|\bar{x}_{t+1} - x^*\|^2 + \frac{3\sigma_*^2\eta}{M} + \frac{2H}{M}\sum_{m=1}^{M}\mathbb{E}\|\bar{x}_t - x_t^m\|^2 \tag{130}$$

$$\leq \frac{1}{\eta}\mathbb{E}\|\bar{x}_t - x^*\|^2 - \frac{1}{\eta}\mathbb{E}\|\bar{x}_{t+1} - x^*\|^2 + \frac{3\sigma_*^2\eta}{M} + 6HK\sigma^2\eta^2 + 12HK^2\eta^2\bar{\zeta}^2 \tag{131}$$

Therefore, by the convexity of $F$

$$\mathbb{E}\left[F\left(\frac{1}{KR}\sum_{t=1}^{KR}\bar{x}_t\right) - F^*\right] \leq \frac{1}{KR}\sum_{t=1}^{KR}\mathbb{E}[F(\bar{x}_t) - F^*] \tag{132}$$

$$\leq \frac{B^2}{\eta KR} + \frac{3\sigma_*^2\eta}{M} + 6HK\sigma^2\eta^2 + 12HK^2\eta^2\bar{\zeta}^2 \tag{133}$$

Choosing

$$\eta = \min\left\{ \frac{1}{10H}, \frac{B\sqrt{M}}{\sigma_*\sqrt{KR}}, \left(\frac{B^2}{HK^2\sigma^2}\right)^{1/3}, \left(\frac{B^2}{HK^2\bar{\zeta}^2}\right)^{1/3} \right\} \tag{134}$$

then ensures

$$\mathbb{E}\left[ F\left( \frac{1}{KR}\sum_{t=1}^{KR} \bar{x}_t \right) - F^* \right] \leq \frac{10HB^2}{KR} \frac{13\left(H\bar{\zeta}^2 B^4\right)^{1/3}}{R^{2/3}} + \frac{7\left(H\sigma^2 B^4\right)^{1/3}}{K^{1/3}R^{2/3}} + \frac{4\sigma_* B}{\sqrt{MKR}} \tag{135}$$

**Strongly convex case:** In the strongly convex case, we take the stepsize $\eta_t = \frac{2}{\lambda(a+t+1)}$ for $a = 20H/\lambda$ which ensures $\eta_t \leq \frac{1}{10H}$. In addition, we define weights $w_t = (a+t)$ and define

$$\bar{x} = \frac{1}{W}\sum_{t=1}^{KR} w_t \bar{x}_t \tag{136}$$

where $W = \sum_{t=1}^{KR} w_t \geq \frac{1}{2}KR(a + KR)$. By the convexity of $F$,

$$\mathbb{E}F(\bar{x}) - F^*$$

$$\leq \frac{1}{W}\sum_{t=1}^{KR}(a+t)\mathbb{E}F(\bar{x}_t) - F^* \tag{137}$$

$$\leq \frac{\lambda(a+1)(a+2)B^2}{2W} + \frac{1}{W}\sum_{t=1}^{KR}\left[ \frac{6\sigma_*^2}{\lambda M} + \frac{2H(a+t)}{M}\sum_{m=1}^{M}\mathbb{E}\|\bar{x}_t - x_t^m\|^2 \right] \tag{138}$$

$$\leq \frac{\lambda(a+1)(a+2)B^2}{2W} + \frac{6\sigma_*^2 KR}{W\lambda M} + \frac{6HK\sigma^2 + 12HK^2\bar{\zeta}^2}{W}\sum_{t=1}^{KR}(a+t)\eta_{t-1}^2 \tag{139}$$

$$\leq \frac{\lambda(a+1)(a+2)B^2}{2W} + \frac{6\sigma_*^2 KR}{W\lambda M} + \frac{6HK\sigma^2 + 12HK^2\bar{\zeta}^2}{\lambda^2 W}(1 + \log(a + KR)) \tag{140}$$

$$\leq \frac{132H^2 B^2}{\lambda KR(10H/\lambda + KR)} + \left(\frac{12H\bar{\zeta}^2}{\lambda^2 R^2} + \frac{6H\sigma^2}{\lambda^2 KR^2}\right)\log\left(\frac{13H}{\lambda} + KR\right) + \frac{6\sigma_*^2}{\lambda MKR} \tag{141}$$

Note that in the strongly convex case, it is likely possible to achieve a first term scaling with $\exp(-KR)$ using a method similar to Lemma 2. However, the recurrence we derived here has a different form, and it is difficult to determine the correct stepsize and weighting schedule to achieve linear convergence. $\qquad\square$

## F Details of Experiments

The training set of MNIST (60,000 examples) was divided by digit into ten groups of equal size $n \approx 6,000$ (which required discarding some examples from the more common digits). PCA was used to reduce the dimensionality to 100, but no other preprocessing was used.

Then, for each of the 25 combinations $(i,j)$ for even $i$ and odd $j$, a binary classification "task" was created, i.e. classifying even $(+1)$ versus odd $(-1)$. These tasks were arbitrarily labelled task $1, 2, \ldots, 25$.

For each $p \in [0.0, 0.2, 0.4, 0.6, 0.8, 1.0]$, machine $m$ was assigned data composed of $p \cdot 2n$ random examples from task $m$, and $(1-p) \cdot 2n$ random examples from a mixture of all the tasks.

Local and Minibatch SGD were then used to optimize the logistic loss for each of the six described local datasets. The constant stepsize was tuned (from a log-scale grid of 10 points ranging from $e^{-6}, \ldots, e^0$ for Minibatch SGD, and a log-scale grid of 10 points ranging from $e^{-8}, \ldots, e^{-1}$ for Local SGD) for each value of $p$, $K$, and $R$ individually, and the average loss over four runs is reported for the best stepsize for each point in the plot. That is, each point in the plot represents the best possible performance of the algorithm for that $p$, $K$, and $R$ specifically.

Finally, we computed the value of $\zeta_*^2$ as a function of $p$ by using Newton's method to compute a very accurate estimate of the minimizer, and then explicitly calculating $\zeta_*^2(p)$ at that point.

## G   Proof of Theorem 4

For this lower bound, the gradients will always be noiseless, so we simply define the expectation of the local functions. Furthermore, we will construct just two local functions $F_1$ and $F_2$. For the case $M > 2$, $F_1$ will be assigned to the first $\lfloor M/2 \rfloor$ machines, and $F_2$ to the next $\lfloor M/2 \rfloor$ machines. If there is an odd number of machines, we simply assign the last machine $F_3(x) = \frac{\lambda}{2}\|x\|^2$, which will reduce the lower bound by a factor of at most $\frac{M-1}{M}$. Therefore, we proceed by focusing on the case $M = 2$.

We define the following $H$-smooth and $\lambda$-strongly convex functions on $\mathbb{R}^d$ for even $d$:

$$F(x) = \frac{1}{2}(F_1(x) + F_2(x)) \tag{142}$$

$$F_1(x) = \frac{H - \lambda}{8}\left(x_1^2 - 2Cx_1 + \beta x_d^2 + \sum_{i=1}^{d/2-1}(x_{2i+1} - x_{2i})^2\right) + \frac{\lambda}{2}\|x\|^2 \tag{143}$$

$$F_2(x) = \frac{H - \lambda}{8}\left(\sum_{i=1}^{d/2}(x_{2i} - x_{2i-1})^2\right) + \frac{\lambda}{2}\|x\|^2 \tag{144}$$

Here, $\beta$ and $C$ are constants which will be chosen later.

These functions are identical to ones used by Woodworth and Srebro [24] to prove lower bounds for finite sum optimization, and are very similar both to classic work by Nesterov [18] on lower bounds and to more closely related work by Arjevani and Shamir [1]. Arjevani and Shamir also prove lower bounds for distributed optimization algorithms, but their slightly different construction made it more difficult to tune $\zeta_*^2$, which is necessary for our lower bound.

These functions have the following important property: let $E_k = \text{span}\{e_1, \ldots, e_k\}$ be the set of vectors whose $k + 1, \ldots, d$ coordinates are all zero, then for all $x_k \in E_k$ for even $k$

$$\nabla F_1(x_k) \in E_{k+1} \qquad \text{and} \qquad \nabla F_2(x_k) \in E_k \tag{145}$$

and for $x_k \in E_k$ for odd $k$

$$\nabla F_1(x_k) \in E_k \qquad \text{and} \qquad \nabla F_2(x_k) \in E_{k+1} \tag{146}$$

For algorithms whose iterates, for example, remain in the span of previous gradients, the only way to access the next coordinate is to query the gradient of one of the two functions—$F_1$ if the next coordinate is odd, and $F_2$ if the next coordinate is even. Since each machine will only have access to one of the two functions throughout each round of communication, this means that each round of communication can only unlock a single new coordinate. We now formalize this.

Following Carmon et al. [5], we define:

**Definition 2** (Distributed zero-respecting algorithm). *For a vector $v$, let $\text{supp}(v) = \{i \in \{1, \ldots, d\} : v_i \neq 0\}$. We say that an optimization algorithm is distributed zero-respecting if for all $t$ and $m$, the $t$th query on the $m$th machine, $x_t^m$ satisfies*

$$\text{supp}(x_t^m) \subseteq \bigcup_{s<t} \text{supp}(\nabla f(x_s^m; z_s^m)) \cup \bigcup_{m' \neq m} \bigcup_{s \leq \pi_m(t, m')} \text{supp}(\nabla f(x_s^{m'}; z_s^{m'}))$$

*where $\pi_m(t, m')$ is the most recent time before $t$ when machines $m$ and $m'$ communicated with each other.*

This definition captures a very wide variety of distributed optimization algorithms, including minibatch SGD, accelerated minibatch SGD, local SGD, coordinate descent methods, and many more. Algorithms which are *not* distributed zero-respecting are those whose iterates have components in directions about which the algorithm has no information, meaning that in some sense, it is just "wild guessing." Using techniques similar to Woodworth and Srebro [Theorem 7 24] and Carmon et al. [5], it should be possible to extend this lower bound beyond distributed zero-respecting algorithms to arbitrary randomized algorithms.

We now argue that the progress of distributed zero-respecting algorithms is controlled by the number of rounds of communication, $R$, regardless of $K$:

**Lemma 9.** *Let $\hat{x}$ be the output after $R$ rounds of communication of a distributed zero-respecting algorithm optimizing $F = \frac{1}{2}(F_1 + F_2)$ as defined in (142). Then,*

$$\text{supp}(x_t^m) \in E_R$$

*Proof.* The definition of a zero-respecting algorithm requires that every machine's initial iterate $x_0^m = 0$. We will now prove the Lemma by induction on the round of communication.

As a base case, for the first iteration of the first round of communication:

$$\nabla F_1(x_0^1) = \nabla F_1(0) = \frac{(\lambda - H)C}{4}e_1 \in E_1 \qquad \text{and} \qquad \nabla F_2(x_1^2) = \nabla F_2(0) = 0 \in E_0 \quad (147)$$

Therefore, by the distributed zero-respecting property, $x_2^1 \in E_1$ and $x_2^2 \in E_0$. Furthermore, for all $y_1 \in E_1$, $\nabla F_1(y_1) \in E_1$ and for all $y_0 \in E_0$, $\nabla F_2(y_0) \in E_0$. Therefore, further gradient queries on each machine will not change the set of coordinates that the distributed zero-respecting property allows to be non-zero. We conclude that $x_t^1 \in E_1$ and $x_t^2 \in E_0$ for all $t$ until machines $1$ and $2$ communicate with each other.

Now, suppose that after $r-1$ rounds of communication, $x_t^1, x_t^2 \in E_{r-1}$. If $r$ is even, then $\nabla F_1(x_t^1) \in E_{r-1}$ and $\nabla F_2(x_t^2) \in E_r$. Furthermore, additional gradient computations within the $r$th round of communication will not expand the set of coordinates that the distributed zero-respecting property will allow to be non-zero. Therefore, both machines' coordinates will remain in $E_r$ until the end of the $r$th round of communication. A similar argument can be made for odd $r$. $\qquad\square$

Now, we will compute the minimizer of $F$. We note that by the definition of $F_1$ and $F_2$,

$$F(x) = \frac{H - \lambda}{16}\left(x_1^2 - 2Cx_1 + \beta x_d^2 + \sum_{i=2}^{d}(x_i - x_{i-1})^2\right) + \frac{\lambda}{2}\|x\|^2 \qquad (148)$$

Calculating the gradient of $F$, we see that $x^* = \arg\min_x F(x)$ must satisfy

$$C = \left(2 + \frac{8\lambda}{H - \lambda}\right)x_1^* - x_2^*$$

$$0 = \left(2 + \frac{8\lambda}{H - \lambda}\right)x_i^* - x_{i+1}^* - x_{i-1}^* \qquad \forall_{i \in \{2,\dots,d-1\}} \qquad (149)$$

$$0 = \left(1 + \beta + \frac{8\lambda}{H - \lambda}\right)x_d^* - x_{d-1}^*$$

Let $q$ be the smaller solution of the quadratic equation

$$1 - \left(2 + \frac{8\lambda}{H - \lambda}\right)q + q^2 = 0 \qquad (150)$$

That is,

$$q = 1 + \frac{4\lambda}{H - \lambda} - \sqrt{\frac{16\lambda^2}{(H - \lambda)^2} + \frac{8\lambda}{H - \lambda}} \qquad (151)$$

$$= 1 + \frac{4\lambda}{H - \lambda}\left(1 - \sqrt{1 + \frac{H - \lambda}{2\lambda}}\right) \qquad (152)$$

$$= 1 - \frac{2}{\left(1 - \sqrt{1 + \frac{H-\lambda}{2\lambda}}\right)\left(1 + \sqrt{1 + \frac{H-\lambda}{2\lambda}}\right)}\left(1 - \sqrt{1 + \frac{H - \lambda}{2\lambda}}\right) \qquad (153)$$

$$= \frac{\sqrt{1 + \frac{H-\lambda}{2\lambda}} - 1}{\sqrt{1 + \frac{H-\lambda}{2\lambda}} + 1} \qquad (154)$$

Let $\alpha = \sqrt{1 + \frac{H-\lambda}{2\lambda}}$ so that $q = \frac{\alpha-1}{\alpha+1}$, and define $\beta = 1 - q$. Then it is straightforward to confirm that

$$x^* = C \sum_{i=1}^{d} q^i e_i \tag{155}$$

satisfies all of the conditions (149), and is thus the minimizer of $F$. This point has value

$$F(x^*) = \frac{C^2(H-\lambda)}{16}\left(q^2 - 2q + \beta q^{2d} + (1-q)^2 \sum_{i=2}^{d} q^{2i-2} + \frac{8\lambda}{H-\lambda}\sum_{i=1}^{d} q^{2i}\right) \tag{156}$$

$$= \frac{C^2(H-\lambda)}{16}\left(-1 + \beta q^{2d} + (1-q)^2 \sum_{i=1}^{d} q^{2i-2} + \frac{8\lambda}{H-\lambda}\sum_{i=1}^{d} q^{2i}\right) \tag{157}$$

$$= \frac{C^2(H-\lambda)}{16}\left(-1 + (1-q)q^{2d} + \frac{8\lambda}{H-\lambda}\sum_{i=1}^{d} q^{2i-1} + q^{2i}\right) \tag{158}$$

$$= \frac{C^2(H-\lambda)}{16}\left(-1 + (1-q)q^{2d} + \frac{8\lambda}{H-\lambda}\left(\frac{q(1-q^{2d})}{1-q^2} + \frac{q^2(1-q^{2d})}{1-q^2}\right)\right) \tag{159}$$

$$= \frac{C^2(H-\lambda)}{16}\left(-1 + (1-q)q^{2d} + \frac{(1-q)^2}{q}\left(\frac{q(1-q^{2d})}{1-q^2} + \frac{q^2(1-q^{2d})}{1-q^2}\right)\right) \tag{160}$$

$$= \frac{C^2(H-\lambda)}{16}\left(-1 + (1-q)q^{2d} + (1-q)(1-q^{2d})\right) \tag{161}$$

$$= \frac{-qC^2(H-\lambda)}{16} \tag{162}$$

For the third equality, we used that (150) implies $(1-q)^2 = \frac{8\lambda q}{H-\lambda}$. For the fifth inequality, we used that $\frac{8\lambda}{H-\lambda} = \frac{(1-q)^2}{q}$. This solution has norm

$$\|x^*\|^2 = C^2 \sum_{i=1}^{d} q^{2i} = C^2 \frac{q^2(1-q^{2d})}{1-q^2} \leq \frac{q^2 C^2}{1-q^2} = \frac{C^2(\alpha-1)^2}{4\alpha} \leq \frac{\alpha C^2}{4} \tag{163}$$

Furthermore,

$$F(0) - F(x^*) = -F(x^*) = \frac{qC^2(H-\lambda)}{16} \tag{164}$$

Finally, we evaluate the degree of heterogeneity:

$$\zeta_*^2 = \frac{1}{2}\sum_{m=1}^{2}\|\nabla F_m(x^*)\|^2 = \|\nabla F_1(x^*)\|^2 = \|\nabla F_2(x^*)\|^2 \tag{165}$$

$$= \frac{(H-\lambda)^2}{64}\left\|2\sum_{i=1}^{d/2}(x_{2i}^* - x_{2i-1}^*)(e_{2i} - e_{2i-1}) + \frac{4\lambda}{H-\lambda}x^*\right\|^2 \tag{166}$$

$$= \frac{(H-\lambda)^2}{16}\sum_{i=1}^{d/2}\left[\left(x_{2i-1}^*\left(-1 + \frac{4\lambda}{H-\lambda}\right)\right)^2 + \left(x_{2i}^*\left(1 + \frac{4\lambda}{H-\lambda}\right)\right)^2\right] \tag{167}$$

$$= \frac{C^2(H-\lambda)^2}{16}\sum_{i=1}^{d/2}\left[q^{4i-2}\frac{(H-5\lambda)^2}{(H-\lambda)^2} + q^{4i}\frac{(H+3\lambda)^2}{(H-\lambda)^2}\right] \tag{168}$$

$$\leq \frac{(H+3\lambda)^2}{16}\|x^*\|^2 \tag{169}$$

$$\leq \frac{\alpha C^2(H+3\lambda)^2}{64} \tag{170}$$

With this, we are ready to prove the lower bound.

**Theorem 4.** *For any $M$, $K$, and $R$, there exist two quadratic objectives satisfying the convex and strongly convex assumptions (for $H \geq 7\lambda$) such that the output of any distributed zero-respecting algorithm will have suboptimality in the convex and strongly convex case respectively,*

$$F(\hat{x}) - F^* \geq c\left(\min\left\{\frac{\zeta_*^2}{HR^2}, \frac{HB^2}{R^2}\right\} + \frac{\sigma B}{\sqrt{MKR}}\right),$$

$$F(\hat{x}) - F^* \geq c\left(\min\left\{\frac{\lambda\zeta_*^2}{H^2}, \frac{\Delta\sqrt{\lambda}}{\sqrt{H}}\right\}\exp\left(-\frac{8R\sqrt{\lambda}}{\sqrt{H}}\right) + \frac{\sigma^2}{\lambda MKR}\right).$$

*Proof.* By Lemma 9, the output of the algorithm $\hat{x} \in E_R$. Furthermore, since $F$ is $\lambda$-strongly convex, $F(\hat{x}) - F^* \geq \frac{\lambda}{2}\|\hat{x} - x^*\|^2$. Therefore,

$$\frac{F(\hat{x}) - F^*}{F(0) - F^*} \geq \frac{\frac{\lambda}{2}\|\hat{x} - x^*\|^2}{\frac{qC^2(H-\lambda)}{16}} \tag{171}$$

$$\geq \frac{8\lambda}{q(H-\lambda)}\sum_{i=R+1}^{d} q^{2i} \tag{172}$$

$$= \frac{8\lambda q(q^{2R} - q^{2d})}{(H-\lambda)(1-q^2)} \tag{173}$$

$$= \frac{(1-q)^2(q^{2R} - q^{2d})}{1-q^2} \tag{174}$$

$$= \frac{(1-q)(q^{2R} - q^{2d})}{1+q} \tag{175}$$

$$= \frac{q^{2R} - q^{2d}}{\alpha} \tag{176}$$

For the third equality we used that (150) implies $\frac{8\lambda q}{H-\lambda} = (1-q)^2$. For the final equality, we used that $q = \frac{\alpha-1}{\alpha+1}$. Taking $d \geq R + \frac{1}{2\ln(1/q)}$ ensures that $q^{2d} \leq \frac{q^{2R}}{2}$ so

$$\frac{F(\hat{x}) - F^*}{F(0) - F^*} \geq \frac{q^{2R}}{2\alpha} = \frac{\left(1 - \frac{2}{\alpha+1}\right)^{2R}}{2\alpha} \tag{177}$$

Therefore,

$$R \leq \frac{\ln\left(\frac{F(0)-F^*}{2\alpha\epsilon}\right)}{\ln\left(1 + \frac{2}{\alpha-1}\right)} \implies F(\hat{x}) - F^* \geq \epsilon \tag{178}$$

Using the fact that $\ln(1+x) \leq x$ and solving the above inequality on $R$ for $\epsilon$, we conclude that

$$F(\hat{x}) - F^* \geq \frac{F(0) - F^*}{2\alpha}\exp\left(-\frac{2R}{\alpha-1}\right) \tag{179}$$

In order to satisfy the strongly convex assumptions, we recall from (170) and (164) that we must choose $C$ such that

$$\frac{\alpha C^2(H+3\lambda)^2}{64} \leq \frac{\alpha C^2 H^2}{16} \leq \zeta_*^2 \tag{180}$$

$$\frac{qC^2(H-\lambda)}{16} \leq \frac{C^2 H}{16} \leq \Delta \tag{181}$$

Therefore, we choose $C^2 = 16 \min\left\{\frac{\zeta_*^2}{\alpha H^2}, \frac{\Delta}{H}\right\}$ meaning that

$$F(\hat{x}) - F^* \geq \frac{F(0) - F^*}{2\alpha} \exp\left(-\frac{2R}{\alpha - 1}\right) \tag{182}$$

$$\geq \frac{\min\left\{\frac{\zeta_*^2}{\alpha H}, \Delta\right\}}{2\alpha} \exp\left(-\frac{2R}{\alpha - 1}\right) \tag{183}$$

$$\geq \min\left\{\frac{\lambda\zeta_*^2}{H^2}, \frac{\sqrt{\lambda}\Delta}{2\sqrt{H}}\right\} \exp\left(-\frac{8\sqrt{\lambda}R}{\sqrt{H}}\right) \tag{184}$$

For the convex case, we note that in order to satisfy the convex assumptions, we must choose $C$ such that

$$\frac{\alpha C^2 (H + 3\lambda)^2}{64} \leq \frac{\alpha C^2 H^2}{16} \leq \zeta_*^2 \tag{185}$$

$$\frac{\alpha C^2}{4} \leq B^2 \tag{186}$$

We therefore choose $C^2 = 4 \min\left\{\frac{\zeta_*^2}{\alpha H^2}, \frac{B^2}{\alpha}\right\}$. Returning to (179), this means

$$F(\hat{x}) - F^* \geq \frac{F(0) - F^*}{2\alpha} \exp\left(-\frac{2R}{\alpha - 1}\right) \tag{187}$$

$$= \frac{qC^2 (H - \lambda)}{32\alpha} \exp\left(-\frac{2R}{\alpha - 1}\right) \tag{188}$$

$$\geq \frac{q(H - \lambda) \min\left\{\frac{\zeta_*^2}{\alpha H^2}, \frac{B^2}{\alpha}\right\}}{8\alpha} \exp\left(-\frac{8\sqrt{\lambda}R}{\sqrt{H}}\right) \tag{189}$$

$$\geq q \min\left\{\frac{\zeta_*^2}{16\alpha^2 H}, \frac{HB^2}{16\alpha^2}\right\} \exp\left(-\frac{8\sqrt{\lambda}R}{\sqrt{H}}\right) \tag{190}$$

From here, we use that $H \geq 7\lambda$ implies $\alpha \geq 2$ so $q \geq 1/3$, so

$$F(\hat{x}) - F^* \geq \min\left\{\frac{\lambda\zeta_*^2}{48H^2}, \frac{\lambda B^2}{48}\right\} \exp\left(-\frac{8\sqrt{\lambda}R}{\sqrt{H}}\right) \tag{191}$$

Finally, this holds for any $\lambda \geq 0$, so it holds, in particular, for $\lambda = \frac{H}{64R^2}$ thus

$$F(\hat{x}) - F^* \geq c \cdot \min\left\{\frac{\zeta_*^2}{HR^2}, \frac{HB^2}{R^2}\right\} \tag{192}$$

Finally, it is well known that any first-order method which accesses at most $MKR$ stochastic gradients with variance $\sigma^2$ for a $\lambda$-strongly convex objective will suffer error at least $\beta \frac{\sigma^2}{\lambda MKR}$ in the worst case for a universal constant $\beta$ [17]. Similarly, any first-order method which accesses at most $MKR$ stochastic gradients with variance $\sigma^2$ for a convex objective with $\|x^*\| \leq B$ will suffer error at least $\beta \frac{\sigma B}{\sqrt{MKR}}$ in the worst case for a universal constant $\beta$ [17]. $\square$

## H    Proof of Corollary 1

**Corollary 1.** *Accelerated Minibatch SGD is optimal when $\zeta_* \geq HB$ in the convex case, and is optimal up to log factors when $\zeta_*^2 \geq H^{3/2}/\sqrt{\lambda}$ in the strongly convex case.*

*Proof.* In the convex case, Theorem 1 ensures Accelerated Minibatch SGD converges at a rate proportional to

$$\frac{HB^2}{R^2} + \frac{\sigma B}{\sqrt{MKR}} \tag{193}$$

The lower bound for convex functions in Theorem 4 precisely matches this whenever

$$\frac{HB^2}{R^2} = \min\left\{\frac{\zeta_*^2}{HR^2}, \frac{HB^2}{R^2}\right\} \implies \zeta_* \geq HB \tag{194}$$

For the strongly convex case, Theorem 1 ensures convergence at a rate porportional to

$$\Delta \exp\left(-\frac{\sqrt{\lambda}R}{c_3\sqrt{H}}\right) + \frac{\sigma^2}{\lambda MKR} \tag{195}$$

The lower bound is given by

$$\min\left\{\frac{\lambda\zeta_*^2}{H^2}, \frac{\Delta\sqrt{\lambda}}{\sqrt{H}}\right\} \exp\left(-\frac{8R\sqrt{\lambda}}{\sqrt{H}}\right) + \frac{\sigma^2}{\lambda MKR} \tag{196}$$

When $\zeta_*^2 \geq H^{3/2}/\sqrt{\lambda}$, this reduces to

$$\frac{\Delta\sqrt{\lambda}}{\sqrt{H}} \exp\left(-\frac{8R\sqrt{\lambda}}{\sqrt{H}}\right) + \frac{\sigma^2}{\lambda MKR} = \Delta \exp\left(-\frac{8R\sqrt{\lambda}}{\sqrt{H}} - \log\frac{\sqrt{\lambda}}{\sqrt{H}}\right) + \frac{\sigma^2}{\lambda MKR} \tag{197}$$

Comparing this with (195), we see that the $R$ needed to guarantee error $\epsilon$ using Theorem 1 is larger than the minimum possible $R$, as lower bounded by (197), by at most a log factor. $\square$

[Supplementary Material 2 · supplementary.pdf]

| Method/Analysis | Worst-Case Error (i.e. $\mathbb{E}F(\hat{x}) - F^* \lesssim$) |
|---|---|
| Minibatch SGD<br>Theorem 1 | $\frac{H\Delta}{\lambda}\exp\left(\frac{-\lambda R}{H}\right) + \frac{\sigma_*^2}{\lambda MKR}$ |
| Accelerated Minibatch SGD<br>Theorem 1 | $\Delta\exp\left(\frac{-\sqrt{\lambda}R}{\sqrt{H}}\right) + \frac{\sigma^2}{\lambda MKR}$ |
| Local SGD<br>Koloskova et al. [12] | $\frac{\sigma_*^2}{\lambda MKR} + \frac{H\zeta_*^2}{\lambda^2 R^2} + \frac{H\sigma_*^2}{\lambda^2 KR^2}$      for $R \geq \tilde{\Omega}\left(\frac{H}{\lambda}\log\Delta\right)$ |
| SCAFFOLD<br>Karimireddy et al. [11] | $\left(H\Delta + \frac{\lambda\zeta_*^2}{H^2}\right)\exp\left(\frac{-\lambda R}{H}\right) + \frac{\sigma^2}{\lambda MKR}$ |
| Local SGD<br>Theorem 3 | $\frac{HB^2}{HKR + \lambda K^2 R^2} + \frac{\sigma_* B}{\sqrt{MKR}} + \frac{H\bar{\zeta}^2}{\lambda^2 R^2} + \frac{H\sigma^2}{\lambda^2 KR^2}$ |
| Local SGD Lower Bound<br>Theorem 2 | $\min\left\{\Delta\exp\left(\frac{-\lambda R}{H}\right), \frac{H\zeta_*^2}{\lambda^2 R^2}\right\} + \frac{\sigma^2}{\lambda MKR} + \min\left\{\Delta, \frac{H\sigma^2}{\lambda^2 K^2 R^2}\right\}$ |
| Algorithm-Independent<br>Lower Bound Theorem 2 | $\min\left\{\frac{\Delta\sqrt{\lambda}}{\sqrt{H}}, \frac{\lambda\zeta_*^2}{H^2}\right\}\exp\left(-\frac{\sqrt{\lambda}R}{\sqrt{H}}\right) + \frac{\sigma^2}{\lambda MKR}$ |

Table 2: Guarantees under the strongly convex assumptions, with log factors omitted. See (12) for a definition and discussion of $\bar{\zeta}$.

## A  Discussion of Local SGD lower bound in strongly convex setting

In the strongly convex case, the lower bound from Theorem 2 nearly matches the upper bound of Koloskova et al.. Focusing on the case $H = B = \sigma = 1$ in order to emphasize the role of $\zeta_*$, the only differences are between a term $1/(KR^2)$ versus $1/(K^2R^2)$—a gap which also exists in the homogeneous case [23]—and between $\exp(-\lambda R) + \zeta_*^2/(\lambda^2 R^2)$ and $\min\{\exp(-\lambda R), \zeta_*^2/(\lambda^2 R^2)\}$. The latter gap is somewhat more substantial than the convex case, but nevertheless indicates that the $\zeta_*^2/(\lambda^2 R^2)$ rate cannot be improved until the number of rounds of communication is at least the condition number (or $\zeta_*^2$ is very large).

## B  Discussion of Karimireddy et al. [11]

We compare our results to Karimireddy et al. [11], who presented an analysis of the inner/outer stepsize variant of Section 6 (as FEDAVG, with a different stepsize parametrization—see below) as well as the novel method SCAFFOLD which incorporates variance reduction.

Karimireddy et al. [11, Theorem V] show that for the inner/outer stepsize updates (13), with optimal choice of stepsizes, in the weakly convex case

$$\mathbb{E}F(\hat{x}) - F^* \leq O\left(\frac{HB^2}{R} + \frac{\sigma B}{\sqrt{SKR}} + \frac{(H\zeta_*^2 B^4)^{1/3}}{R^{2/3}} + \sqrt{1 - \frac{S}{M}} \cdot \frac{\zeta_* B}{\sqrt{SR}}\right) \qquad (16)$$

and in the strongly convex case (for $R \geq \Omega\left(\frac{H}{\lambda}\right)$)

$$\mathbb{E}F(\hat{x}) - F^* \leq O\left(\lambda B^2 \exp\left(\frac{-\lambda R}{H}\right) + \frac{\sigma^2}{\lambda SKR} + \frac{H\zeta_*^2}{\lambda^2 R^2} + \left(1 - \frac{S}{M}\right) \cdot \frac{\zeta_*^2}{\lambda SR}\right). \qquad (17)$$

But these are loose upper bounds: as discussed in Section 6, the Minibatch SGD guarantees also apply to the inner/outer variant (by using $\eta_{\text{inner}} = 0$). The Minibatch SGD guarantees (14) and (15) can therefor be viewed also as guarantees on the inner/outer variant (i.e. Karimireddy et al.'s FEDAVG) that improve over (16) and (17) in several ways: (a) they avoid the the third terms in (16) and (17); (b) they improve the fourth terms by a factor of $\sqrt{S}$ and $S$ respectively; and (c) they avoid the requirement $R > H/\lambda$.

Karimireddy et al.'s presentation actually uses a different step-size parametrization that does not allow for $\eta_{\text{inner}} = 0$: they use $\eta_l = \eta_{\text{inner}}$ and $\eta_g = \eta_{\text{outer}}/\eta_{\text{inner}}$. We prefer the presentation using

$\eta_{\text{inner}}$ and $\eta_{\text{outer}}$ in order to emphasize the relationship with Minibatch SGD and in order to explicitly allow $\eta_{\text{inner}} = 0$. But in any case, even using their parametrization, $\eta_l = \eta_{\text{inner}}$ could be taken arbitrarily close to zero making the deviation from Minibatch SGD arbitrarily small. Indeed, the Karimireddy et al.'s bounds (16) and (17) are obtained when $\eta_{\text{inner}}$ is already so small that the algorithm is essentially equivalent to Minibatch SGD.

But Karimireddy et al.'s main contribution was the presentation of SCAFFOLD, which incorporates machine-specific control iterates that reduce the inter-machine variances. For SCAFFOLD, [11, Theorem VII] show that in the weakly convex case[7]:

$$\mathbb{E}F(\hat{x}) - F^* \leq O\left(\frac{HB^2}{R} + \frac{\sigma B}{\sqrt{SKR}} + \frac{M\zeta_*^2}{HSR} + \frac{\sigma\zeta_*\sqrt{M}}{HS\sqrt{KR}}\right), \tag{18}$$

and in the strongly convex case (for $R \geq \max\left\{\frac{H}{\lambda}, \frac{M}{S}\right\}$)

$$\mathbb{E}F(\hat{x}) - F^* \leq O\left(\lambda\left(B^2 + \frac{M\zeta_*^2}{SH^2}\right)\exp\left(-\min\left\{\frac{\lambda}{H}, \frac{S}{M}\right\}R\right) + \frac{\sigma^2}{\lambda SKR}\right). \tag{19}$$

These guarantees are also obtained when $\eta_{\text{inner}}$ is so close to zero that this is essentially a minibatch method, in this case "Minibatch SAGA" [cf. 7].

Although SCAFFOLD is aimed specifically at the setting where a subset of machines are used in each round (i.e. $S < M$), let us first check whether it provides benefits in our "standard" setting (introduced in Sections 2), where all machines are used each round (i.e. $S = M$). In this case, the SCAFFOLD bounds (18) and (19) may improve over the loose upper bounds (16) and (17), but this is only due to the looseness in these upper bounds. The SCAFFOLD upper bounds (for $S = M$) do not actually improve over the Minibatch SGD guarantees of Theorem 1, and only include additional terms (see Tables 1 and 2). That is, the SCAFFOLD upper bounds, when $S = M$, are valid also for FEDAVG (since as discussed above, Minibatch SGD gurantees are valid also for FEDAVG) and so do not show a benefit in the setting where all machines are used each round. This is perhaps not surprising, since in this setting there is no need to reduce inter-machine variance, and so no benefit from variance reduction.

Let us turn then to the setting of Section 7, where only a random subset of $S < M$ machines are used in each iteration, and for which SCAFFOLD was developed. In this setting, SCAFFOLD does show a benefit in some regimes. Let us focus on the weakly convex case and compare the SCAFFOLD guarantee (18) with the Minibatch SGD guarantee (14). We can verify that, e.g. when $\sigma = 0$ and $\zeta_* = HB$, SCAFFOLD improves over Minibatch SGD if $\frac{M}{R} \ll \frac{S}{M} \ll \frac{R}{M}$, and $S < M$, but the SCAFFOLD guarantee (18) is worse then Minibatch SGD if $\frac{S}{M} \ll \frac{M}{R}$. More generally, the SCAFFOLD guarantee is worse than Minibatch SGD if $\sigma = 0$ and $\frac{S}{M} \ll \frac{\zeta_*^2}{H^2B^2}\min(1, \frac{M}{R})$. Also for strongly convex objectives, SCAFFOLD improves over Minibatch SGD in some regimes but the guarantee (19) is worse than Minibatch SGD in other regimes. Care is required to map out the precise regimes and how they depend on the various problem parameters.

## C  Proof of Theorem 1

In this Appendix, we prove Theorem 1, starting with an analysis of Minibatch SGD, and proceeding to analyze Accelerated Minibatch SGD.

### C.1  Minibatch SGD for heterogeneous objectives

For the proof, we will use the following standard property of convex functions:

**Lemma 1** (Co-Coercivity of the Gradient). *For any $H$-smooth and convex $F$, and any $x$, and $y$,*

$$\|\nabla F(x) - \nabla F(y)\|^2 \leq H\langle \nabla F(x) - \nabla F(y),\, x - y\rangle,$$

*and*

$$\|\nabla F(x) - \nabla F(y)\|^2 \leq 2H(F(x) - F(y) - \langle\nabla F(y),\, x - y\rangle).$$

Also note the following result from Stich [21], which is useful for optimizing the step-sizes.

**Lemma 2** (Stich [21], Lemma 3). *Consider non-negative sequences$\{r_t\}_{t\geq 0}$ and $\{s_t\}_{t\geq 0}$, which satisfy:*

$$r_{t+1} \leq (1 - a\eta_t)r_t - b\eta_t s_t + c\eta_t^2, \tag{20}$$

*for non-negative step-sizes $\eta_t \leq \frac{1}{d}, \forall t$, for a parameter $d \geq a$, $d > 0$. Then there exist a choice of step-sizes $\eta_t$ and averaging weights $w_t$, such that:*

$$\frac{b}{W_T} \sum_{t=0}^{T} sw_t + ar_{T+1} \leq 32dr_0 \exp\left[-\frac{aT}{2d}\right] + \frac{36c}{aT}, \tag{21}$$

*for $W_T := \sum_{t=0}^{T} w_t$.*

Finally we can prove the following result for Minibatch SGD in this setting.

**Theorem 5.** *Under the convex assumptions, the average of the iterates of Minibatch SGD guarantees for a universal constant c,*

$$\mathbb{E}F(\hat{x}) - F^* \leq c \cdot \frac{HB^2}{R} + c \cdot \frac{\sigma_* B}{\sqrt{MKR}}.$$

*Under the strongly convex assumptions, a weighted average of its iterates guarantees*

$$\mathbb{E}F(\hat{x}) - F^* \leq c \cdot \frac{H\Delta}{\lambda} \exp\left[-\frac{\lambda R}{8H}\right] + c \cdot \frac{\sigma_*^2}{\lambda MKR}.$$

*Proof.* By the $\lambda$-strong convexity of $F$, where $\lambda$ might be equal to zero:

$$\mathbb{E}\|x_{t+1} - x^*\|^2 = \mathbb{E}\left\|x_t - \eta_t \frac{1}{KM} \sum_{m=1}^{M}\sum_{k=1}^{K} \nabla f(x_t; z_t^{m,k}) - x^*\right\|^2, \tag{22}$$

$$= \mathbb{E}\|x_t - x^*\|^2 - 2\eta_t \mathbb{E}\langle \nabla F(x_t),\, x_t - x^*\rangle$$

$$+ \eta_t^2 \mathbb{E}\left\|\frac{1}{KM} \sum_{m=1}^{M}\sum_{k=1}^{K} \nabla f(x_t; z_t^{m,k})\right\|^2, \tag{23}$$

$$\leq (1 - \lambda\eta_t)\mathbb{E}\|x_t - x^*\|^2 - 2\eta_t \mathbb{E}[F(x_t) - F^*]$$

$$+ \eta_t^2 \mathbb{E}\left\|\frac{1}{KM} \sum_{m=1}^{M}\sum_{k=1}^{K} \nabla f(x_t; z_t^{m,k})\right\|^2. \tag{24}$$

By the $H$-smoothness of $f(\cdot\,; z)$, we can bound the final term with

$$\mathbb{E}\left\|\frac{1}{KM}\sum_{m=1}^{M}\sum_{k=1}^{K}\nabla f(x_t; z_t^{m,k})\right\|^2$$

$$= \mathbb{E}\left\|\frac{1}{KM}\sum_{m=1}^{M}\sum_{k=1}^{K}\left[\nabla f(x_t; z_t^{m,k}) - \nabla f(x^*; z_t^{m,k}) + \nabla f(x^*; z_t^{m,k})\right]\right\|^2, \tag{25}$$

$$\leq 2\mathbb{E}\left\|\frac{1}{KM}\sum_{m=1}^{M}\sum_{k=1}^{K}\left[\nabla f(x_t; z_t^{m,k}) - \nabla f(x^*; z_t^{m,k})\right]\right\|^2$$

$$+ 2\mathbb{E}\left\|\frac{1}{KM}\sum_{m=1}^{M}\sum_{k=1}^{K}\nabla f(x^*; z_t^{m,k})\right\|^2, \tag{26}$$

$$\leq \frac{2}{KM}\sum_{m=1}^{M}\sum_{k=1}^{K}\mathbb{E}\left\|\nabla f(x_t; z_t^{m,k}) - \nabla f(x^*; z_t^{m,k})\right\|^2$$

$$+ 2\mathbb{E}\left\|\frac{1}{KM}\sum_{m=1}^{M}\sum_{k=1}^{K}\nabla f(x^*; z_t^{m,k}) - \nabla F_m(x^*)\right\|^2, \tag{27}$$

$$\leq \frac{4H}{KM}\sum_{m=1}^{M}\sum_{k=1}^{K}\mathbb{E}\left[f(x_t; z_t^{m,k}) - f(x^*; z_t^{m,k}) - \left\langle\nabla f(x^*; z_t^{m,k}), x_t - x^*\right\rangle\right] + \frac{2\sigma_*^2}{MK}, \tag{28}$$

$$= 4H\mathbb{E}[F(x_t) - F^*] + \frac{2\sigma_*^2}{MK}. \tag{29}$$

Where, for the third inequality we used Lemma 1. Plugging this back into (24), then for $\eta_t \leq \frac{1}{4H}$,

$$\mathbb{E}\|x_{t+1} - x^*\|^2 \leq (1 - \lambda\eta_t)\mathbb{E}\|x_t - x^*\|^2 - 2\eta_t(1 - 2H\eta_t)\mathbb{E}[F(x_t) - F^*] + \frac{2\eta_t^2\sigma_*^2}{MK}, \tag{30}$$

$$\leq (1 - \lambda\eta_t)\mathbb{E}\|x_t - x^*\|^2 - \eta_t\mathbb{E}[F(x_t) - F^*] + \frac{2\eta_t^2\sigma_*^2}{MK}, \tag{31}$$

$$\mathbb{E}[F(x_t) - F^*] \leq \left(\frac{1}{\eta_t} - \lambda\right)\mathbb{E}\|x_t - x^*\|^2 - \frac{1}{\eta_t}\mathbb{E}\|x_{t+1} - x^*\|^2 + \frac{2\eta_t\sigma_*^2}{MK}. \tag{32}$$

Now we look at $\lambda = 0$ and $\lambda > 0$ separately.

**Convex case ($\lambda = 0$):**   Choose a constant step-size,

$$\eta_t = \eta = \min\left\{\frac{1}{4H}, \frac{B\sqrt{MK}}{\sigma_*\sqrt{T}}\right\}. \tag{33}$$

Then the averaged iterate $\bar{x}_R = \frac{1}{R}\sum_{t=1}^{R} x_t$ satisfies:

$$\mathbb{E}F(\bar{x}_R) - F^* \leq \frac{1}{R}\sum_{t=1}^{R}\mathbb{E}F(x_t) - F^*, \tag{34}$$

$$\leq \frac{1}{R}\sum_{t=1}^{R}\frac{1}{\eta}\mathbb{E}\|x_t - x^*\|^2 - \frac{1}{\eta}\mathbb{E}\|x_{t+1} - x^*\|^2 + \frac{2\eta\sigma_*^2}{MK}, \tag{35}$$

$$\leq \frac{\|x_0 - x^*\|^2}{\eta R} + \frac{2\eta\sigma_*^2}{MK}, \tag{36}$$

$$\leq \max\left\{\frac{4HB^2}{R}, \frac{\sigma_* B}{\sqrt{MKR}}\right\} + \frac{2\sigma_* B}{\sqrt{MKR}}, \tag{37}$$

$$\leq \frac{4HB^2}{R} + \frac{3\sigma_* B}{\sqrt{MKR}}. \tag{38}$$

**Strongly convex case ($\lambda > 0$):**    Rewriting (31),

$$\mathbb{E}\|x_{t+1} - x^*\|^2 \leq (1 - \lambda\eta_t)\mathbb{E}\|x_t - x^*\|^2 - \eta_t\mathbb{E}[F(x_t) - F^*] + \frac{2\eta_t^2\sigma_*^2}{MK}, \qquad (39)$$

we note that it satisfies the conditions for Lemma 2 for the specific assignment:

$$r_t = \mathbb{E}\|x_t - x^*\|^2, \ s_t = \mathbb{E}[F(x_t) - F^*], \qquad (40)$$

$$a = \lambda, \ b = 1, \ c = \frac{2\sigma_*^2}{MK}, \ d = 4H, \ T = R. \qquad (41)$$

Thus using Lemma 2, and applying Jensen's inequality we can guarantee the following convergence rate for the averaged iterate $\hat{x}_R = \frac{1}{W_R}\sum_{t=1}^{R} w_t x_t$,

$$\mathbb{E}[F(\hat{x}_R) - F^*] \leq 128H\|x_0 - x^*\|^2 \exp\left[-\frac{\lambda R}{8H}\right] + \frac{72\sigma_*^2}{\lambda MKR}, \qquad (42)$$

using step-size $\eta_t$ and $w_t$ given by,

$$if \ R \leq \frac{4H}{\lambda}, \qquad \eta_t = \frac{1}{4H}, \qquad w_t = (1 - \lambda\eta_t)^{-(t+1)},$$

$$if \ R > \frac{4H}{\lambda} \ and \ t < t_0, \qquad \eta_t = \frac{1}{4H}, \qquad w_t = 0,$$

$$if \ R > \frac{4H}{\lambda} \ and \ t \geq t_0, \qquad \eta_t = \frac{2}{\lambda(\kappa + t - t_0)}, \qquad w_t = (\kappa + t - t_0)^2,$$

where $\kappa = \frac{8H}{\lambda}$ and $t_0 = \lceil\frac{R}{2}\rceil$. We conclude by observing that $HB^2 \leq \frac{H\Delta}{\lambda}$. □

## C.2    Accelerated Minibatch SGD for heterogeneous objectives

We first recall some classical results from Ghadimi and Lan [8, 9] for accelerated variants of minibatch SGD. These results are for minimizing $F(x) := \mathbb{E}_{z\sim\mathcal{D}}f(x, z)$ where $F$ is $H$-smooth and $\lambda(\geq 0)$-strongly convex. The algorithms use unbiased stochastic gradients $\{g_t\}_{t\in[T]}$, i.e., for all $t$, $\mathbb{E}[g_t(x)] = \nabla F(x)$ which have bounded variance for all $x$, i.e., $\mathbb{E}\|g_t(x) - \nabla F(x)\|^2 \leq \sigma^2$.

First consider the AC-SA algorithm Ghadimi and Lan [c.f., Sec 3.1, 8]), with step-size parameters $\{\alpha_t\}_{t\geq 1}$ and $\{\gamma_t\}_{t\geq 1}$ s.t. $\alpha_1 = 1, \alpha_t \in (0, 1)$ for any $t \geq 2$ and $\gamma_t > 0$ for any $t \geq 1$. The algorithm maintains three intertwined sequences $\{x_t\}$, $\{x_t^{ag}\}$, and $\{x_t^{md}\}$, updated as follows:

1. Set the initial points $x_0^{ag} = x_0 \in X$ and $t = 1$;

2. Set $x_t^{md} = \frac{(1-\alpha_t)(\lambda+\gamma_t)}{\gamma_t + (1-\alpha_t^2)\lambda}x_{t-1}^{ag} + \frac{\alpha_t[(1-\alpha_t)\mu + \gamma_t]}{\gamma_t + (1-\alpha_t^2)\lambda}x_{t-1}$;

3. Call the stochastic oracle to get the gradient $g_t$ at the point $x_t^{md}$;

4. Set $x_t = \arg\min_{x\in X}\left\{\alpha_t[\langle g_t, x\rangle + \frac{\lambda}{2}\|x_t^{md} - x\|^2] + [(1-\alpha_t)\frac{\lambda}{2} + \frac{\gamma_t}{2}]\|x_{t-1} - x\|^2\right\}$;

5. Set $x_t^{ag} = \alpha_t x_t + (1 - \alpha_t)x_{t-1}^{ag}$;

6. Set $t \leftarrow t + 1$ and go to step 1.

We have the following (almost optimal) convergence rate for strongly convex functions using AC-SA (see, Sec 3.1 in [8]).

**Lemma 3.** *(Ghadimi and Lan [8], Proposition 9) Let $\hat{x}^{ag}$ be computed by $T$ steps of AC-SA using stochastic gradients of variance $\sigma^2$, then for a universal constant c,*

$$\mathbb{E}\left[F(x^{ag}) - F(x^\star)\right] \leq c \cdot \frac{H\|x_0 - x^\star\|^2}{T^2} + c \cdot \frac{\sigma^2}{\lambda T}.$$

It can be adapted to the weakly convex case by noting that, if $\tilde{F}(x) := F(x) + \frac{\lambda}{2}\|x_0 - x\|^2$ for any $\lambda, x_0$, and $x^\star = \arg\min F(x)$ then,

$$\min_y \mathbb{E}\left[\tilde{F}(y)\right] \leq \mathbb{E}\left[F(x^\star) + \frac{\lambda}{2}\|x_0 - x^\star\|^2\right],$$

$$\Rightarrow -\mathbb{E}\left[F(x^\star)\right] \leq -\mathbb{E}\left[\min_y \tilde{F}(y)\right] + \frac{\lambda}{2}\|x_0 - x^\star\|^2,$$

$$\Rightarrow \mathbb{E}\left[F(x) - F(x^\star)\right] \leq \mathbb{E}\left[\tilde{F}(x) - \min_y \tilde{F}(y)\right] + \frac{\lambda}{2}\|x_0 - x^\star\|^2, \forall x.$$

This also holds if we optimize the right hand side w.r.t. $\lambda$. In other words, a guarantee for the strongly convex case, can be converted to the weakly convex case, by regularizing with $\frac{\lambda}{2}\|x_0 - x^\star\|^2$ with optimal value of $\lambda$. This gives the following result,

**Lemma 4.** *Let $\hat{x}^{ag}$ be computed by $T$ steps of AC-SA on the regularized objective $\tilde{F}(x) = F(x) + \frac{\sigma}{2\|x_0 - x^*\|\sqrt{T}}\|x - x_0\|^2$, where the stochastic gradients have variance $\sigma^2$, then for a constant $c$*

$$\mathbb{E}\left[F(\hat{x}^{ag}) - F(x^\star)\right] \leq c \cdot \frac{H\|x_0 - x^\star\|^2}{T^2} + c \cdot \frac{\sigma\|x_0 - x^\star\|}{\sqrt{T}}.$$

This is minimax optimal for weakly convex functions. Next we consider the multi-stage accelerated SGD algorithm Ghadimi and Lan [c.f., Sec 3, 9] which uses the above AC-SA algorithm. Let $p_0 \in X$, have bounded sub-optimality $F(p_0) - F(x^\star) \leq \Delta$, then for $k = 1, 2, \ldots$,

1. Run $N_k$ iterations of the generic AC-SA by using $x_0 = p_{k-1}$, $\{\alpha_t\}_{t \geq 1}$, and $\{\gamma_t\}_{t \geq 1}$, with relevant definitions as follows,

$$N_k = \left\lceil \max\left\{4\sqrt{\frac{2H}{\lambda}}, \frac{128\sigma^2}{3\lambda\Delta 2^{-(k+1)}}\right\} \right\rceil,$$

$$\alpha_t = \frac{2}{t+1}, \gamma_t = \frac{4\phi_k}{t(t+1)},$$

$$\phi_k = \max\left\{2H, \left[\frac{\lambda\sigma^2}{3\Delta 2^{-(k-1)}N_k(N_k+1)(N_k+2)}\right]^{1/2}\right\};$$

2. Set $p_k = x_{N_k}^{ag}$ where $x_{N_k}^{ag}$ is the solution obtained in the previous step.

We have following optimal rate for strongly convex functions for this algorithm,

**Lemma 5.** *(Ghadimi and Lan [9], Proposition 7) Let $\hat{x}^{ag}$ be computed by $T$ steps of multi-stage AC-SA using stochastic gradients of variance $\sigma^2$, then for a universal constant $c$,*

$$\mathbb{E}\left[F(\hat{x}^{ag}) - F(x^\star)\right] \leq c \cdot \Delta \exp\left[-\sqrt{\frac{\lambda}{H}}T\right] + c \cdot \frac{\sigma^2}{\lambda T}.$$

**Theorem 6.** *Under the convex assumptions, performing AC-SA on the regularized objective $\tilde{F}(x) = F(x) + \frac{\sigma}{2B\sqrt{MKR}}\|x\|^2$ guarantees for a universal constant $c$*

$$\mathbb{E}F(\hat{x}) - F^* \leq c \cdot \frac{HB^2}{R^2} + c \cdot \frac{\sigma B}{\sqrt{MKR}}.$$

*Under the strongly convex assumptions, the multi-stage AC-SA algorithm guarantees*

$$\mathbb{E}F(\hat{x}) - F^* \leq c \cdot \Delta \exp\left[-\sqrt{\frac{\lambda}{H}}R\right] + c \cdot \frac{\sigma^2}{\lambda MKR}.$$

*Proof.* In order to use the previous lemmas, first note that the stochastic gradient at time $t$ at point $x$ is given by $\frac{1}{MK}\sum_{m=1}^{M}\sum_{k=1}^{K}\nabla f(x; z_{m,k}^t)$, where $z_{m,k}^t \sim^{i.i.d.} \mathcal{D}^m$ for all machines. Fortunately,

its still an unbiased gradient estimate, i.e., $\mathbb{E}\left[\frac{1}{MK}\sum_{m=1}^{M}\sum_{k=1}^{K}\nabla f(x; z_{m,k}^t)\right] = \nabla F(x)$ since the iterates on each machine are sampled i.i.d. Its variance is given by,

$$\mathbb{E}\left\|\frac{1}{MK}\sum_{m=1}^{M}\sum_{k=1}^{K}\nabla f(x; z_{m,k}^t) - \nabla F(x)\right\|^2 \tag{43}$$

$$= \frac{1}{M^2 K^2}\sum_{m=1}^{M}\sum_{k=1}^{K}\mathbb{E}\left\|\nabla f(x; z_{m,k}^t) - \nabla F_m(x)\right\|^2 \tag{44}$$

$$\leq \frac{1}{M^2 K^2}\sum_{m=1}^{M}\sum_{k=1}^{K}\sigma^2 \tag{45}$$

$$= \frac{\sigma^2}{MK} \tag{46}$$

Plugging this into Lemmas 4 and 5 completes the proof. $\qquad\square$

## D Proof of Theorem 2

Consider the following function $F : \mathbb{R}^4 \to \mathbb{R}$:

$$F(x) = \frac{1}{2}(F_1(x) + F_2(x)) = \frac{1}{2}\left(\mathbb{E}_{z^1 \sim \mathcal{D}^1} f(x; z^1) + \mathbb{E}_{z^2 \sim \mathcal{D}^2} f(x; z^2)\right) \tag{47}$$

The distribution $z^1 \sim \mathcal{D}^1$ is described by $z^1 = (1, z)$ for $z \sim \mathcal{N}(0, \sigma^2)$. Similarly, $z^2 \sim \mathcal{D}^2$ is specified by $z^2 = (2, z)$ for $z \sim \mathcal{N}(0, \sigma^2)$. The lower bound construction will be based on just two functions. For $M > 2$ machines, we simply assign the first $\lfloor M/2 \rfloor$ machines $F_1$ and the next $\lfloor M/2 \rfloor$ machines $F_2$. This diminishes the lower bound by at most a $(M - 1)/M$ factor. Therefore, we continue with the case $M = 2$.

Following Woodworth et al. [23], we define the local functions $F_1$ and $F_2$ via the auxiliary function

$$g(x_1, x_2, x_3, z) = \frac{\mu}{2}(x_1 - c)^2 + \frac{H}{2}\left(x_2 - \frac{\sqrt{\mu}c}{\sqrt{H}}\right)^2 + \frac{H}{8}\left(x_3^2 + [x_3]_+^2\right) + z^\top x_3 \tag{48}$$

$$G(x_1, x_2, x_3) = \mathbb{E}_z g(x_1, x_2, x_3, z) \tag{49}$$

where $c > 0$ and $\mu \in \left[\lambda, \frac{H}{16}\right]$ are parameters to be determined later, and where $[x]_+ := \max\{x, 0\}$. Then, we define

$$f(x; (1, z)) = g(x_1, x_2, x_3, z) + \frac{Lx_4^2}{2} + \zeta_* x_4 \tag{50}$$

$$f(x; (2, z)) = g(x_1, x_2, x_3, z) + \frac{\lambda x_4^2}{2} - \zeta_* x_4 \tag{51}$$

for a parameter $L \in [\lambda, H]$ to be determined later. Therefore,

$$F_1(x) = \mathbb{E}_{z^1 \sim \mathcal{D}^1} f(x; z^1) = G(x_1, x_2, x_3) + \frac{Lx_4^2}{2} + \zeta_* x_4 \tag{52}$$

$$F_2(x) = \mathbb{E}_{z^2 \sim \mathcal{D}^2} f(x; z^2) = G(x_1, x_2, x_3) + \frac{\lambda x_4^2}{2} - \zeta_* x_4 \tag{53}$$

It is clear from inspection that both $F_1$ and $F_2$, and consequently $F$, are $H$-smooth and $\lambda$-strongly convex. Furthermore, the variance of the gradients is bounded by $\sigma^2$ for both $\mathcal{D}^1$ and $\mathcal{D}^2$.

It is clear that $G$ attains its minimum of zero at $\left[c, \frac{\sqrt{\mu}c}{\sqrt{H}}, 0\right]$ so $\nabla G\left(c, \frac{\sqrt{\mu}c}{\sqrt{H}}, 0\right) = 0$, and thus

$$\nabla F\left(c, \frac{\sqrt{\mu}c}{\sqrt{H}}, 0, 0\right) = \nabla G\left(c, \frac{\sqrt{\mu}c}{\sqrt{H}}, 0\right) + \left(\frac{\zeta_*}{2} - \frac{\zeta_*}{2}\right)e_4 = 0 \tag{54}$$

From now on, we use $x^* = \left[c, \frac{\sqrt{\mu}c}{\sqrt{H}}, 0, 0\right]$ to denote the minimizer of $F$, which has norm

$$\|x^*\|^2 = \left(1 + \frac{\mu}{H}\right)c^2 \leq 2c^2 \tag{55}$$

We can therefore ensure $\|x^*\|^2 \leq B^2$ by choosing $c^2 \leq \frac{B^2}{2}$. Furthermore, the initial suboptimality

$$F(0,0,0,0) - F^* = \mu c^2 \tag{56}$$

Therefore, we can ensure $F(0,0,0,0) - F^* \leq \Delta$ by choosing $c^2 \leq \frac{\Delta}{\mu}$. We conclude by showing that for this objective, $\zeta_*^2$ bounded by

$$\frac{1}{2}\sum_{m=1}^{2}\|\nabla F_m(x^*)\|^2 = \|\nabla F_2(x^*)\|^2 = \|\nabla F_1(x^*)\|^2 = \zeta_*^2 \tag{57}$$

Therefore, this objective has the desired level of heterogeneity.

Therefore, we have shown that the objective satisfies all of the necessary conditions for the lower bound. All that remains is to lower bound the error of Local SGD with a constant stepsize $\eta$ applied to this function.

**Lemma 6.** *For $\mu \leq 2L$, then Local SGD with any constant stepsize $\eta \leq \frac{1}{L}$ applied to $F_1$ and $F_2$ after being initialized at zero results in $\hat{x}_4$ such that*

$$\frac{(L+\mu)\hat{x}_4^2}{4} \geq \frac{\zeta_*^2(L+\mu)}{16\mu^2}\left(\frac{L-\mu}{L} - (1-\mu\eta)^K\right)^2 \mathbb{1}_{\left\{\eta \leq \frac{1}{L}\right\}}\mathbb{1}_{\left\{(1-\mu\eta)^K \leq \frac{L-\mu}{L}\right\}}$$

*Proof.* Since the coordinates of $F_1$ and $F_2$ are completely decoupled, the behavior of the fourth coordinate of the iterates can be analyzed separately from the others.

Let $x_{k,r}^{(1)}$ denote the fourth coordinate of machine 1's iterate at the $k$th iteration of round $r$, and similarly for $x_{k,r}^{(2)}$. The local SGD dynamics give

$$x_{k+1,r}^{(1)} = x_{k,r}^{(1)} - \eta\left(Lx_{k,r}^{(1)} + \zeta_*\right) = -\frac{\zeta_*}{L} + (1-L\eta)\left(x_{k,r}^{(1)} + \frac{\zeta_*}{L}\right) \tag{58}$$

$$x_{k,r}^{(2)} = x_{k,r}^{(2)} - \eta\left(-\zeta_* + \mu x_{k,r}^{(2)}\right) = \frac{\zeta_*}{\mu} + (1-\mu\eta)\left(x_{k,r}^{(2)} - \frac{\zeta_*}{\mu}\right) \tag{59}$$

and $\hat{x}_4 = \frac{1}{2}\left(x_{K,R}^{(1)} + x_{K,R}^{(2)}\right) = x_{0,R+1}$. Unravelling this recursion, we have that

$$x_{0,r+1} = x_{0,r+1}^{(1)} = x_{0,r+1}^{(2)} = \frac{1}{2}\left(\frac{\zeta_*}{\mu} - \frac{\zeta_*}{L} + (1-\mu\eta)^K\left(x_{0,r} - \frac{\zeta_*}{\mu}\right) + (1-L\eta)^K\left(x_{0,r} + \frac{\zeta_*}{L}\right)\right) \tag{60}$$

Furthermore, if $\eta \leq \frac{1}{L}$ then $(1-L\eta) \geq 0$, so if $x_{0,r} \geq 0$ then

$$x_{0,r+1} \geq \frac{\zeta_*}{2\mu} - \frac{\zeta_*}{2L} + (1-\mu\eta)^K\left(\frac{x_{0,r}}{2} - \frac{\zeta_*}{2\mu}\right) \geq \frac{\zeta_*}{2\mu}\left(\frac{L-\mu}{L} - (1-\mu\eta)^K\right) \tag{61}$$

Finally, since $x_{0,0} = 0 \geq 0$, the condition $x_{0,r} \geq 0$ will hold throughout optimization, so

$$\hat{x}_4 \geq \frac{\zeta_*}{2\mu}\left(\frac{L-\mu}{L} - (1-\mu\eta)^K\right) \tag{62}$$

Therefore, if $\eta \leq \frac{1}{L}$ and $(1-\mu\eta)^K \leq \frac{L-\mu}{L}$ then

$$\frac{(L+\mu)\hat{x}_4^2}{4} \geq \frac{\zeta_*^2(L+\mu)}{16\mu^2}\left(\frac{L-\mu}{L} - (1-\mu\eta)^K\right)^2 \tag{63}$$

This completes the proof. □

We now prove the theorem:

**Theorem 2.** *For any $M$, $K$, and $R$ there exist objectives in four dimensions such that Local SGD initialized at zero and using any fixed stepsize $\eta$ will have suboptimality at least*

$$\mathbb{E}F(\hat{x}) - F^* \geq c \cdot \left(\min\left\{\frac{HB^2}{R}, \frac{(H\zeta_*^2 B^4)^{1/3}}{R^{2/3}}\right\} + \frac{(H\sigma^2 B^4)^{1/3}}{K^{2/3}R^{2/3}} + \frac{\sigma B}{\sqrt{MKR}}\right)$$

$$\mathbb{E}F(\hat{x}) - F^* \geq c \cdot \left(\min\left\{\Delta\exp\left(-\frac{6\lambda R}{H}\right), \frac{H\zeta_*^2}{\lambda^2 R^2}\right\} + \min\left\{\Delta, \frac{H\sigma^2}{\lambda^2 K^2 R^2}\right\} + \frac{\sigma^2}{\lambda MKR}\right)$$

*under the convex and strongly convex assumptions (for $H \geq 16\lambda$), respectively.*

*Proof.* Since the four different coordinates are completely decoupled from each other, it suffices to analyze each coordinate separately.

In the course of proving [Theorem 3 23], Woodworth et al. prove that

$$\mathbb{E}G(\hat{x}_1, \hat{x}_2, \hat{x}_3) - G\left(c, \frac{\sqrt{\mu}c}{\sqrt{H}}, 0\right)$$

$$\geq \frac{\mu c^2(1-\mu\eta)^{KR}}{2} + \frac{\mu c^2}{2}\mathbb{1}_{\{\eta > \frac{2}{H}\}} + \frac{H\eta^2\sigma^2}{18432}\mathbb{1}_{\{\eta \leq \frac{2}{H}\}}\mathbb{1}_{\{\eta \geq \frac{8}{HKR}\}} \quad (64)$$

Furthermore, by Lemma 6

$$\frac{(L+\lambda)\hat{x}_4^2}{4} \geq \frac{\zeta_*^2(L+\mu)}{16\mu^2}\left(\frac{L-\mu}{L} - (1-\mu\eta)^K\right)^2 \mathbb{1}_{\{\eta \leq \frac{1}{L}\}}\mathbb{1}_{\{(1-\mu\eta)^K \leq \frac{L-\mu}{L}\}} \quad (65)$$

Therefore, choosing $L = \frac{H}{2}$

$$\mathbb{E}F(\hat{x}) - F^* = \mathbb{E}G(\hat{x}_1, \hat{x}_2, \hat{x}_3) - G\left(c, \frac{\sqrt{\mu}c}{\sqrt{H}}, 0\right) + \frac{H+2\lambda}{8}\hat{x}_4^2 \quad (66)$$

$$\geq \frac{\mu c^2(1-\mu\eta)^{KR}}{2} + \frac{\mu c^2}{2}\mathbb{1}_{\{\eta > \frac{2}{H}\}} + \frac{H\eta^2\sigma^2}{18432}\mathbb{1}_{\{\eta \leq \frac{2}{H}\}}\mathbb{1}_{\{\eta \geq \frac{8}{HKR}\}}$$

$$+ \frac{\zeta_*^2(H+2\mu)}{32\mu^2}\left(\frac{H-2\mu}{H} - (1-\mu\eta)^K\right)^2 \mathbb{1}_{\{\eta \leq \frac{2}{H}\}}\mathbb{1}_{\{(1-\mu\eta)^K \leq \frac{H-2\mu}{H}\}} \quad (67)$$

**Stochastic terms**

First, we will show a lower bound in terms of $\sigma^2$ using solely the first three terms of (67). Consider three cases:

**Case 1** $\eta \geq \frac{2}{H}$**:**   In this case, from the second term of (67) we see that

$$\mathbb{E}F(\hat{x}) - F^* \geq \frac{\mu c^2}{2} \quad (68)$$

**Case 2** $\frac{1}{2\mu KR} \leq \eta \leq \frac{2}{H}$**:**   In this case, the third term of (67) shows

$$\mathbb{E}F(\hat{x}) - F^* \geq \frac{H\eta^2\sigma^2}{18432} \quad (69)$$

where we recalled that $\mu \leq \frac{H}{16}$, so $\eta \geq \frac{1}{2\mu KR} \geq \frac{8}{HKR}$. This is non-decreasing in $\eta$, so for any $\eta$

$$\mathbb{E}F(\hat{x}) - F^* \geq \frac{H\sigma^2}{73728\mu^2 K^2 R^2} \quad (70)$$

**Case 3** $\eta \leq \frac{2}{H}$ **and** $\eta \leq \frac{1}{2\mu KR}$**:**   In this case, from the first term of (67),

$$\mathbb{E}F(\hat{x}) - F^* \geq \frac{\mu c^2(1-\mu\eta)^{KR}}{2} \geq \frac{\mu c^2\left(1-\frac{1}{2KR}\right)^{KR}}{2} \geq \frac{\mu c^2}{4} \quad (71)$$

**Combination:**   Combining these three cases, we conclude that for any $\eta$

$$\mathbb{E}F(\hat{x}) - F^* \geq \min\left\{\frac{\mu c^2}{2}, \frac{H\sigma^2}{73728\mu^2 K^2 R^2}, \frac{\mu c^2}{4}\right\} = \min\left\{\frac{\mu c^2}{3}, \frac{H\sigma^2}{73728\mu^2 K^2 R^2}\right\} \quad (72)$$

This lower bound holds for any stepsize, and any $\mu \in \left[\lambda, \frac{H}{16}\right]$ and regardless of $\zeta_*$. In the strongly convex case, we recall that $F(0) - F(x^*) = \mu c^2$, therefore, we choose $\mu = \lambda$, and $c^2 = \frac{\Delta}{\lambda}$ so the lower bound reads (for a universal constant $\beta$)

$$\mathbb{E}F(\hat{x}) - F^* \geq \beta \cdot \min\left\{\Delta, \frac{H\sigma^2}{\lambda^2 K^2 R^2}\right\} \quad (73)$$

To conclude, it is well known that any first-order method which accesses at most $MKR$ stochastic gradients with variance $\sigma^2$ for a $\lambda$-strongly convex objective will suffer error at least $\beta \frac{\sigma^2}{\lambda MKR}$ in the worst case [17]. Therefore, the strongly convex lower bound is

$$\mathbb{E}F(\hat{x}) - F^* \geq \beta \cdot \min\left\{\Delta, \frac{H\sigma^2}{\lambda^2 K^2 R^2}\right\} + \beta \cdot \frac{\sigma^2}{\lambda MKR} \tag{74}$$

In the convex case, we recall that $\|x^*\|^2 \leq 2c^2$, so we choose $c^2 = \frac{B^2}{2}$, and set $\mu = \left(\frac{H\sigma^2}{B^2 K^2 R^2}\right)^{1/3}$ so the lower bound reads

$$\mathbb{E}F(\hat{x}) - F^* \geq \beta \cdot \frac{\left(H\sigma^2 B^4\right)}{K^{2/3} R^{2/3}} \tag{75}$$

To conclude, it is well known that any first-order method which accesses at most $MKR$ stochastic gradients with variance $\sigma^2$ for a convex objective with $\|x^*\| \leq B$ will suffer error at least $\beta \frac{\sigma B}{\sqrt{MKR}}$ in the worst case [17]. Therefore, the convex lower bound is

$$\mathbb{E}F(\hat{x}) - F^* \geq \beta \cdot \frac{\left(H\sigma^2 B^4\right)}{K^{2/3} R^{2/3}} + \beta \cdot \frac{\sigma B}{\sqrt{MKR}} \tag{76}$$

**Heterogeneity terms**

Next, we consider solely the first, second, and fourth terms of (67) in order to show a lower bound with respect to $\zeta_*$. Again, we consider three cases:

**Case 1** $\eta \geq \frac{2}{H}$: Again, in this case, from the second term of (67) we see that

$$\mathbb{E}F(\hat{x}) - F^* \geq \frac{\mu c^2}{2} \tag{77}$$

**Case 2** $\eta \leq \frac{2}{H}$ and $(1 - \mu\eta)^K > \frac{H - 2\mu}{H}$: In this case, from the first term of (67), we have

$$\mathbb{E}F(\hat{x}) - F^* \geq \frac{\mu c^2 (1 - \mu\eta)^{KR}}{2} \tag{78}$$

$$\geq \frac{\mu c^2}{2}\left(1 - \frac{2\mu}{H}\right)^R \tag{79}$$

$$\geq \frac{\mu c^2}{2}\left(\left(1 - \frac{4\mu}{H}\left(1 - \frac{1}{e}\right)\right)^{\frac{H}{4\mu}}\right)^{\frac{4\mu R}{H}} \tag{80}$$

$$\geq \frac{\mu c^2}{2}\exp\left(-\frac{4\mu R}{H}\right) \tag{81}$$

**Case 3** $\eta \leq \frac{2}{H}$ and $(1 - \mu\eta)^K \leq \frac{H - 2\mu}{H}$: In this case, from the first and fourth terms of (67), we have

$$\mathbb{E}F(\hat{x}) - F^* \geq \frac{\mu c^2}{2}(1 - \mu\eta)^{KR} + \frac{\zeta_*^2(H + 2\mu)}{32\mu^2}\left(\frac{H - 2\mu}{H} - (1 - \mu\eta)^K\right)^2 \tag{82}$$

Suppose that $(1 - \mu\eta)^K \geq \frac{H - 2\mu}{H} - \frac{1}{4R}$, then

$$\frac{\mu c^2}{2}(1 - \mu\eta)^{KR} \geq \frac{\mu c^2}{2}\left(1 - \frac{2\mu}{H} - \frac{1}{4R}\right)^R \tag{83}$$

Then, if $R \geq \frac{H}{4\mu}$, then

$$\frac{\mu c^2}{2}(1 - \mu\eta)^{KR} \geq \frac{\mu c^2}{2}\left(1 - \frac{3\mu}{H}\right)^R \geq \frac{\mu c^2}{2}\left(\left(1 - \frac{6\mu}{H}\left(1 - \frac{1}{e}\right)\right)^{\frac{H}{6\mu}}\right)^{\frac{6\mu R}{H}} \geq \frac{\mu c^2}{2}\exp\left(-\frac{6\mu R}{H}\right) \tag{84}$$

Otherwise, if $R \leq \frac{H}{4\mu}$, then

$$\frac{\mu c^2}{2}(1 - \mu\eta)^{KR} \geq \frac{\mu c^2}{2}\left(1 - \frac{1}{2R}\right)^R \geq \frac{\mu c^2}{4} \geq \frac{\mu c^2}{4}\exp\left(-\frac{6\mu R}{H}\right) \tag{85}$$

Therefore, when $(1 - \mu\eta)^K \geq \frac{H - 2\mu}{H} - \frac{1}{4R}$,

$$\mathbb{E}F(\hat{x}) - F^* \geq \frac{\mu c^2}{4}\exp\left(-\frac{6\mu R}{H}\right) \tag{86}$$

On the other hand, if $(1 - \mu\eta)^K \leq \frac{H - 2\mu}{H} - \frac{1}{4R}$, then

$$\mathbb{E}F(\hat{x}) - F^* \geq \frac{\zeta_*^2(H + 2\mu)}{32\mu^2}\left(\frac{H - 2\mu}{H} - (1 - \mu\eta)^K\right)^2 \tag{87}$$

$$\geq \frac{\zeta_*^2(H + 2\mu)}{32\mu^2}\left(\frac{1}{4R}\right)^2 \tag{88}$$

$$\geq \frac{H\zeta_*^2}{512\mu^2 R^2} \tag{89}$$

**Combination:** Combining these three cases, we conclude that

$$\mathbb{E}F(\hat{x}) - F^* \geq \min\left\{\frac{\mu c^2}{4}\exp\left(-\frac{6\mu R}{H}\right), \frac{H\zeta_*^2}{512\mu^2 R^2}\right\} \tag{90}$$

In the strongly convex case, we recall that $F(0) - F(x^*) = \mu c^2$, so we choose $\mu = \lambda$ and $c^2 = \frac{\Delta}{\lambda}$ so that the objective satisfies the strongly convex assumptions. Now, the lower bound reads (for a universal constant $\beta$)

$$\mathbb{E}F(\hat{x}) - F^* \geq \beta \cdot \min\left\{\Delta\exp\left(-\frac{6\lambda R}{H}\right), \frac{H\zeta_*^2}{512\lambda^2 R^2}\right\} \tag{91}$$

In the convex case, we recall that $\|x^*\|^2 \leq 2c^2$, so we choose $c^2 = \frac{B}{2}$ so that the convex assumptions are satisfied. We now have two options, if $R \leq \frac{H^2 B^2}{\zeta_*^2}$, then we pick $\mu = \left(\frac{H\zeta_*^2}{B^2 R^2}\right)^{1/3}$ so that the lower bound reads

$$\mathbb{E}F(\hat{x}) - F^* \geq \beta \cdot \frac{\left(H\zeta_*^2 B^4\right)^{1/3}}{R^{2/3}}\exp\left(-\frac{6\zeta_*^{2/3}R^{1/3}}{H^{2/3}B^{2/3}}\right) \tag{92}$$

$$\geq \beta \cdot \frac{\left(H\zeta_*^2 B^4\right)^{1/3}}{R^{2/3}}\exp(-6) \tag{93}$$

$$\geq \beta' \cdot \frac{\left(H\zeta_*^2 B^4\right)^{1/3}}{R^{2/3}} \tag{94}$$

On the other hand, if $R \geq \frac{H^2 B^2}{\zeta_*^2}$, then we pick $\mu = \frac{H}{6R}$ so the lower bound reads

$$\mathbb{E}F(\hat{x}) - F^* \geq \beta \cdot \min\left\{\frac{HB^2}{R}, \frac{\zeta_*^2}{H}\right\} = \beta \cdot \frac{HB^2}{R} \tag{95}$$

Consequently,

$$\mathbb{E}F(\hat{x}) - F^* \geq \beta \cdot \min\left\{\frac{HB^2}{R}, \frac{\left(H\zeta_*^2 B^4\right)^{1/3}}{R^{2/3}}\right\} \tag{96}$$

Combining these with the stochastic terms completes the proof. $\qquad\square$

# E    Proof of Theorem 3

We prove the theorem with the help of several technical lemmas.

**Lemma 7.** *For any stepsize $\eta_t \leq \frac{1}{10H}$*

$$\mathbb{E}[F(\bar{x}_t) - F^*] \leq \left(\frac{1}{\eta_t} - \lambda\right)\mathbb{E}\|\bar{x}_t - x^*\|^2 - \frac{1}{\eta_t}\mathbb{E}\|\bar{x}_{t+1} - x^*\|^2 + \frac{3\sigma_*^2\eta_t}{M} + \frac{2H}{M}\sum_{m=1}^{M}\mathbb{E}\|\bar{x}_t - x_t^m\|^2$$

*Proof.* This lemma and its proof are nearly identical to [Lemma 8 12]. We include a proof here in order to keep the paper self-contained.

Let $\bar{x}_{t+1} = \frac{1}{M}\sum_{m=1}^{M} x_t^m$ be the average of the machines' local iterates at time $t$. Then,

$$\mathbb{E}\|\bar{x}_{t+1} - x^*\|^2 = \mathbb{E}\left\|\bar{x}_t - \frac{\eta_t}{M}\sum_{m=1}^{M}\nabla F_m(x_t^m) - x^*\right\|^2 + \eta_t^2\mathbb{E}\left\|\frac{1}{M}\sum_{m=1}^{M}\nabla f(x_t^m; z_t^m) - \nabla F_m(x_t^m)\right\|^2 \tag{97}$$

Beginning with the first term of (97):

$$\mathbb{E}\left\|\bar{x}_t - \frac{\eta_t}{M}\sum_{m=1}^{M}\nabla F_m(x_t^m) - x^*\right\|^2$$

$$= \mathbb{E}\|\bar{x}_t - x^*\|^2 + \eta_t^2\mathbb{E}\left\|\frac{1}{M}\sum_{m=1}^{M}\nabla F_m(x_t^m)\right\|^2 - \frac{2\eta_t}{M}\sum_{m=1}^{M}\mathbb{E}\left\langle\bar{x}_t - x^*, \nabla F_m(x_t^m)\right\rangle \tag{98}$$

We can bound the second term of (98) with:

$$\eta_t^2\mathbb{E}\left\|\frac{1}{M}\sum_{m=1}^{M}\nabla F_m(x_t^m)\right\|^2$$

$$\leq 2\eta_t^2\mathbb{E}\left\|\frac{1}{M}\sum_{m=1}^{M}\nabla F_m(x_t^m) - \nabla F_m(\bar{x}_t)\right\|^2 + 2\eta_t^2\mathbb{E}\left\|\frac{1}{M}\sum_{m=1}^{M}\nabla F_m(\bar{x}_t) - \nabla F_m(x^*)\right\|^2 \tag{99}$$

$$\leq \frac{2\eta_t^2}{M}\sum_{m=1}^{M}\mathbb{E}\|\nabla F_m(x_t^m) - \nabla F_m(\bar{x}_t)\|^2 + 2\eta_t^2\mathbb{E}\|\nabla F(\bar{x}_t) - \nabla F(x^*)\|^2 \tag{100}$$

$$\leq \frac{2H^2\eta_t^2}{M}\sum_{m=1}^{M}\mathbb{E}\|x_t^m - \bar{x}_t\|^2 + 4H\eta_t^2\mathbb{E}[F(\bar{x}_t) - F(x^*)] \tag{101}$$

For the third term of (98):

$$-\frac{2\eta_t}{M}\sum_{m=1}^{M}\mathbb{E}\left\langle\bar{x}_t - x^*, \nabla F_m(x_t^m)\right\rangle$$

$$= -\frac{2\eta_t}{M}\sum_{m=1}^{M}\mathbb{E}\left\langle x_t^m - x^*, \nabla F_m(x_t^m)\right\rangle + \frac{2\eta_t}{M}\sum_{m=1}^{M}\mathbb{E}\left\langle x_t^m - \bar{x}_t, \nabla F_m(x_t^m)\right\rangle \tag{102}$$

$$\leq -\frac{2\eta_t}{M}\sum_{m=1}^{M}\mathbb{E}\left[F_m(x_t^m) - F_m(x^*) + \frac{\lambda}{2}\|x_t^m - x^*\|^2\right]$$

$$+ \frac{2\eta_t}{M}\sum_{m=1}^{M}\mathbb{E}\left[F_m(x_t^m) - F_m(\bar{x}_t) + \frac{H}{2}\|x_t^m - \bar{x}_t\|^2\right] \tag{103}$$

$$\leq -2\eta_t\mathbb{E}\left[F(\bar{x}_t) - F(x^*) + \frac{\lambda}{2}\|\bar{x}_t - x^*\|^2\right] + \frac{H\eta_t}{M}\sum_{m=1}^{M}\|x_t^m - \bar{x}_t\|^2 \tag{104}$$

Finally, for the second term of (97)

$$\eta_t^2 \mathbb{E} \left\| \frac{1}{M} \sum_{m=1}^{M} \nabla f(x_t^m; z_t^m) - \nabla F_m(x_t^m) \right\|^2$$

$$= \frac{\eta_t^2}{M^2} \sum_{m=1}^{M} \mathbb{E} \| \nabla f(x_t^m; z_t^m) - \nabla F_m(x_t^m) \|^2 \tag{105}$$

$$\leq \frac{3\eta_t^2}{M^2} \sum_{m=1}^{M} \left[ \mathbb{E} \| \nabla f(x_t^m; z_t^m) - \nabla f(\bar{x}_t; z_t^m) \|^2 + \mathbb{E} \| \nabla f(\bar{x}_t; z_t^m) - \nabla f(x^*; z_t^m) \|^2 \right.$$

$$\left. + \mathbb{E} \| \nabla f(x^*; z_t^m) - \nabla F_m(x^*) \|^2 \right] \tag{106}$$

$$\leq \frac{3\eta_t^2}{M^2} \sum_{m=1}^{M} \left[ H^2 \mathbb{E} \| x_t^m - \bar{x}_t \|^2 + 2H \mathbb{E}[F_m(\bar{x}_t) - F_m(x^*)] + \sigma_{*,m}^2 \right] \tag{107}$$

$$\leq \frac{3\eta_t^2}{M^2} \sum_{m=1}^{M} \left[ H^2 \mathbb{E} \| x_t^m - \bar{x}_t \|^2 + 2H \mathbb{E}[F(\bar{x}_t) - F(x^*)] + \sigma_{*,m}^2 \right] \tag{108}$$

Combining all these results back into (97), we have

$$\mathbb{E} \| \bar{x}_{t+1} - x^* \|^2 \leq (1 - \lambda \eta_t) \mathbb{E} \| \bar{x}_t - x^* \|^2 + \frac{H \eta_t + 5H^2 \eta_t^2}{M} \sum_{m=1}^{M} \mathbb{E} \| x_t^m - \bar{x}_t \|^2$$

$$+ (10H \eta_t^2 - 2\eta_t) \mathbb{E}[F(\bar{x}_t) - F(x^*)] + \frac{3\eta_t^2 \sigma_*^2}{M} \tag{109}$$

$$\leq (1 - \lambda \eta_t) \mathbb{E} \| \bar{x}_t - x^* \|^2 + \frac{2H \eta_t}{M} \sum_{m=1}^{M} \mathbb{E} \| x_t^m - \bar{x}_t \|^2$$

$$- \eta_t \mathbb{E}[F(\bar{x}_t) - F(x^*)] + \frac{3\eta_t^2 \sigma_*^2}{M} \tag{110}$$

where for the final line we used that $\eta_t \leq \frac{1}{10H}$. Rearranging completes the proof. □

**Lemma 8.** *If* $\sup_{x,m} \| \nabla F_m(x) - \nabla F(x) \|^2 \leq \bar{\zeta}^2$, *then for any fixed stepsize* $\eta$

$$\frac{1}{M} \sum_{m=1}^{M} \mathbb{E} \| x_t^m - \bar{x}_t \|^2 \leq 3K \sigma^2 \eta^2 + 6K^2 \eta^2 \bar{\zeta}^2$$

*Similarly, the decreasing stepsize* $\eta_t = \frac{2}{\lambda(a+t+1)}$ *for any* $a$

$$\frac{1}{M} \sum_{m=1}^{M} \mathbb{E} \| x_t^m - \bar{x}_t \|^2 \leq 3K \sigma^2 \eta_{t-1}^2 + 6K^2 \bar{\zeta}^2 \eta_{t-1}^2$$

*Proof.* By Jensen's inequality

$$\mathbb{E} \| x_t^m - \bar{x}_t \|^2 \leq \frac{1}{M} \sum_{n=1}^{M} \mathbb{E} \| x_t^m - x_t^n \|^2 \tag{111}$$

Therefore, it suffices to bound $\mathbb{E}\|x_t^m - x_t^n\|^2$, which we do now:

$$
\begin{aligned}
&\mathbb{E}\|x_t^m - x_t^n\|^2 \\
&\leq \mathbb{E}\left\|x_{t-1}^m - x_{t-1}^n - \eta_{t-1}\bigl(\nabla F(x_{t-1}^m) - \nabla F(x_{t-1}^n)\bigr)\right. \\
&\quad + \eta_{t-1}\bigl(\nabla F(x_{t-1}^m) - \nabla F_m(x_{t-1}^m) - \nabla F(x_{t-1}^n) + \nabla F_n(x_{t-1}^n)\bigr)\Bigr\|^2 + \eta_{t-1}^2\sigma_m^2 & (112)
\end{aligned}
$$

$$
\begin{aligned}
&\leq \inf_{\gamma>0}\left(1 + \frac{1}{\gamma}\right)\mathbb{E}\left\|x_{t-1}^m - x_{t-1}^n - \eta_{t-1}\bigl(\nabla F(x_{t-1}^m) - \nabla F(x_{t-1}^n)\bigr)\right\|^2 \\
&\quad + (1+\gamma)\eta_{t-1}^2\mathbb{E}\left\|\nabla F(x_{t-1}^m) - \nabla F_m(x_{t-1}^m) - \nabla F(x_{t-1}^n) + \nabla F_n(x_{t-1}^n)\right\|^2 + \eta_{t-1}^2\sigma_m^2 & (113)
\end{aligned}
$$

$$
\begin{aligned}
&\leq \inf_{\gamma>0}\left(1 + \frac{1}{\gamma}\right)(1 - \lambda\eta_{t-1})\mathbb{E}\left\|x_{t-1}^m - x_{t-1}^n\right\|^2 + \eta_{t-1}^2\sigma_m^2 \\
&\quad + (1+\gamma)\eta_{t-1}^2\mathbb{E}\left\|\nabla F(x_{t-1}^m) - \nabla F_m(x_{t-1}^m)\right\|^2 \\
&\quad + (1+\gamma)\eta_{t-1}^2\mathbb{E}\left\|\nabla F(x_{t-1}^n) - \nabla F_n(x_{t-1}^n)\right\|^2 \\
&\quad - 2(1+\gamma)\eta_{t-1}^2\mathbb{E}\left\langle \nabla F(x_{t-1}^m) - \nabla F_m(x_{t-1}^m), \nabla F(x_{t-1}^n) - \nabla F_n(x_{t-1}^n)\right\rangle & (114)
\end{aligned}
$$

For the third inequality we used Lemma 1. Therefore,

$$
\begin{aligned}
&\frac{1}{M^2}\sum_{m=1}^{M}\sum_{n=1}^{M}\mathbb{E}\|x_t^m - x_t^n\|^2 \\
&\quad \leq \frac{1}{M^2}\sum_{m=1}^{M}\inf_{\gamma>0}\left(1 + \frac{1}{\gamma}\right)(1 - \lambda\eta_{t-1})\mathbb{E}\left\|x_{t-1}^m - x_{t-1}^n\right\|^2 + \eta_{t-1}^2\sigma_m^2 + 2(1+\gamma)\eta_{t-1}^2\bar{\zeta}^2 & (115)
\end{aligned}
$$

We will unroll this recurrence, using that $x_{t_0}^m = x_{t_0}^n$ for all $m, n$ where $t_0$ is the most recent time that the iterates were synchronized, so $t - t_0 \leq K - 1$. Taking $\gamma = K - 1$, we have

$$
\begin{aligned}
\frac{1}{M^2}\sum_{m=1}^{M}\sum_{n=1}^{M}\mathbb{E}\|x_t^m - x_t^n\|^2 &= \sum_{i=t_0}^{t-1}\bigl(\eta_i^2\sigma^2 + 2(1+\gamma)\eta_i^2\bar{\zeta}^2\bigr)\prod_{j=i+1}^{t-1}\left(1 + \frac{1}{\gamma}\right)(1 - \lambda\eta_j) & (116) \\
&\leq \sum_{i=t_0}^{t-1}\bigl(\eta_i^2\sigma^2 + 2K\eta_i^2\bar{\zeta}^2\bigr)\prod_{j=i+1}^{t-1}\left(1 + \frac{1}{K-1}\right)(1 - \lambda\eta_j) & (117) \\
&\leq \sum_{i=t_0}^{t-1}\bigl(\eta_i^2\sigma^2 + 2K\eta_i^2\bar{\zeta}^2\bigr)\left(1 + \frac{1}{K-1}\right)^{K-1}\prod_{j=i+1}^{t-1}(1 - \lambda\eta_j) & (118) \\
&\leq 3\bigl(\sigma^2 + 2K\bar{\zeta}^2\bigr)\sum_{i=t_0}^{t-1}\eta_i^2\prod_{j=i+1}^{t-1}(1 - \lambda\eta_j) & (119)
\end{aligned}
$$

For a constant stepsize $\eta$,

$$
\begin{aligned}
\frac{1}{M^2}\sum_{m=1}^{M}\sum_{n=1}^{M}\mathbb{E}\|x_t^m - x_t^n\|^2 &\leq 3\bigl(\sigma^2 + 2K\bar{\zeta}^2\bigr)\sum_{i=t_0}^{t-1}\eta^2 & (120) \\
&\leq 3K\bigl(\sigma^2 + 2K\bar{\zeta}^2\bigr)\eta^2 & (121)
\end{aligned}
$$

For decreasing stepsize $\eta_t = \frac{2}{\lambda(a+t+1)}$

$$\frac{1}{M^2} \sum_{m=1}^{M} \sum_{n=1}^{M} \mathbb{E}\|x_t^m - x_t^n\|^2 \leq 3(\sigma^2 + 2K\bar{\zeta}^2) \sum_{i=t_0}^{t-1} \eta_i^2 \prod_{j=i+1}^{t-1} \frac{a+j-1}{a+j+1} \tag{122}$$

$$= 3(\sigma^2 + 2K\bar{\zeta}^2) \sum_{i=t_0}^{t-1} \eta_i^2 \frac{(a+i)(a+i+1)}{(a+t)(a+t+1)} \tag{123}$$

$$= 3(\sigma^2 + 2K\bar{\zeta}^2) \sum_{i=t_0}^{t-1} \eta_i^2 \frac{\eta_{t-1}\eta_t}{\eta_{i-1}\eta_i} \tag{124}$$

$$\leq 3(\sigma^2 + 2K\bar{\zeta}^2) \sum_{i=t_0}^{t-1} \eta_i^2 \frac{\eta_{t-1}^2}{\eta_i^2} \tag{125}$$

$$= 3K(\sigma^2 + 2K\bar{\zeta}^2)\eta_{t-1}^2 \tag{126}$$

$\square$

**Theorem 3.** *When* $\sup_x \frac{1}{M} \sum_{m=1}^{M} \|\nabla F_m(x) - \nabla F(x)\|^2 \leq \bar{\zeta}^2$, *an average of the Local SGD iterates guarantees under the convex and strongly convex assumptions, respectively*

$$\mathbb{E}F(\hat{x}) - F^* \leq c \cdot \left( \frac{HB^2}{KR} + \frac{(H\bar{\zeta}^2 B^4)^{1/3}}{R^{2/3}} + \frac{(H\sigma^2 B^4)^{1/3}}{K^{1/3}R^{2/3}} + \frac{\sigma_* B}{\sqrt{MKR}} \right),$$

$$\mathbb{E}F(\hat{x}) - F^* \leq c \cdot \left( \frac{H^2 B^2}{HKR + \lambda K^2 R^2} + \left( \frac{H\bar{\zeta}^2}{\lambda^2 R^2} + \frac{H\sigma^2}{\lambda^2 KR^2} \right) \log\left( \frac{H}{\lambda} + KR \right) + \frac{\sigma_*^2}{\lambda MKR} \right).$$

*Proof.* By Lemma 7, for any $\eta_t \leq \frac{1}{10H}$

$$\mathbb{E}[F(\bar{x}_t) - F^*] \leq \left( \frac{1}{\eta_t} - \lambda \right) \mathbb{E}\|\bar{x}_t - x^*\|^2 - \frac{1}{\eta_t}\mathbb{E}\|\bar{x}_{t+1} - x^*\|^2 + \frac{3\sigma_*^2 \eta_t}{M} + \frac{2H}{M} \sum_{m=1}^{M} \mathbb{E}\|\bar{x}_t - x_t^m\|^2 \tag{127}$$

By Lemma 8, when $\eta_t = \eta$ is constant then

$$\frac{1}{M} \sum_{m=1}^{M} \mathbb{E}\|x_t^m - \bar{x}_t\|^2 \leq 3K\sigma^2\eta^2 + 6K^2\eta^2\bar{\zeta}^2 \tag{128}$$

and when $\eta_t = \frac{2}{\lambda(a+t+1)}$

$$\frac{1}{M} \sum_{m=1}^{M} \mathbb{E}\|x_t^m - \bar{x}_t\|^2 \leq 3K\sigma^2\eta_{t-1}^2 + 6K^2\bar{\zeta}^2\eta_{t-1}^2 \tag{129}$$

We now consider the convex and strongly convex cases separately:

**Convex case:** In the convex case, we use a constant stepsize $\eta$, so

$$\mathbb{E}[F(\bar{x}_t) - F^*]$$

$$\leq \frac{1}{\eta}\mathbb{E}\|\bar{x}_t - x^*\|^2 - \frac{1}{\eta}\mathbb{E}\|\bar{x}_{t+1} - x^*\|^2 + \frac{3\sigma_*^2 \eta}{M} + \frac{2H}{M} \sum_{m=1}^{M} \mathbb{E}\|\bar{x}_t - x_t^m\|^2 \tag{130}$$

$$\leq \frac{1}{\eta}\mathbb{E}\|\bar{x}_t - x^*\|^2 - \frac{1}{\eta}\mathbb{E}\|\bar{x}_{t+1} - x^*\|^2 + \frac{3\sigma_*^2 \eta}{M} + 6HK\sigma^2\eta^2 + 12HK^2\eta^2\bar{\zeta}^2 \tag{131}$$

Therefore, by the convexity of $F$

$$\mathbb{E}\left[ F\left( \frac{1}{KR} \sum_{t=1}^{KR} \bar{x}_t \right) - F^* \right] \leq \frac{1}{KR} \sum_{t=1}^{KR} \mathbb{E}[F(\bar{x}_t) - F^*] \tag{132}$$

$$\leq \frac{B^2}{\eta KR} + \frac{3\sigma_*^2 \eta}{M} + 6HK\sigma^2\eta^2 + 12HK^2\eta^2\bar{\zeta}^2 \tag{133}$$

Choosing

$$\eta = \min\left\{\frac{1}{10H}, \frac{B\sqrt{M}}{\sigma_*\sqrt{KR}}, \left(\frac{B^2}{HK^2\sigma^2}\right)^{1/3}, \left(\frac{B^2}{HK^2\bar{\zeta}^2}\right)^{1/3}\right\} \tag{134}$$

then ensures

$$\mathbb{E}\left[F\left(\frac{1}{KR}\sum_{t=1}^{KR}\bar{x}_t\right) - F^*\right] \le \frac{10HB^2}{KR}\frac{13\left(H\bar{\zeta}^2B^4\right)^{1/3}}{R^{2/3}} + \frac{7\left(H\sigma^2B^4\right)^{1/3}}{K^{1/3}R^{2/3}} + \frac{4\sigma_*B}{\sqrt{MKR}} \tag{135}$$

**Strongly convex case:** In the strongly convex case, we take the stepsize $\eta_t = \frac{2}{\lambda(a+t+1)}$ for $a = 20H/\lambda$ which ensures $\eta_t \le \frac{1}{10H}$. In addition, we define weights $w_t = (a+t)$ and define

$$\bar{x} = \frac{1}{W}\sum_{t=1}^{KR} w_t\bar{x}_t \tag{136}$$

where $W = \sum_{t=1}^{KR} w_t \ge \frac{1}{2}KR(a+KR)$. By the convexity of $F$,

$$\mathbb{E}F(\bar{x}) - F^*$$

$$\le \frac{1}{W}\sum_{t=1}^{KR}(a+t)\mathbb{E}F(\bar{x}_t) - F^* \tag{137}$$

$$\le \frac{\lambda(a+1)(a+2)B^2}{2W} + \frac{1}{W}\sum_{t=1}^{KR}\left[\frac{6\sigma_*^2}{\lambda M} + \frac{2H(a+t)}{M}\sum_{m=1}^{M}\mathbb{E}\|\bar{x}_t - x_t^m\|^2\right] \tag{138}$$

$$\le \frac{\lambda(a+1)(a+2)B^2}{2W} + \frac{6\sigma_*^2KR}{W\lambda M} + \frac{6HK\sigma^2 + 12HK^2\bar{\zeta}^2}{W}\sum_{t=1}^{KR}(a+t)\eta_{t-1}^2 \tag{139}$$

$$\le \frac{\lambda(a+1)(a+2)B^2}{2W} + \frac{6\sigma_*^2KR}{W\lambda M} + \frac{6HK\sigma^2 + 12HK^2\bar{\zeta}^2}{\lambda^2 W}(1+\log(a+KR)) \tag{140}$$

$$\le \frac{132H^2B^2}{\lambda KR(10H/\lambda + KR)} + \left(\frac{12H\bar{\zeta}^2}{\lambda^2R^2} + \frac{6H\sigma^2}{\lambda^2KR^2}\right)\log\left(\frac{13H}{\lambda} + KR\right) + \frac{6\sigma_*^2}{\lambda MKR} \tag{141}$$

Note that in the strongly convex case, it is likely possible to achieve a first term scaling with $\exp(-KR)$ using a method similar to Lemma 2. However, the recurrence we derived here has a different form, and it is difficult to determine the correct stepsize and weighting schedule to achieve linear convergence. □

## F    Details of Experiments

The training set of MNIST (60,000 examples) was divided by digit into ten groups of equal size $n \approx 6,000$ (which required discarding some examples from the more common digits). PCA was used to reduce the dimensionality to 100, but no other preprocessing was used.

Then, for each of the 25 combinations $(i,j)$ for even $i$ and odd $j$, a binary classification "task" was created, i.e. classifying even (+1) versus odd (−1). These tasks were arbitrarily labelled task $1, 2, \ldots, 25$.

For each $p \in [0.0, 0.2, 0.4, 0.6, 0.8, 1.0]$, machine $m$ was assigned data composed of $p \cdot 2n$ random examples from task $m$, and $(1-p) \cdot 2n$ random examples from a mixture of all the tasks.

Local and Minibatch SGD were then used to optimize the logistic loss for each of the six described local datasets. The constant stepsize was tuned (from a log-scale grid of 10 points ranging from $e^{-6}, \ldots, e^0$ for Minibatch SGD, and a log-scale grid of 10 points ranging from $e^{-8}, \ldots, e^{-1}$ for Local SGD) for each value of $p$, $K$, and $R$ individually, and the average loss over four runs is reported for the best stepsize for each point in the plot. That is, each point in the plot represents the best possible performance of the algorithm for that $p$, $K$, and $R$ specifically.

Finally, we computed the value of $\zeta_*^2$ as a function of $p$ by using Newton's method to compute a very accurate estimate of the minimizer, and then explicitly calculating $\zeta_*^2(p)$ at that point.

## G   Proof of Theorem 4

For this lower bound, the gradients will always be noiseless, so we simply define the expectation of the local functions. Furthermore, we will construct just two local functions $F_1$ and $F_2$. For the case $M > 2$, $F_1$ will be assigned to the first $\lfloor M/2 \rfloor$ machines, and $F_2$ to the next $\lfloor M/2 \rfloor$ machines. If there is an odd number of machines, we simply assign the last machine $F_3(x) = \frac{\lambda}{2}\|x\|^2$, which will reduce the lower bound by a factor of at most $\frac{M-1}{M}$. Therefore, we proceed by focusing on the case $M = 2$.

We define the following $H$-smooth and $\lambda$-strongly convex functions on $\mathbb{R}^d$ for even $d$:

$$F(x) = \frac{1}{2}(F_1(x) + F_2(x)) \tag{142}$$

$$F_1(x) = \frac{H-\lambda}{8}\left(x_1^2 - 2Cx_1 + \beta x_d^2 + \sum_{i=1}^{d/2-1}(x_{2i+1} - x_{2i})^2\right) + \frac{\lambda}{2}\|x\|^2 \tag{143}$$

$$F_2(x) = \frac{H-\lambda}{8}\left(\sum_{i=1}^{d/2}(x_{2i} - x_{2i-1})^2\right) + \frac{\lambda}{2}\|x\|^2 \tag{144}$$

Here, $\beta$ and $C$ are constants which will be chosen later.

These functions are identical to ones used by Woodworth and Srebro [24] to prove lower bounds for finite sum optimization, and are very similar both to classic work by Nesterov [18] on lower bounds and to more closely related work by Arjevani and Shamir [1]. Arjevani and Shamir also prove lower bounds for distributed optimization algorithms, but their slightly different construction made it more difficult to tune $\zeta_*^2$, which is necessary for our lower bound.

These functions have the following important property: let $E_k = \text{span}\{e_1, \ldots, e_k\}$ be the set of vectors whose $k+1, \ldots, d$ coordinates are all zero, then for all $x_k \in E_k$ for even $k$

$$\nabla F_1(x_k) \in E_{k+1} \qquad \text{and} \qquad \nabla F_2(x_k) \in E_k \tag{145}$$

and for $x_k \in E_k$ for odd $k$

$$\nabla F_1(x_k) \in E_k \qquad \text{and} \qquad \nabla F_2(x_k) \in E_{k+1} \tag{146}$$

For algorithms whose iterates, for example, remain in the span of previous gradients, the only way to access the next coordinate is to query the gradient of one of the two functions—$F_1$ if the next coordinate is odd, and $F_2$ if the next coordinate is even. Since each machine will only have access to one of the two functions throughout each round of communication, this means that each round of communication can only unlock a single new coordinate. We now formalize this.

Following Carmon et al. [5], we define:

**Definition 2** (Distributed zero-respecting algorithm). *For a vector $v$, let $\text{supp}(v) = \{i \in \{1, \ldots, d\} : v_i \neq 0\}$. We say that an optimization algorithm is distributed zero-respecting if for all $t$ and $m$, the $t$th query on the $m$th machine, $x_t^m$ satisfies*

$$\text{supp}(x_t^m) \subseteq \bigcup_{s<t} \text{supp}(\nabla f(x_s^m; z_s^m)) \cup \bigcup_{m' \neq m}\bigcup_{s \leq \pi_m(t,m')} \text{supp}(\nabla f(x_s^{m'}; z_s^{m'}))$$

*where $\pi_m(t, m')$ is the most recent time before $t$ when machines $m$ and $m'$ communicated with each other.*

This definition captures a very wide variety of distributed optimization algorithms, including minibatch SGD, accelerated minibatch SGD, local SGD, coordinate descent methods, and many more. Algorithms which are *not* distributed zero-respecting are those whose iterates have components in directions about which the algorithm has no information, meaning that in some sense, it is just "wild guessing." Using techniques similar to Woodworth and Srebro [Theorem 7 24] and Carmon et al. [5], it should be possible to extend this lower bound beyond distributed zero-respecting algorithms to arbitrary randomized algorithms.

We now argue that the progress of distributed zero-respecting algorithms is controlled by the number of rounds of communication, $R$, regardless of $K$:

**Lemma 9.** *Let $\hat{x}$ be the output after $R$ rounds of communication of a distributed zero-respecting algorithm optimizing $F = \frac{1}{2}(F_1 + F_2)$ as defined in (142). Then,*

$$\text{supp}(x_t^m) \in E_R$$

*Proof.* The definition of a zero-respecting algorithm requires that every machine's initial iterate $x_0^m = 0$. We will now prove the Lemma by induction on the round of communication.

As a base case, for the first iteration of the first round of communication:

$$\nabla F_1(x_0^1) = \nabla F_1(0) = \frac{(\lambda - H)C}{4}e_1 \in E_1 \qquad \text{and} \qquad \nabla F_2(x_1^2) = \nabla F_2(0) = 0 \in E_0 \quad (147)$$

Therefore, by the distributed zero-respecting property, $x_2^1 \in E_1$ and $x_2^2 \in E_0$. Furthermore, for all $y_1 \in E_1$, $\nabla F_1(y_1) \in E_1$ and for all $y_0 \in E_0$, $\nabla F_2(y_0) \in E_0$. Therefore, further gradient queries on each machine will not change the set of coordinates that the distributed zero-respecting property allows to be non-zero. We conclude that $x_t^1 \in E_1$ and $x_t^2 \in E_0$ for all $t$ until machines 1 and 2 communicate with each other.

Now, suppose that after $r-1$ rounds of communication, $x_t^1, x_t^2 \in E_{r-1}$. If $r$ is even, then $\nabla F_1(x_t^1) \in E_{r-1}$ and $\nabla F_2(x_t^2) \in E_r$. Furthermore, additional gradient computations within the $r$th round of communication will not expand the set of coordinates that the distributed zero-respecting property will allow to be non-zero. Therefore, both machines' coordinates will remain in $E_r$ until the end of the $r$th round of communication. A similar argument can be made for odd $r$. $\qquad \square$

Now, we will compute the minimizer of $F$. We note that by the definition of $F_1$ and $F_2$,

$$F(x) = \frac{H - \lambda}{16}\left(x_1^2 - 2Cx_1 + \beta x_d^2 + \sum_{i=2}^{d}(x_i - x_{i-1})^2\right) + \frac{\lambda}{2}\|x\|^2 \qquad (148)$$

Calculating the gradient of $F$, we see that $x^* = \arg\min_x F(x)$ must satisfy

$$C = \left(2 + \frac{8\lambda}{H - \lambda}\right)x_1^* - x_2^*$$

$$0 = \left(2 + \frac{8\lambda}{H - \lambda}\right)x_i^* - x_{i+1}^* - x_{i-1}^* \qquad \forall_{i \in \{2,\ldots,d-1\}} \qquad (149)$$

$$0 = \left(1 + \beta + \frac{8\lambda}{H - \lambda}\right)x_d^* - x_{d-1}^*$$

Let $q$ be the smaller solution of the quadratic equation

$$1 - \left(2 + \frac{8\lambda}{H - \lambda}\right)q + q^2 = 0 \qquad (150)$$

That is,

$$q = 1 + \frac{4\lambda}{H - \lambda} - \sqrt{\frac{16\lambda^2}{(H - \lambda)^2} + \frac{8\lambda}{H - \lambda}} \qquad (151)$$

$$= 1 + \frac{4\lambda}{H - \lambda}\left(1 - \sqrt{1 + \frac{H - \lambda}{2\lambda}}\right) \qquad (152)$$

$$= 1 - \frac{2}{\left(1 - \sqrt{1 + \frac{H-\lambda}{2\lambda}}\right)\left(1 + \sqrt{1 + \frac{H-\lambda}{2\lambda}}\right)}\left(1 - \sqrt{1 + \frac{H - \lambda}{2\lambda}}\right) \qquad (153)$$

$$= \frac{\sqrt{1 + \frac{H-\lambda}{2\lambda}} - 1}{\sqrt{1 + \frac{H-\lambda}{2\lambda}} + 1} \qquad (154)$$

Let $\alpha = \sqrt{1 + \frac{H-\lambda}{2\lambda}}$ so that $q = \frac{\alpha-1}{\alpha+1}$, and define $\beta = 1 - q$. Then it is straightforward to confirm that

$$x^* = C \sum_{i=1}^{d} q^i e_i \tag{155}$$

satisfies all of the conditions (149), and is thus the minimizer of $F$. This point has value

$$F(x^*) = \frac{C^2(H-\lambda)}{16} \left( q^2 - 2q + \beta q^{2d} + (1-q)^2 \sum_{i=2}^{d} q^{2i-2} + \frac{8\lambda}{H-\lambda} \sum_{i=1}^{d} q^{2i} \right) \tag{156}$$

$$= \frac{C^2(H-\lambda)}{16} \left( -1 + \beta q^{2d} + (1-q)^2 \sum_{i=1}^{d} q^{2i-2} + \frac{8\lambda}{H-\lambda} \sum_{i=1}^{d} q^{2i} \right) \tag{157}$$

$$= \frac{C^2(H-\lambda)}{16} \left( -1 + (1-q)q^{2d} + \frac{8\lambda}{H-\lambda} \sum_{i=1}^{d} q^{2i-1} + q^{2i} \right) \tag{158}$$

$$= \frac{C^2(H-\lambda)}{16} \left( -1 + (1-q)q^{2d} + \frac{8\lambda}{H-\lambda} \left( \frac{q(1-q^{2d})}{1-q^2} + \frac{q^2(1-q^{2d})}{1-q^2} \right) \right) \tag{159}$$

$$= \frac{C^2(H-\lambda)}{16} \left( -1 + (1-q)q^{2d} + \frac{(1-q)^2}{q} \left( \frac{q(1-q^{2d})}{1-q^2} + \frac{q^2(1-q^{2d})}{1-q^2} \right) \right) \tag{160}$$

$$= \frac{C^2(H-\lambda)}{16} \left( -1 + (1-q)q^{2d} + (1-q)(1-q^{2d}) \right) \tag{161}$$

$$= \frac{-qC^2(H-\lambda)}{16} \tag{162}$$

For the third equality, we used that (150) implies $(1-q)^2 = \frac{8\lambda q}{H-\lambda}$. For the fifth inequality, we used that $\frac{8\lambda}{H-\lambda} = \frac{(1-q)^2}{q}$. This solution has norm

$$\|x^*\|^2 = C^2 \sum_{i=1}^{d} q^{2i} = C^2 \frac{q^2(1-q^{2d})}{1-q^2} \leq \frac{q^2 C^2}{1-q^2} = \frac{C^2(\alpha-1)^2}{4\alpha} \leq \frac{\alpha C^2}{4} \tag{163}$$

Furthermore,

$$F(0) - F(x^*) = -F(x^*) = \frac{qC^2(H-\lambda)}{16} \tag{164}$$

Finally, we evaluate the degree of heterogeneity:

$$\zeta_*^2 = \frac{1}{2} \sum_{m=1}^{2} \|\nabla F_m(x^*)\|^2 = \|\nabla F_1(x^*)\|^2 = \|\nabla F_2(x^*)\|^2 \tag{165}$$

$$= \frac{(H-\lambda)^2}{64} \left\| 2\sum_{i=1}^{d/2} (x_{2i}^* - x_{2i-1}^*)(e_{2i} - e_{2i-1}) + \frac{4\lambda}{H-\lambda} x^* \right\|^2 \tag{166}$$

$$= \frac{(H-\lambda)^2}{16} \sum_{i=1}^{d/2} \left[ \left( x_{2i-1}^* \left( -1 + \frac{4\lambda}{H-\lambda} \right) \right)^2 + \left( x_{2i}^* \left( 1 + \frac{4\lambda}{H-\lambda} \right) \right)^2 \right] \tag{167}$$

$$= \frac{C^2(H-\lambda)^2}{16} \sum_{i=1}^{d/2} \left[ q^{4i-2} \frac{(H-5\lambda)^2}{(H-\lambda)^2} + q^{4i} \frac{(H+3\lambda)^2}{(H-\lambda)^2} \right] \tag{168}$$

$$\leq \frac{(H+3\lambda)^2}{16} \|x^*\|^2 \tag{169}$$

$$\leq \frac{\alpha C^2 (H+3\lambda)^2}{64} \tag{170}$$

With this, we are ready to prove the lower bound.

**Theorem 4.** *For any M, K, and R, there exist two quadratic objectives satisfying the convex and strongly convex assumptions (for $H \geq 7\lambda$) such that the output of any distributed zero-respecting algorithm will have suboptimality in the convex and strongly convex case respectively,*

$$F(\hat{x}) - F^* \geq c\left(\min\left\{\frac{\zeta_*^2}{HR^2}, \frac{HB^2}{R^2}\right\} + \frac{\sigma B}{\sqrt{MKR}}\right),$$

$$F(\hat{x}) - F^* \geq c\left(\min\left\{\frac{\lambda\zeta_*^2}{H^2}, \frac{\Delta\sqrt{\lambda}}{\sqrt{H}}\right\}\exp\left(-\frac{8R\sqrt{\lambda}}{\sqrt{H}}\right) + \frac{\sigma^2}{\lambda MKR}\right).$$

*Proof.* By Lemma 9, the output of the algorithm $\hat{x} \in E_R$. Furthermore, since $F$ is $\lambda$-strongly convex, $F(\hat{x}) - F^* \geq \frac{\lambda}{2}\|\hat{x} - x^*\|^2$. Therefore,

$$\frac{F(\hat{x}) - F^*}{F(0) - F^*} \geq \frac{\frac{\lambda}{2}\|\hat{x} - x^*\|^2}{\frac{qC^2(H-\lambda)}{16}} \tag{171}$$

$$\geq \frac{8\lambda}{q(H-\lambda)}\sum_{i=R+1}^{d} q^{2i} \tag{172}$$

$$= \frac{8\lambda q(q^{2R} - q^{2d})}{(H-\lambda)(1-q^2)} \tag{173}$$

$$= \frac{(1-q)^2(q^{2R} - q^{2d})}{1-q^2} \tag{174}$$

$$= \frac{(1-q)(q^{2R} - q^{2d})}{1+q} \tag{175}$$

$$= \frac{q^{2R} - q^{2d}}{\alpha} \tag{176}$$

For the third equality we used that (150) implies $\frac{8\lambda q}{H-\lambda} = (1-q)^2$. For the final equality, we used that $q = \frac{\alpha-1}{\alpha+1}$. Taking $d \geq R + \frac{1}{2\ln(1/q)}$ ensures that $q^{2d} \leq \frac{q^{2R}}{2}$ so

$$\frac{F(\hat{x}) - F^*}{F(0) - F^*} \geq \frac{q^{2R}}{2\alpha} = \frac{\left(1 - \frac{2}{\alpha+1}\right)^{2R}}{2\alpha} \tag{177}$$

Therefore,

$$R \leq \frac{\ln\left(\frac{F(0)-F^*}{2\alpha\epsilon}\right)}{\ln\left(1 + \frac{2}{\alpha-1}\right)} \implies F(\hat{x}) - F^* \geq \epsilon \tag{178}$$

Using the fact that $\ln(1+x) \leq x$ and solving the above inequality on $R$ for $\epsilon$, we conclude that

$$F(\hat{x}) - F^* \geq \frac{F(0) - F^*}{2\alpha}\exp\left(-\frac{2R}{\alpha-1}\right) \tag{179}$$

In order to satisfy the strongly convex assumptions, we recall from (170) and (164) that we must choose $C$ such that

$$\frac{\alpha C^2(H+3\lambda)^2}{64} \leq \frac{\alpha C^2 H^2}{16} \leq \zeta_*^2 \tag{180}$$

$$\frac{qC^2(H-\lambda)}{16} \leq \frac{C^2 H}{16} \leq \Delta \tag{181}$$

Therefore, we choose $C^2 = 16 \min\left\{\frac{\zeta_*^2}{\alpha H^2}, \frac{\Delta}{H}\right\}$ meaning that

$$F(\hat{x}) - F^* \geq \frac{F(0) - F^*}{2\alpha} \exp\left(-\frac{2R}{\alpha - 1}\right) \tag{182}$$

$$\geq \frac{\min\left\{\frac{\zeta_*^2}{\alpha H}, \Delta\right\}}{2\alpha} \exp\left(-\frac{2R}{\alpha - 1}\right) \tag{183}$$

$$\geq \min\left\{\frac{\lambda\zeta_*^2}{H^2}, \frac{\sqrt{\lambda}\Delta}{2\sqrt{H}}\right\} \exp\left(-\frac{8\sqrt{\lambda}R}{\sqrt{H}}\right) \tag{184}$$

For the convex case, we note that in order to satisfy the convex assumptions, we must choose $C$ such that

$$\frac{\alpha C^2 (H + 3\lambda)^2}{64} \leq \frac{\alpha C^2 H^2}{16} \leq \zeta_*^2 \tag{185}$$

$$\frac{\alpha C^2}{4} \leq B^2 \tag{186}$$

We therefore choose $C^2 = 4 \min\left\{\frac{\zeta_*^2}{\alpha H^2}, \frac{B^2}{\alpha}\right\}$. Returning to (179), this means

$$F(\hat{x}) - F^* \geq \frac{F(0) - F^*}{2\alpha} \exp\left(-\frac{2R}{\alpha - 1}\right) \tag{187}$$

$$= \frac{qC^2(H - \lambda)}{32\alpha} \exp\left(-\frac{2R}{\alpha - 1}\right) \tag{188}$$

$$\geq \frac{q(H - \lambda)\min\left\{\frac{\zeta_*^2}{\alpha H^2}, \frac{B^2}{\alpha}\right\}}{8\alpha} \exp\left(-\frac{8\sqrt{\lambda}R}{\sqrt{H}}\right) \tag{189}$$

$$\geq q \min\left\{\frac{\zeta_*^2}{16\alpha^2 H}, \frac{HB^2}{16\alpha^2}\right\} \exp\left(-\frac{8\sqrt{\lambda}R}{\sqrt{H}}\right) \tag{190}$$

From here, we use that $H \geq 7\lambda$ implies $\alpha \geq 2$ so $q \geq 1/3$, so

$$F(\hat{x}) - F^* \geq \min\left\{\frac{\lambda\zeta_*^2}{48H^2}, \frac{\lambda B^2}{48}\right\} \exp\left(-\frac{8\sqrt{\lambda}R}{\sqrt{H}}\right) \tag{191}$$

Finally, this holds for any $\lambda \geq 0$, so it holds, in particular, for $\lambda = \frac{H}{64R^2}$ thus

$$F(\hat{x}) - F^* \geq c \cdot \min\left\{\frac{\zeta_*^2}{HR^2}, \frac{HB^2}{R^2}\right\} \tag{192}$$

Finally, it is well known that any first-order method which accesses at most $MKR$ stochastic gradients with variance $\sigma^2$ for a $\lambda$-strongly convex objective will suffer error at least $\beta \frac{\sigma^2}{\lambda MKR}$ in the worst case for a universal constant $\beta$ [17]. Similarly, any first-order method which accesses at most $MKR$ stochastic gradients with variance $\sigma^2$ for a convex objective with $\|x^*\| \leq B$ will suffer error at least $\beta \frac{\sigma B}{\sqrt{MKR}}$ in the worst case for a universal constant $\beta$ [17]. □

## H  Proof of Corollary 1

**Corollary 1.** *Accelerated Minibatch SGD is optimal when $\zeta_* \geq HB$ in the convex case, and is optimal up to log factors when $\zeta_*^2 \geq H^{3/2}/\sqrt{\lambda}$ in the strongly convex case.*

*Proof.* In the convex case, Theorem 1 ensures Accelerated Minibatch SGD converges at a rate proportional to

$$\frac{HB^2}{R^2} + \frac{\sigma B}{\sqrt{MKR}} \tag{193}$$

The lower bound for convex functions in Theorem 4 precisely matches this whenever

$$\frac{HB^2}{R^2} = \min\left\{\frac{\zeta_*^2}{HR^2}, \frac{HB^2}{R^2}\right\} \implies \zeta_* \geq HB \tag{194}$$

For the strongly convex case, Theorem 1 ensures convergence at a rate porportional to

$$\Delta \exp\left(-\frac{\sqrt{\lambda}R}{c_3\sqrt{H}}\right) + \frac{\sigma^2}{\lambda MKR} \tag{195}$$

The lower bound is given by

$$\min\left\{\frac{\lambda\zeta_*^2}{H^2}, \frac{\Delta\sqrt{\lambda}}{\sqrt{H}}\right\} \exp\left(-\frac{8R\sqrt{\lambda}}{\sqrt{H}}\right) + \frac{\sigma^2}{\lambda MKR} \tag{196}$$

When $\zeta_*^2 \geq H^{3/2}/\sqrt{\lambda}$, this reduces to

$$\frac{\Delta\sqrt{\lambda}}{\sqrt{H}} \exp\left(-\frac{8R\sqrt{\lambda}}{\sqrt{H}}\right) + \frac{\sigma^2}{\lambda MKR} = \Delta \exp\left(-\frac{8R\sqrt{\lambda}}{\sqrt{H}} - \log\frac{\sqrt{\lambda}}{\sqrt{H}}\right) + \frac{\sigma^2}{\lambda MKR} \tag{197}$$

Comparing this with (195), we see that the $R$ needed to guarantee error $\epsilon$ using Theorem 1 is larger than the minimum possible $R$, as lower bounded by (197), by at most a log factor. $\qquad \square$

## Footnotes

[7]This is different than the bound stated as [11, Theorem III], but is what was proven [11, Theorem VII, Appendix E].