[Reviews · NeurIPS 2020]

Review 1

Summary and Contributions: The paper addresses the problem of heterogeneous distributed optimization, i.e., when the data is split across M workers connected with some server, each worker has access to the stochastic gradients of his own (local) smooth convex objective function, and the goal is to minimize an average of these objectives. Moreover, the local objectives can be arbitrary dissimilar. Authors discuss a lot of open questions about the performance of Local-SGD, which became extremely popular during the last 1-2 years, applied to solve this problem, and, in particular, compare its performance with Minibatch SGD (MB-SGD) and Accelerated MB-SGD (either in the form of AC-SA or Multistage AC-SA) for both strongly convex and regularly convex cases. What is more, the paper proposes a new Lower Bound for the performance of Local-SGD in the heterogeneous case as well as algorithmically-independent lower bound. Finally, authors indicate the regime when Local-SGD can be provably better than Minibatch-SGD for heterogeneous data under the assumption of the uniform boundedness of local objectives dissimilarity. Moreover, the authors prove that when the dissimilarity is big, then Accelerated MB-SGD is optimal for such problems (in the class of SFO methods). ==========After Reading the Response============= I have read the response and other reviews. Thank you for providing extra details on the comparison with SCAFFOLD, I expect to see them in the final version of the paper. About \bar\zeta: the example that you provided in the response (about empirical risk) should be added in the main part of the paper since it is a reasonable motivation to consider such an assumption. About \sigma_*^2 for accelerated methods: indeed, it is not a trivial question. I still encourage authors to apply the suggestion (2) from "Weaknesses" part of my review. Taking all into account, I decided to keep my score unchanged.

Strengths: 1. First non-trivial lower bounds for heterogeneous distributed optimization were obtained: both for Local-SGD and algorithmically independent. For the complete complexity theory for federated optimization, it is highly important, especially now, when the interest in this field grows progressively. Even though there are still a lot of open questions in federated optimization, this paper makes a big step towards constructing meaningful and complete theory there. 2. The first convergence result for Local-SGD that improves over MB-SGD in heterogeneous case. However, the result was obtained uniformly bounded dissimilarity assumption that makes the analysis similar to the analysis in homogeneous case. Nevertheless, this regime is important to consider and emphasizes the limitations of existing approaches.

Weaknesses: 1. I was surprised not to see the discussion of the convergence results for SCAFFOLD (Karimireddy et al., 2019) in the paper. To the best of my knowledge, SCAFFOLD is the only known federated method that is able to converge with a linear rate in the strongly convex case when workers compute full gradients of local functions. Therefore, in order to provide a complete picture of the best-known results to this moment, it is needed to add SCAFFOLD into consideration. 2. Although the paper is mainly theoretical, the experimental part of the paper seems to be underdeveloped and does not support all the new insights suggested by the theory. For example, it would be interesting to see the experiments showing that Accelerated MB-SGD outperforms MB-SGD and Local-SGD when \zeta_*^2 is big. Next, for large K (K = 100) it is not easy to see from the presented plots how Local-SGD and MB-SGD differ in terms of achieved error when \zeta_*^2 is small.

Correctness: The proofs of the upper bounds (Theorem 1 and 3) follow the existing proofs of convergence of MB-SGD, AC-SA and Local-SGD and seem to be correct. The lower bounds also seem to be correct, although I did not have time to check all the details.

Clarity: The paper is easy to follow and presents the result in the transparent way.

Relation to Prior Work: In general, the discussion of the prior work is made clearly and makes it easy to understand how the proposed upper and lower bounds relates to the known results. However, as I mentioned before, it is very important to discuss the convergence of SCAFFOLD.

Reproducibility: Yes

Additional Feedback: 1. It would be nice to add a discussion on the difference of the variance bound that appears in the complexity results for MB-SGD and Accelerated MB-SGD. This is an important question since when the variance of the stochastic gradient is uniformly bounded one can typically prove the convergence rates depending on the smoothness constant of the expectation, not the worst one for all stochastic realizations f(x, z) like you use in eq. (5). Regarding the footnote 3 – to the best of my knowledge, it is an open question. 2. I encourage authors to add a detailed comparison with SCAFFOLD. To the best of my knowledge, it has a state-of-the-art convergence guarantees in convex and strongly convex case. 3. What is the lower bound for uniformly bounded dissimilarity case, i.e., when \zeta^2 bounds dissimilarity of local functions for all x uniformly? I did not find the discussion of this case, although the upper bound for Local-SGD was obtained for this case in Theorem 3.


Review 2

Summary and Contributions: This paper analyzes local SGD and mini batch SGD methods under heterogenous settings. The authors provide theoretical lower and upper bounds for local SGD. They improve the existing lower bounds for heterogenous data. They also show that local SGD has worse performance compared to mini batch SGD for heterogenous data which cannot be further improved. Their performance might be comparable or competitive only in homogenous or near homogenous settings. They also show accelerated minibatch SGD is optimal for high heterogeneity settings. *Post Rebuttal*: I thank the authors for addressing all my questions. I am satisfied with the author's response and updated the score accordingly.

Strengths: The paper provides rigorous theoretical results that help understand the performance of local SGD in heterogenous settings. They improve existing lower bounds for heterogenous data.

Weaknesses: The results show that local SGD is not an efficient method compared to mini batch SGD in the heterogenous settings. While I understand that local SGD has gained attention from the research community, in my opinion, to some extent, the conclusions are intuitive from consensus optimization perspective. Consensus Optimization: It is known that the bottleneck is the requirement that the nodes in the network have to communicate, and if this communication doesn't happen every iteration then the methods suffer due to additional error. There are works that analyze these settings and show how the inexactness due to lack of communication affects the overall theoretical and empirical performance. Moreover, given the existing works for local GD on heterogenous data which show that these methods have worse bounds compared to GD, to some extent, it is expected one arrives at similar conclusions to local SGD methods. Also, I believe the following article published in AISTATS 2020 is closely related to this work. “Tighter Theory for Local SGD on Identical and Heterogeneous Data” by Ahmed Khaled Konstantin Mishchenko and Peter Richt´arik.

Correctness: I haven't carefully verified all the theoretical proofs provided in the supplement material, however, I believe, the interpretations and the dependency on heterogenous and other parameters are correct.

Clarity: The paper is well written and easy to read.

Relation to Prior Work: The paper shows the relevant work and discusses how this work differs from previous contributions. However, I believe the following article published in AISTATS 2020 is closely related to this work. “Tighter Theory for Local SGD on Identical and Heterogeneous Data” by Ahmed Khaled Konstantin Mishchenko and Peter Richt´arik.

Reproducibility: Yes

Additional Feedback: Some suggestions/comments/questions • For the variances of stochastic gradients in eq(6) and eq(7), isn’t it better to write sigma^2 as distribution dependent so that the overall dependency is on average of these sigma^2’s as opposed to the maximum over all machines? • The assumption in eq (12) is strong and in the unconstrained settings, this constant on the right-hand side can be large or even unbounded for many convex functions. Therefore, although I understand that the authors intend to find regimes where local SGD bounds are better than mini-batch SGD, the regime that this bound is less than 1/R might be too restrictive. • Given the existing works of local GD for heterogenous data, local SGD for homogenous data, it would be helpful for the reader if the authors could highlight the challenges faced in providing local SGD for heterogenous data.


Review 3

Summary and Contributions: This paper considers local SGD in heterogeneous settings (where samples in different machines come from different distributions), and compares its performance against mini-batch SGD. The primary results of this paper are negative in that local SGD is strictly worse than mini-batch SGD in the heterogeneous case in most situations except when the degree of heterogeneity is somewhat controlled. The paper also presents upper bounds that achieve the prescribed speedups of local SGD over mini-batch SGD (under appropriate bounded heterogeneity assumptions), which, according to the paper is among the first of its kind. ## Update post author feedback: I have gone over points made by other reviewers and the author feedback. One after thought with regards to the case of quadratics with \sigma^* assumption is that one can consider the use of accelerated schemes that have been developed with variance at opt assumptions for quadratics [Jain et al. COLT 2018], which I believe is a significant challenge going beyond the quadratic case (as the authors note). Thank you for your good paper.

Strengths: The paper is well written and the results are described in good detail. I am generally satisfied with the work, especially in terms of its theoretical contributions.

Weaknesses: None particularly.

Correctness: In general, I believe the results are technically correct. Specifically speaking, I went over the claim of the lower bound and believe it is technically correct.

Clarity: The paper is pretty well written. One suggestion that I'd like to check with the authors is: Table 1 appears to have results that use both the variance of the stochastic gradient at opt (and otherwise), one of which is strictly a more reasonable assumption than the other, because, the global bound on the variance of the stochastic gradients implies a compactness assumption requires to be made on the parameter space. These differences in assumptions in turn makes the tabular column pretty hard to read and digest - it’d be good to rethink if the organization of this tabular column can be improved upon.

Relation to Prior Work: The authors clearly know of related work in this area, but, I do imagine the related work section can be placed in more context and strengthened to include several works in distributed SGD with “homogeneous” data with various variance assumptions on the noise and several schemes including mini-batch SGD, model-averaging and other possible schemes, for e.g., Zhang Duchi Wainwright (JMLR 2013), Dekel et al (JMLR 2009), Jain et al. (JMLR 2018), Mann et al. (Neurips ’09), and perhaps many other distributed SGD efforts that work regardless of whether the data is homogeneous or not (e.g. mini-batch SGD based methods).

Reproducibility: Yes

Additional Feedback: Can the regimes where local SGD can improve on mini-batch SGD can be potentially improved upon by examining the quadratics case with a variance at OPT style assumption?

[Author Response · NeurIPS 2020]

We thank the reviewers for their helpful feedback.

**Related Work**: We have actually already revised our paper to include a more thorough discussion of the SCAFFOLD
paper. It is important to note that SCAFFOLD is presented in a related, but different, FL setting where only a subset
$S < M$ of the machines are available in each round of communication. Specialized to our setting ($S = N$ in their
notation), the SCAFFOLD analysis actually does not show any improvement at all over MBSGD (compare their
Thm III to Table 1 in our paper). As we will describe in the final version, SCAFFOLD is like Local SGD with
variance-reduction for the inter-machine variance; this helps in the FL setting when some machines aren't available in
each round; but it does not in our setting. In addition, for the stepsizes analyzed in their theorems (specifically, very
small $\eta_l$), SCAFFOLD is actually little different MBSGD.

We compare to the Khaled et al 2020 paper mentioned by Rev #2 as ref [9], under the name of an earlier version of that
paper. We will update the citation (the relevant content is unchanged).

Regarding the homogenous case (requested by Rev #3): the paper [22] studies the homogenous case in detail and
includes most of the relevant references–we will add a comment directing to [22] as well as a brief mention of the
references. Our focus here is the difference between the homogeneous and heterogeneous settings.

**Relationship to consensus optimization**: As Rev #2 points out and as highlighted by our results, communication
between the machines is often the bottleneck in heterogeneous optimization, and consequently, (Acc) MBSGD will
often significantly outperform Local SGD.

This may be intuitive in the context of consensus optimization, but it is our experience (eg based on papers on
federated/local SGD, talks on distributed learning, and comments from other researchers) that this is far from clear to
everyone working on distributed learning. E.g., following demonstration that Local SGD can be worse than MBSGD in
the homogeneous case [22], a recurring sentiment is "well that's just the homogeneous case, in the harder heterogeneous
setting, you'll see more of an advantage for Local SGD." But we show (as Reviewer #2's intuition correctly indicates)
that this is backwards!

For this reason, we feel there is significant value in understanding and highlighting the relationship between Local SGD,
MBSGD, and other algorithms in the heterogeneous setting, and in carefully considering how the level of heterogeneity
affects the comparison. It's also important to test the limit of this intuition. E.g., we do show in Theorem 3 that in some
heterogeneous regime, additional computation as in Local SGD DOES improve over MBSGD (this is not captured nor
hinted by work we are aware of on consensus optimization).

$\bar{\zeta}$: Indeed, eq (12) is a strong assumption and the tightest bound on $\bar{\zeta}$ may be large (or infinite). However, in cases where
this is bounded and smaller than $1/R$, our analysis shows that local SGD can outperform minibatch SGD, which we feel
is useful information. As an example of where this could arise, consider the following: data is shuffled and randomly
partitioned across the M machines and the local distributions are the empirical distribution over the local sample. These
local distributions ARE heterogeneous, nevertheless, as long as there are enough samples on each machine and, for
example, the loss is Lipschitz, $\zeta$ will be bounded and small, and we can conclude that local SGD might be advantageous
over MBSGD in finding the over empirical minimizer.

$\sigma_*$ **vs** $\sigma$: As Rev #1 mentioned, we used $\sigma_*$ for the MBSGD analysis and sigma for the Accelerated MBSGD analysis.
We agree that analyzing Acc MBSGD in terms of $\sigma_*$ is interesting and valuable, but the known analysis (due to Lan) is
in terms of $\sigma$ and quite delicate, and as also acknowledged by Lan, extending it to $\sigma_*$ is a significant challenge.

**Variance definitions**: We thank Rev #2 for the suggestion that the variances in eq (6)/(7) be the average of the local
distributions' variances, this indeed gives stronger bounds and easily fits into our analysis.

**Quadratics**: Rev #3 raises an interesting question about whether the Local SGD analysis can be improved in the special
case of quadratic objectives. This is definitely possible in the homogeneous case, where Local SGD strictly dominates
MBSGD in all regimes for quadratic objectives. In the heterogeneous case, the argument does not go through in the
same way and in addition, the $\zeta_*$ term in our lower bound for Local SGD comes from a "quadratic part" of the hard
instance construction. Therefore, Local SGD won't benefit significantly from the local objectives being quadratic, at
least not in terms of the $\zeta_*$ dependence.

[Meta-Review · NeurIPS 2020]

The reviewers were unanimously in favor of accepting the paper given its contributions. The rebuttal also resolved some of the reviewer confusions. The authors are strongly encouraged to include the comparisons with Scaffold in the final version and consider the experimental suggestions raised by Reviewer 1.